# Provably Fast Finite Particle Variants of SVGD via Virtual Particle Stochastic Approximation

**Aniket Das**
Google Research
Bangalore, India
ketd@google.com

**Dheeraj Nagaraj**
Google Research
Bangalore, India
dheerajnagaraj@google.com

## Abstract

Stein Variational Gradient Descent (SVGD) is a popular particle-based variational inference algorithm with impressive empirical performance across various domains. Although the population (i.e, infinite-particle) limit dynamics of SVGD is well characterized, its behavior in the finite-particle regime is far less understood. To this end, our work introduces the notion of *virtual particles* to develop novel stochastic approximations of population-limit SVGD dynamics in the space of probability measures, that are exactly realizable using finite particles. As a result, we design two computationally efficient variants of SVGD, namely VP-SVGD and GB-SVGD, with provably fast finite-particle convergence rates. Our algorithms can be viewed as specific random-batch approximations of SVGD, which are computationally more efficient than ordinary SVGD. We show that the $n$ particles output by VP-SVGD and GB-SVGD, run for $T$ steps with batch-size $K$, are at-least as good as i.i.d samples from a distribution whose Kernel Stein Discrepancy to the target is at most $O(d^{1/3}/(KT)^{1/6})$ under standard assumptions. Our results also hold under a mild growth condition on the potential function, which is much weaker than the isoperimetric (e.g. Poincare Inequality) or information-transport conditions (e.g. Talagrand's Inequality $\mathsf{T}_1$) generally considered in prior works. As a corollary, we analyze the convergence of the empirical measure (of the particles output by VP-SVGD and GB-SVGD) to the target distribution and demonstrate a *double exponential improvement* over the best known finite-particle analysis of SVGD. Beyond this, our results present the *first known oracle complexities for this setting with polynomial dimension dependence*, thereby completely eliminating the curse of dimensionality exhibited by previously known finite-particle rates.

## 1 Introduction

Sampling from a distribution over $\mathbb{R}^d$ whose density $\pi^\star(\mathbf{x}) \propto \exp(-F(\mathbf{x}))$ is known only upto a normalizing constant, is a fundamental problem in machine learning [52, 20], statistics [41, 36], theoretical computer science [28, 17] and statistical physics [39, 15]. A popular approach to this is the Stein Variational Gradient Descent (SVGD) algorithm introduced by Liu and Wang [31], which uses a positive definite kernel $k$ to evolve a set of $n$ interacting particles $(\mathbf{x}_t^{(i)})_{i \in [n], t \in \mathbb{N}}$ as follows:

$$\mathbf{x}_{t+1}^{(i)} \leftarrow \mathbf{x}_t^{(i)} - \frac{\gamma}{n} \sum_{j=1}^n \left[ k(\mathbf{x}_t^{(i)}, \mathbf{x}_t^{(j)}) \nabla F(\mathbf{x}_t^{(j)}) - \nabla_2 k(\mathbf{x}_t^{(i)}, \mathbf{x}_t^{(j)}) \right] \tag{1}$$

SVGD exhibits remarkable empirical performance in a variety of Bayesian inference, generative modeling and reinforcement learning tasks [31, 51, 54, 22, 50, 34, 45, 19] and usually converges rapidly to the target density while using only a few particles, often outperforming Markov Chain Monte Carlo (MCMC) methods. However, in contrast to its wide practical applicability, theoretical

37th Conference on Neural Information Processing Systems (NeurIPS 2023).

analysis of the behavior SVGD is a relatively unexplored problem. Prior works on the analysis of SVGD [26, 14, 30, 42, 7] mainly consider the population limit, where the number of particles $n \to \infty$. These works assume that the initial distribution of the (infinite number of) particles has a finite KL divergence to the target $\pi^\star$ and subsequently, interpret the dynamics of population-limit SVGD as projected gradient descent updates for the KL divergence on the space of probability measures, equipped with the Wasserstein geometry. Under appropriate assumptions on the target density, one can then use the theory of Wasserstein Gradient Flows to establish non-asymptotic (in time) convergence of population-limit SVGD to $\pi^\star$ in the Kernel Stein Discrepancy (KSD) metric.

While the framework of Wasserstein Gradient Flows suffices to explain the behavior of SVGD in the population limit, the same techniques are insufficient to effectively analyze SVGD in the finite-particle regime. This is primarily due to the fact that the empirical measure $\hat{\mu}^{(n)}$ of a finite number of particles does not admit a density with respect to the Lebesgue measure, and thus, its KL divergence to the target is always infinite (i.e. $\mathsf{KL}\left(\hat{\mu}^{(n)} \| \pi^\star\right) = \infty$). In such a setting, a direct analysis of the dynamics of finite-particle SVGD becomes prohibitively difficult due to complex inter-particle dependencies. To the best of our knowledge, the pioneering work of Shi and Mackey [43] is the only result that obtains an explicit convergence rate of finite-particle SVGD by tracking the deviation between the law of $n$-particle SVGD and that of its population-limit. To this end, Shi and Mackey [43] show that for subgaussian target densities, the empirical measure of $n$-particle SVGD converges to $\pi^\star$ at a rate of $O(\sqrt{\frac{\mathsf{poly}(d)}{\log\log n^{\Theta(1/d)}}})$ in KSD [1]. The obtained convergence rate is quite slow and fails to adequately explain the impressive practical performance of SVGD.

Our work takes a starkly different approach to this problem and deliberately deviates from tracking population-limit SVGD using a finite number of particles. Instead, we directly analyze the dynamics of KL divergence along a carefully constructed trajectory in the space of distributions. To this end, our proposed algorithm, Virtual Particle SVGD (VP-SVGD) devises an *unbiased stochastic approximation (in the space of measures) of the population-limit dynamics of SVGD*. We achieve this by considering additional particles called *virtual particles* [2] which evolve in time but aren't part of the output (i.e. *real particles*). These virtual particles are used only to compute information about the current population-level distribution of the real particles, and enable exact implementation of our stochastic approximation to population-limit SVGD, while using only a finite number of particles.

Our analysis is similar in spirit to non-asymptotic analyses of stochastic gradient descent (SGD) that generally do not attempt to track gradient descent (analogous to population-limit SVGD in this case), but instead directly track the evolution of the objective function along the SGD trajectory using appropriate stochastic descent lemmas [24, 21, 11]. The key feature of our proposed stochastic approximation is the fact that it can be implemented using only a finite number of particles. This allows us to design faster variants of SVGD with provably fast finite-particle convergence.

## 1.1 Contributions

**VP-SVGD and GB-SVGD** We propose two variants of SVGD that enjoy provably fast finite-particle convergence guarantees to the target distribution: Virtual Particle SVGD (VP-SVGD in Algorithm 1) and Global Batch SVGD (GB-SVGD in Algorithm 2). VP-SVGD is a conceptually elegant stochastic approximation (in the space of probability measures) of population-limit SVGD, and GB-SVGD is a practically efficient version of SVGD which achieves good empirical performance. Our analysis of GB-SVGD builds upon that of VP-SVGD. When the potential $F$ is smooth and satisfies a quadratic growth condition (which holds under subgaussianity of $\pi^\star$, a common assumption in prior works [42, 43]), we show that the $n$ particles output by $T$ steps of our algorithms, run with batch-size $K$, are at least as good as i.i.d draws from a distribution whose Kernel Stein Discrepancy to $\pi^\star$ is at most $O(d^{1/3}/(KT)^{1/6})$. Our results also hold under a mild subquadratic growth condition for $F$, which is much weaker than isoperimetric (e.g. Poincare Inequality) or information-transport (e.g. Talagrand's Inequality $\mathsf{T}_1$) assumptions generally considered in the sampling literature [47, 42, 43, 8, 2].

**State-of-the-art Finite Particle Guarantees** As corollaries of the above result, we establish that *VP-SVGD and GB-SVGD exhibit the best known finite-particle guarantees in the literature which significantly outperform that of prior works*. Our results are summarized in Table 1. In particular,

---

[1] We explicate the dimension dependence in Shi and Mackey [43] by closely following their analysis

[2] (roughly) analogous to virtual particles in quantum field theory that enable interactions between real particles

| Result | Algorithm | Assumption | Rate | Oracle Complexity |
|---|---|---|---|---|
| Korba et al. [26] | Population Limit SVGD | Uniformly Bounded $\mathsf{KSD}_{\pi^\star}(\bar{\mu}_t\|\pi^\star)$ | $\frac{\mathrm{poly}(d)}{\sqrt{T}}$ | Not Implementable |
| Salim et al. [42] | Population Limit SVGD | Sub-gaussian $\pi^\star$ | $\frac{d^{3/2}}{\sqrt{T}}$ | Not Implementable |
| Shi and Mackey [43] | SVGD | Sub-gaussian $\pi^\star$ | $\frac{\mathrm{poly}(d)}{\sqrt{\log\log n^{\Theta(1/d)}}}$ | $\frac{\mathrm{poly}(d)}{\epsilon^2}e^{\Theta(de^{\mathrm{poly}(d)/\epsilon^2})}$ |
| **Ours, Corollary 1** | **VP-SVGD** | **Sub-gaussian $\pi^\star$** | $(d/n)^{1/4}+(d/n)^{1/2}$ | $d^4/\epsilon^{12}$ |
| **Ours, Corollary 1** | **GB-SVGD** | **Sub-gaussian $\pi^\star$** | $d^{1/3}/n^{1/12}+(d/n)^{1/2}$ | $d^6/\epsilon^{18}$ |
| **Ours, Corollary 1** | **VP-SVGD** | **Sub-exponential $\pi^\star$** | $\frac{d^{1/3}}{n^{1/6}}+\frac{d}{n^{1/2}}$ | $d^6/\epsilon^{16}$ |
| **Ours, Corollary 1** | **GB-SVGD** | **Sub-exponential $\pi^\star$** | $\frac{d^{3/8}}{n^{1/16}}+\frac{d}{n^{1/2}}$ | $d^9/\epsilon^{24}$ |

Table 1: Comparison of our results with prior works. $d$, $T$, and $n$ denote the dimension, no. of iterations and no. of output particles respectively. Oracle Complexity denotes number of evaluations of $\nabla F$ needed to achieve $\mathsf{KSD}_{\pi^\star}(\cdot\|\pi^\star) \leq \epsilon$ and Rate denotes convergence rate w.r.t KSD metric. Note that: 1. Population Limit SVGD is not implementable as it requires infinite particles 2. The uniformly bounded $\mathsf{KSD}_{\pi^\star}(\bar{\mu}_t\|\pi^\star)$ assumption cannot be verified apriori and is much stronger than subgaussianity (see [42] Lemma C.1)

under subgaussianity of the target distribution $\pi^\star$, we show that the empirical measure of the $n$ particles output by VP-SVGD converges to $\pi^\star$ in KSD at a rate of $O((d/n)^{1/4}+(d/n)^{1/2})$. Similarly, the empirical measure of the $n$ output particles of GB-SVGD converges to $\pi^\star$ at a KSD rate of $O(d^{1/3}/n^{1/12}+(d/n)^{1/2})$. Both these results represent a **double exponential improvement** over the $O(\frac{\mathrm{poly}(d)}{\sqrt{\log\log n^{\Theta(1/d)}}})$ KSD rate of $n$-particle SVGD obtained by Shi and Mackey [43], which, to our knowledge, is the best known prior result for SVGD in the finite particle regime. When benchmarked in terms of gradient oracle complexity, i.e., the number of evaluations of $\nabla F$ required by an algorithm to achieve $\mathsf{KSD}_{\pi^\star}(\cdot\|\pi^\star) \leq \epsilon$, we demonstrate that for subgaussian $\pi^\star$, the oracle complexity of VP-SVGD is $O(d^4/\epsilon^{12})$ while that of GB-SVGD is $O(d^6/\epsilon^{18})$. To the best of our knowledge, our result presents the *first known oracle complexity guarantee with polynomial dimension dependence*, and consequently, does not suffer from a curse of dimensionality unlike prior works. Furthermore, as discused above, the conditions under which our result holds is far weaker than subgaussianity of $\pi^\star$, and as such, includes sub-exponential targets and beyond. In particular, *our guarantees for sub-exponential target distributions are (to the best of our knowledge) the first of its kind.*

**Computational Benefits:** VP-SVGD and GB-SVGD can be viewed as specific random batch approximations of SVGD. Our experiments (Section 8) show that GB-SVGD obtains similar performance as SVGD but requires fewer computations. In this context, a different kind of random batch method that divides the particles into random subsets of interacting particles, has been proposed by Li et al. [29]. However, the objective in Li et al. [29] is to approximate finite-particle SVGD dynamics using the random batch method, instead of analyzing convergence of the random batch method itself. Beyond this, their guarantees also suffer from an exponential dependence on the time $T$. As explained below, their approach is also conceptually different from our method since we use the *same* random batch to evolve *every* particle, allowing us to interpret this as a stochastic approximation in the space of distributions instead of in the path space.

## 1.2 Technical Challenges

We resolve the following important conceptual challenges, which may be of independent interest.

**Stochastic Approximation in the Space of Probability Measures** Stochastic approximations are widely used in optimization, control and sampling [27, 52, 23]. In the context of sampling, stochastic approximations are generally implemented in path space, e.g., Stochastic Gradient Langevin Dynamics (SGLD) [52] takes a random batch approximation of the drift term via the update $\mathbf{x}_{t+1} = \mathbf{x}_t - \frac{\eta}{K}\sum_{j=0}^{K-1}\nabla f(\mathbf{x}_t,\xi_j) + \sqrt{2\eta}\epsilon_t$, $\epsilon_t \sim \mathcal{N}(0,\mathbf{I})$ where $\mathbb{E}[f(\mathbf{x}_t,\xi_j)|\mathbf{x}_t] = F(\mathbf{x}_t)$. Such stochastic approximations are then analyzed using the theory of stochastic processes over $\mathbb{R}^d$ [12, 40, 55, 25]. However, when viewed in the space of probability measures (i.e, $\mu_t = \mathrm{Law}(\mathbf{x}_t)$), the time-evolution

of these algorithms is deterministic. In contrast, our approach designs *stochastic approximations in the space of probability measures*. In particular, the time-evolution of the law of any particle in VP-SVGD and GB-SVGD are a stochastic approximation of the dynamics of *population-limit* SVGD. We ensure that requires only a finite number of particles for exact implementation.

**Tracking KL Divergence in the Finite-Particle Regime** The population limit ($n \to \infty$) ensures that the initial empirical distribution ($\mu_0$) of SVGD admits a density (w.r.t the Lebesgue measure). When the KL divergence between $\mu_0$ and $\pi^\star$ is finite, prior works on population-limit SVGD analyze the time-evolution of the KL divergence to $\pi^\star$. However, this approach cannot be directly used to analyze finite-particle SVGD since the empirical distribution of a finite number of particles does not admit a density, and thus its KL divergence to $\pi^\star$ is infinite. Our analysis of VP-SVGD and GB-SVGD circumvents this obstacle by considering the dynamics of an infinite number of particles, whose empirical measure then admits a density. However, the careful design ensures that the dynamics of $n$ of these particles can be computed exactly, using only a finite total number of (real + virtual) particles. When conditioned on the virtual particles, these particles are i.i.d. and their conditional law is close to the target distribution with high probability.

## 2 Notation and Problem Setup

We use $\|\cdot\|, \langle\cdot,\cdot\rangle$ to denote the Euclidean norm and inner product over $\mathbb{R}^d$ respectively. All other norms and inner products are subscripted to indicate their underlying space. $\mathcal{P}_2(\mathbb{R}^d)$ denotes the space of probability measures on $\mathbb{R}^d$ with finite second moment, with the Wasserstein-2 metric denoted as $\mathcal{W}_2(\mu,\nu)$ for $\mu,\nu \in \mathcal{P}_2(\mathbb{R}^d)$. For any two probability measures $\mu,\nu$, we denote their KL divergence as $\mathsf{KL}(\mu\|\nu)$. For any function $f : X \to Y$ and any probability measure $\mu$ over $X$, we let $f_\#\mu$ denote the law of $f(\mathbf{x}) : \mathbf{x} \sim \mu$. Given a sigma algebra $\mathcal{F}$ over some space $\Omega$, and a measurable space $\mathcal{X}$, $\mu(\cdot;\cdot) : \mathcal{F} \times \mathcal{X} \to \mathbb{R}^+$ is a probability kernel if for every $\mathbf{x} \in \mathcal{X}$, $\mu(\cdot;\mathbf{x})$ is a measure over $\mathcal{F}$ and for every $A \in \mathcal{F}$, the map $\mathbf{x} \to \mu(A;\mathbf{x})$ is measurable. We make use of probability measures $\mu(\cdot;\mathbf{x})$ where $\mathbf{x}$ is a random element of some appropriate space $\mathcal{X}$, resulting in random probability measures. We use $[m]$ and $(m)$ to denote the sets $\{1,\ldots,m\}$ and $\{0,\ldots,m-1\}$ respectively. For any finite set $A$, $\mathbb{S}_A$ denotes the group of all permutations of $A$. We use the $O$ notation to characterize the dependence of our rates on the number of iterations $T$, dimension $d$ and batch-size $K$, suppressing numerical and problem-dependent constants. We use $\lesssim$ to denote $\leq$ upto universal constants.

We fix a symmetric positive definite reproducing kernel $k : \mathbb{R}^d \times \mathbb{R}^d \to \mathbb{R}$ and let the corresponding reproducing kernel Hilbert space (RKHS) [44] be denoted as $\mathcal{H}_0$. We denote the product RKHS as $\mathcal{H} = \prod_{i=1}^d \mathcal{H}_0$, equipped with the standard inner product for product spaces. We assume $k$ is differentiable in both its arguments and let $\nabla_2 k(\mathbf{x},\mathbf{y})$ denote the gradient of $k(\cdot,\cdot)$ with respect to the second argument. For any $\mu \in \mathcal{P}_2(\mathbb{R}^d)$, we assume $\mathcal{H} \subset L^2(\mu)$ and the inclusion map $i_\mu : \mathcal{H} \to L^2(\mu)$ is continuous. We use $P_\mu : L^2(\mu) \to \mathcal{H}$ to denote the adjoint of $i_\mu$, i.e., the unique operator which satisfies $\langle f, i_\mu g\rangle_{L^2(\mu)} = \langle P_\mu f, g\rangle_{\mathcal{H}}$ for any $f \in L^2(\mu), g \in \mathcal{H}$. Carmeli et al. [6] shows that $P_\mu$ can be expressed as a kernel convolution, i.e., $(P_\mu f)(\mathbf{x}) = \int k(\mathbf{x},\mathbf{y})f(\mathbf{y})\mathrm{d}\mu(\mathbf{y})$

We define the function $h : \mathbb{R}^d \times \mathbb{R}^d \to \mathbb{R}$ as $h(\mathbf{x},\mathbf{y}) = k(\mathbf{x},\mathbf{y})\nabla F(\mathbf{y}) - \nabla_2 k(\mathbf{x},\mathbf{y})$ and $h_\mu \in \mathcal{H}$ as $h_\mu = P_\mu(\nabla_\mathbf{x} \log(\frac{\mathrm{d}\mu}{\mathrm{d}\pi^\star}(\mathbf{x})))$ for any $\mu \in \mathcal{P}_2(\mathbb{R}^d)$. Integration by parts shows that $h_\mu(\mathbf{x}) = \int h(\mathbf{x},\mathbf{y})\mathrm{d}\mu(\mathbf{y})$. The convergence metric we use is the Kernel Stein Discrepancy (KSD) metric, which is widely used for comparing probability distributions [32, 9] and analyzing SVGD [42, 26].

**Definition 1** (Kernel Stein Discrepancy)**.** *Define the Langevin Stein Operator of $\pi^\star$ acting on any differentiable $g : \mathbb{R}^d \to \mathbb{R}^d$:*

$$(T_{\pi^\star}g)(\mathbf{x}) = \nabla \cdot g(\mathbf{x}) - \langle\nabla F(\mathbf{x}), g(\mathbf{x})\rangle$$

*For any two probability measures $\mu,\nu$, the Kernel Stein Discrepancy between $\mu$ and $\nu$ (with respect to $\pi^\star$), denoted as $\mathsf{KSD}_{\pi^\star}(\mu\|\nu)$ is defined as*

$$\mathsf{KSD}_{\pi^\star}(\mu\|\nu) = \sup_{\|g\|_{\mathcal{H}} \leq 1} \mathbb{E}_\mu[T_{\pi^\star}g] - \mathbb{E}_\nu[T_{\pi^\star}g]$$

*Using integration by parts (see Chwialkowski et al. [9]), it follows that $\mathsf{KSD}_{\pi^\star}(\mu\|\nu) = \|h_\mu - h_\nu\|_{\mathcal{H}}$*

**Organization:** We review population-limit SVGD in Section 3 and derive VP-SVGD and GB-SVGD in Section 4. We state our technical assumptions in Section 5, main results in Section 6 and provide a proof sketch in Section 7. We present empirical evaluation in Section 8.

## 3 Background on Population-Limit SVGD

We briefly introduce the analysis of population-limit SVGD using the theory of Wasserstein Gradient Flows and refer the readers to Korba et al. [26] and Salim et al. [42] for a detailed treatment.

The space $\mathcal{P}_2(\mathbb{R}^d)$ equipped with the 2-Wasserstein metric $\mathcal{W}_2$ is known as the Wasserstein space, which admits the following Riemannian structure : For any $\mu \in \mathcal{P}_2(\mathbb{R}^d)$, the tangent space $T_\mu \mathcal{P}_2(\mathbb{R}^d)$ can be identified with the Hilbert space $L^2(\mu)$. We can then define differentiable functionals $\mathcal{L} : \mathcal{P}_2(\mathbb{R}^d) \to \mathbb{R}$ and compute their Wasserstein gradients, denoted as $\nabla_{\mathcal{W}_2}\mathcal{L}$. Note that the target $\pi^\star$ is the unique minimizer over $\mathcal{P}_2(\mathbb{R}^d)$ for the functional $\mathcal{L}[\mu] = \mathsf{KL}\,(\mu\|\pi^\star)$. The Wasserstein Gradient of $\mathcal{L}[\mu]$ is $\nabla_{\mathcal{W}_2}\mathcal{L}[\mu] = \nabla_{\mathbf{x}} \log(\frac{d\mu}{d\pi^\star}(\mathbf{x}))$ [1]. This powerful machinery has served as a backbone for the analysis of algorithms such as LMC [53, 3, 2] and population-limit SVGD [14, 26, 42, 45, 7].

The updates of population-limit SVGD can be viewed as Projected Gradient Descent in the Wasserstein space. Recall from Section 2 that the function $h_\mu(\mathbf{x}) = P_\mu(\nabla \log(\frac{d\mu}{d\pi^\star}))(\mathbf{x}) = \int h(\mathbf{x}, \mathbf{y})d\mu(\mathbf{y})$. Let $\hat{\mu}_t^n$ denote the empirical measures of the SVGD particles $(\mathbf{x}_t^{(i)})_{i\in[n]}$ at timestep $t$. We note that the SVGD updates in (1) can be recast as $\hat{\mu}_{t+1}^n = (I - \gamma h_{\hat{\mu}_t^n})_\# \hat{\mu}_n^t$. In the limit of infinite particles $n \to \infty$, suppose the empirical measure $\hat{\mu}_t^n$ converges to the population measure $\bar{\mu}_t$. In this population limit, the updates of SVGD can be expressed as,

$$\bar{\mu}_{t+1} = (I - h_{\bar{\mu}_t})_\# \bar{\mu}_t = \left(I - \gamma P_{\bar{\mu}_t}\left(\nabla \log(\frac{d\bar{\mu}_t}{d\pi^\star})\right)\right)_\# \bar{\mu}_t = (I - \gamma P_{\bar{\mu}_t}(\nabla_{\mathcal{W}_2}\mathsf{KL}\,(\bar{\mu}_t\|\pi^\star)))_\# \bar{\mu}_t$$

Recall from Section 2 that $P_{\bar{\mu}_t} : L^2(\bar{\mu}_t) \to \mathcal{H}$ is the Hilbert adjoint of $i_{\bar{\mu}_t}$. Since $\mathcal{H} \subset L^2(\bar{\mu}_t)$, the updates of SVGD in the population limit can be seen as Projected Wasserstein Gradient Descent for $\mathcal{L}[\mu] = \mathsf{KL}\,(\mu\|\pi^\star)$, with the Wasserstein Gradient at each step being projected onto the RKHS $\mathcal{H}$. Assuming $\mathsf{KL}\,(\bar{\mu}_0\|\pi^\star) < \infty$, convergence of population limit SVGD is then established by tracking the evolution of $\mathsf{KL}\,(\bar{\mu}_t\|\pi^\star)$ under appropriate structural assumptions (such as subgaussianity) on $\pi^\star$.

## 4 Algorithm and Intuition

In this section, we derive VP-SVGD (Algorithm 1), and build upon it to obtain GB-SVGD. Consider a countably infinite collection of particles $\mathbf{x}_0^{(l)} \in \mathbb{R}^d$, $l \in \mathbb{N} \cup \{0\}$, sampled i.i.d from a measure $\mu_0$, having a density w.r.t. the Lebesgue measure. By the strong law of large numbers, the empirical measure of $\mathbf{x}_0^{(l)}$ is almost surely equal to $\mu_0$ (see Dudley [13, Theorem 11.4.1]). Let batch size $K \in \mathbb{N}$ denote the batch size, and $\mathcal{F}_t$ denote the filtration $\mathcal{F}_t$, $t \geq 0$ as $\mathcal{F}_t = \sigma(\{\mathbf{x}_0^{(l)} \mid l \leq Kt-1\})$, $\forall\, t \in \mathbb{N}$, with $\mathcal{F}_0$ being the trivial $\sigma$ algebra. For ease of exposition, we discuss the case of $K = 1$ in this section and present a complete derivation for arbitrary $K \geq 1$ in Appendix C. Recall from Section 3 that the updates of population-limit SVGD in $\mathcal{P}_2(\mathbb{R}^d)$ can be expressed as follows:

$$\bar{\mu}_{t+1} = (I - \gamma h_{\bar{\mu}_t})_\# \bar{\mu}_t \tag{2}$$

We aim to design a stochastic approximation in $\mathcal{P}_2(\mathbb{R}^d)$ for the updates (2), such that it admits a finite-particle realization. To this end, we propose the following dynamics in $\mathbb{R}^d$

$$\mathbf{x}_{t+1}^{(s)} = \mathbf{x}_t^{(s)} - \gamma h(\mathbf{x}_t^{(s)}, \mathbf{x}_t^{(t)}), \quad s \in \mathbb{N} \cup \{0\} \tag{3}$$

Now, for each time-step $t$, we focus on the time evolution of the particles $(\mathbf{x}_t^{(l)})_{l \geq t}$ (called the *lower triangular evolution*). From (3), we observe that for any $t \in \mathbb{N}$ and $l \geq t$, $\mathbf{x}_t^{(l)}$ depends only on $\mathbf{x}_0^{(0)}, \ldots, \mathbf{x}_0^{(t-1)}, \mathbf{x}_0^{(l)}$. Hence, there exists a deterministic, measurable function $H_t$ such that:

$$\mathbf{x}_t^{(l)} = H_t(\mathbf{x}_0^{(0)}, \ldots, \mathbf{x}_0^{(t-1)}, \mathbf{x}_0^{(l)}); \quad \text{for every } l \geq t \tag{4}$$

Since $\mathbf{x}_0^{(0)}, \ldots, \mathbf{x}_0^{(t-1)}, \mathbf{x}_0^{(l)} \overset{i.i.d.}{\sim} \mu_0$, we conclude from (4) that $(\mathbf{x}_t^{(l)})_{l \geq t}$ are i.i.d when conditioned on $\mathbf{x}_0^{(0)}, \ldots, \mathbf{x}_0^{(t-1)}$. To this end, we define the random measure $\mu_t | \mathcal{F}_t$ as the law of $\mathbf{x}_t^{(t)}$ conditioned on $\mathcal{F}_t$, i.e., $\mu_t | \mathcal{F}_t$ is a probability kernel $\mu_t(\cdot\,; \mathbf{x}_0^{(0)}, \ldots, \mathbf{x}_0^{(t-1)})$, where $\mu_0 | \mathcal{F}_0 := \mu_0$. By the strong law of large numbers, $\mu_t | \mathcal{F}_t$ is equal to the empirical measure of $(\mathbf{x}_t^{(l)})_{l \geq t}$ conditioned on $\mathcal{F}_t$. We will use $\mu_t | \mathcal{F}_t$ and $\mu_t(\cdot\,; \mathbf{x}_0^{(0)}, \ldots, \mathbf{x}_0^{(t-1)})$ interchangeably.

Define the random function $g_t : \mathbb{R}^d \to \mathbb{R}^d$ as $g_t(\mathbf{x}) := h(\mathbf{x}, \mathbf{x}_t^{(t)})$. From (4), we note that $g_t$ is $\mathcal{F}_{t+1}$ measurable. From (3), we infer that the particles satisfy the following relation:

$$\mathbf{x}_{t+1}^{(s)} = (I - \gamma g_t)(\mathbf{x}_t^{(s)}), \quad s \geq t+1$$

Recall that $\mathbf{x}_{t+1}^{(s)}|\mathbf{x}_0^{(0)}, \ldots, \mathbf{x}_0^{(t)} \sim \mu_{t+1}|\mathcal{F}_{t+1}$ for any $s \geq t+1$. Furthermore, from Equation (4), we note that for $s \geq t+1$, $\mathbf{x}_t^{(s)}$ depends only on $\mathbf{x}_0^{(0)}, \ldots, \mathbf{x}_0^{(t-1)}$ and $\mathbf{x}_0^{(s)}$. Hence, we conclude that $\mathrm{Law}(\mathbf{x}_t^{(s)}|\mathbf{x}_0^{(0)}, \ldots, \mathbf{x}_0^{(t)}) = \mathrm{Law}(\mathbf{x}_t^{(s)}|\mathbf{x}_0^{(0)}, \ldots, \mathbf{x}_0^{(t-1)}) = \mu_t|\mathcal{F}_t$. With this insight, the dynamics of the lower-triangular evolution in $\mathcal{P}_2(\mathbb{R}^d)$ that the following holds almost surely:

$$\mu_{t+1}|\mathcal{F}_{t+1} = (I - \gamma g_t)_{\#}\mu_t|\mathcal{F}_t \tag{5}$$

$\mathbf{x}_t^{(t)}|\mathcal{F}_t \sim \mu_t|\mathcal{F}_t$ implies $\mathbb{E}[g_t(\mathbf{x})|\mathcal{F}_t] = h_{\mu_t|\mathcal{F}_t}(\mathbf{x})$. Thus *lower triangular dynamics* (5) is a stochastic approximation in $\mathcal{P}_2(\mathbb{R}^d)$ to the population limit of SVGD (2). Setting the batch size to general $K$ and tracking the evolution of the first $KT + n$ particles, we obtain VP-SVGD (Algorithm 1).

---

**Algorithm 1** Virtual Particle SVGD (`VP-SVGD`)

---

**Input**: Number of steps $T$, number of output particles $n$, batch size $K$, Initial positions $\mathbf{x}_0^{(0)}, \ldots, \mathbf{x}_0^{(n+KT-1)} \overset{i.i.d.}{\sim} \mu_0$, Kernel $k$, step size $\gamma$.

1: **for** $t \in \{0, \ldots, T-1\}$ **do**
2:     **for** $s \in \{0, \ldots, KT+n-1\}$ **do**
3:         $\mathbf{x}_{t+1}^{(s)} = \mathbf{x}_t^{(s)} - \frac{\gamma}{K}\sum_{l=0}^{K-1}[k(\mathbf{x}_t^{(s)}, \mathbf{x}_t^{(tK+l)})\nabla F(\mathbf{x}_t^{(tK+l)}) - \nabla_2 k(\mathbf{x}_t^{(s)}, \mathbf{x}_t^{(tK+l)})]$
4:     **end for**
5: **end for**
6: Draw $S$ uniformly at random from $\{0, \ldots, T-1\}$
7: Output $(\mathbf{y}^{(0)}, \ldots, \mathbf{y}^{(n-1)}) = (\mathbf{x}_S^{(TK)}, \ldots, \mathbf{x}_S^{(TK+n-1)})$

---

**Virtual Particles** In Algorithm 1, $(\mathbf{x}_t^{(l)})_{KT \leq l \leq KT+n-1}$ are the *real particles* which constitute the output. $(\mathbf{x}_t^{(l)})_{l < KT}$ are *virtual particles* which propagate information about the probability measure $\mu_t|\mathcal{F}_t$ to enable computation of $g_t$, an unbiased estimate of the projected Wasserstein gradient $h_{\mu_t|\mathcal{F}_t}$.

**VP-SVGD as SVGD Without Replacement** VP-SVGD is a without-replacement random-batch approximation of SVGD (1), where a different batch is used across timesteps, but the same batch is across particles given a fixed timestep. With i.i.d. initialization, picking the 'virtual particles' in a fixed order or from a random permutation does not change the evolution of the real particles. With this insight, we design GB-SVGD (Algorithm 2) where we consider $n$ particles *and* output $n$ particles (instead of wasting $KT$ particles as 'virtual particles') via a random-batch approximation of SVGD.

---

**Algorithm 2** Global Batch SVGD (`GB-SVGD`)

---

**Input**: # of time steps $T$, # of particles $n$, $\mathbf{x}_0^{(0)}, \ldots, \mathbf{x}_0^{(n-1)} \overset{i.i.d.}{\sim} \mu_0$, Kernel $k$, step size $\gamma$, Batch size $K$, Sampling method $\in \{\text{with replacement, without replacement}\}$

1: **for** $t \in \{0, \ldots, T-1\}$ **do**
2:     $\mathcal{K}_t \leftarrow$ random subset of $[n]$ of size $K$ (via. sampling method)
3:     **for** $s \in \{0, \ldots, n-1\}$ **do**
4:         $\mathbf{x}_{t+1}^{(s)} = \mathbf{x}_t^{(s)} - \frac{\gamma}{K}\sum_{r \in \mathcal{K}_t}[k(\mathbf{x}_t^{(s)}, \mathbf{x}_t^{(r)})\nabla F(\mathbf{x}_t^{(r)}) - \nabla_2 k(\mathbf{x}_t^{(s)}, \mathbf{x}_t^{(r)})]$
5:     **end for**
6: **end for**
7: Draw $S$ uniformly at random from $\{0, 1, \ldots, T-1\}$
8: Output $(\bar{\mathbf{y}}^{(0)}, \ldots, \bar{\mathbf{y}}^{(n-1)}) = (\mathbf{x}_S^{(0)}, \ldots, \mathbf{x}_S^{(n-1)})$

---

In Algorithm 2, with replacement sampling means selecting a batch of $K$ particles i.i.d. from the uniform distribution over $[n]$. Without replacement sampling means fixing a random permutation $\sigma$ over $\{0, \ldots, n-1\}$ and selecting the batches in the order specified by the permutation.

# 5  Assumptions

In this section, we discuss the key assumptions required for our analysis of VP-SVGD and GB-SVGD. Our first assumption is smoothness of $F$, which is standard in optimization and sampling.

**Assumption 1** (L-Smoothness). *$\nabla F$ exists and is $L$ Lipschitz. Moreover $\|\nabla F(0)\| \leq \sqrt{L}$.*

It is easy find a point such that $\|\nabla F(\mathbf{x}^*)\| \leq \sqrt{L}$ (e.g., using $\Theta(1)$ gradient descent steps [37]) and center the initialization at $\mu_0$ at $\mathbf{x}^*$. For clarity, we take $\mathbf{x}^* = 0$ without loss of generality. We now impose the following growth condition on $F$.

**Assumption 2** (Growth Condition). *There exist $\alpha, d_1, d_2 > 0$ such that*

$$F(\mathbf{x}) \geq d_1 \|\mathbf{x}\|^\alpha - d_2 \quad \forall \mathbf{x} \in \mathbb{R}^d$$

Note that Assumption 1 ensures $\alpha \leq 2$. Assumption 2 is essentially a tail decay assumption on the target density $\pi^\star(\mathbf{x}) \propto e^{-F(\mathbf{x})}$. In fact, as we shall show in Appendix B, Assumption 2 ensures that the tails of $\pi^\star$ decay as $\propto e^{-\|\mathbf{x}\|^\alpha}$. Consequently, Assumption 2 holds with $\alpha = 2$ when $\pi^\star$ is subgaussian and with $\alpha = 1$ when $\pi^\star$ is subexponential. Subgaussianity is equivalent to $\pi^\star$ satisfying the $\mathsf{T}_1$ inequality [5, 49], commonly assumed in prior works on SVGD [42, 43]. We also note that subexponentiality is implied when $\pi^\star$ satisfies the Poincare Inequality [4, Section 4], which is considered a mild condition in the sampling literature [47, 8, 2, 12, 7]. This makes Assumption 1 significantly weaker than the isoperimetric or information-transport assumptions considered in prior works.

Next, we impose a mild assumption on the RKHS of the kernel $k$, which has been used by several prior works [42, 26, 45, 43].

**Assumption 3** (Bounded RKHS Norm). *For any $\mathbf{y} \in \mathbb{R}^d$, $k(\cdot, \mathbf{y})$ satisfies $\|k(\cdot, \mathbf{y})\|_{\mathcal{H}_0} \leq B$. Furthermore, $\nabla_2 k(\cdot, \mathbf{y}) \in \mathcal{H}$ and $\|\nabla_2 k(\cdot, \mathbf{y})\|_{\mathcal{H}} \leq B$*

Assumption 3 ensures that the adjoint operator $P_\mu$, used in Sections 2 and 3, is well-defined. We also make the following assumptions on the kernel $k$, which is satisfied by a large class of standard kernels such as Radial Basis Function kernels and Matérn kernels of order $\geq 3/2$.

**Assumption 4** (Kernel Decay). *The kernel $k$ satisfies the following for constants $A_1, A_2, A_3 > 0$.*

$$0 \leq k(\mathbf{x}, \mathbf{y}) \leq \frac{A_1}{1 + \|\mathbf{x} - \mathbf{y}\|^2}, \quad \|\nabla_2 k(\mathbf{x}, \mathbf{y})\| \leq A_2, \quad \|\nabla_2 k(\mathbf{x}, \mathbf{y})\|^2 \leq A_3 k(\mathbf{x}, \mathbf{y})$$

Finally, we make the following mild assumption on the initialization.

**Assumption 5** (Initialization). *The initial distribution $\mu_0$ is such that $\mathsf{KL}(\mu_0 \| \pi^\star) < \infty$. Furthermore, $\mu_0$ is supported in $\mathcal{B}(R)$, the $\ell_2$ ball of radius $R$*

Since prior works usually assume Gaussian initialization [42, 47], Assumption 5 may seem slightly non-standard. However, this is not a drawback. In fact, whenever $R = \Theta(\sqrt{d} + \text{polylog}(n/\delta))$, Gaussian initialization can be made indistinguishable from $\mathsf{Uniform}(\mathcal{B}(R))$ initialization, with probability at least $1 - \delta$, via a coupling argument. To this end, we impose Assumption 5 for ease of exposition and our results can be extended to consider Gaussian initialization. In Appendix B we show that taking $R = \sqrt{d/L}$ and $\mu_0 = \mathsf{Uniform}(\mathcal{B}(R))$ suffices to ensure $\mathsf{KL}(\mu_0 \| \pi^\star) = O(d)$.

# 6  Results

## 6.1  VP-SVGD

Our first result, proved in Appendix C, shows that the law of the *real particles of* VP-SVGD , when conditioned on the virtual particles, is close to $\pi^\star$ in KSD. As a consequence, it shows that the particles output by VP-SVGD are i.i.d. samples from a random probability measure $\bar{\mu}(\cdot; \mathbf{x}_0^{(0)}, \ldots, \mathbf{x}_0^{(KT-1)}, S)$ which is close to $\pi^\star$ in KSD. We also present a high-probability version of this result in Appendix C.

**Theorem 1** (**Convergence of VP-SVGD**). *Let $\mu_t$ be as defined in Section 4. Let Assumptions 1 2, 3, 4, and 5 be satisfied and let $\gamma \leq \min\{1/2A_1L, 1/(4+L)B\}$. There exist $(\zeta_i)_{0 \leq i \leq 3}$ depending polynomially*

*on $A_1, A_2, A_3, B, L, d_1, d_2$ for any fixed $\alpha \in (0,2]$, such that whenever $\gamma\xi \leq \frac{1}{2B}$, with $\xi = \zeta_0 + \zeta_1(\gamma T)^{1/\alpha} + \zeta_2(\gamma^2 T)^{1/\alpha} + \zeta_3 R^{2/\alpha}$, the following holds:*

$$\frac{1}{T}\sum_{t=0}^{T-1}\mathbb{E}\left[\mathsf{KSD}^2_{\pi^\star}(\mu_t|\mathcal{F}_t\|\pi^\star)\right] \leq \frac{2\mathsf{KL}(\mu_0\|\pi^\star)}{\gamma T} + \frac{\gamma B(4+L)\xi^2}{K}$$

*Define the probability kernel $\bar{\mu}(\cdot \ ; \cdot)$ as follows: For any $x_\tau \in \mathbb{R}^d$, $\tau \in (KT)$ and $s \in (T)$, $\bar{\mu}(\cdot \ ; x_0, \ldots, x_{KT-1}, s) := \mu_s(\cdot \ ; x_0, \ldots, x_{Ks-1})$ and $\bar{\mu}(\cdot \ ; x_0, \ldots, x_{KT-1}, s = 0) := \mu_0(\cdot)$. Conditioned on $\mathbf{x}_\tau^{(0)} = x_\tau$, $S = s$ for every $\tau \in (KT)$, the outputs $\mathbf{y}^{(0)}, \ldots, \mathbf{y}^{(n-1)}$ of VP-SVGD are i.i.d samples from $\bar{\mu}(\cdot \ ; x_0, \ldots, x_{KT-1}, s)$. Furthermore,*

$$\mathbb{E}[\mathsf{KSD}^2_{\pi^\star}(\bar{\mu}(\cdot \ ; \mathbf{x}_0^{(0)}, \ldots, \mathbf{x}_0^{(KT-1)}, S)\|\pi^\star)] \leq \frac{2\mathsf{KL}(\mu_0\|\pi^\star)}{\gamma T} + \frac{\gamma B(4+L)\xi^2}{K}$$

**Convergence Rates** Taking $\mu_0 = \mathsf{Uniform}(\mathcal{B}(R))$ with $R = \sqrt{d/L}$ ensures $\mathsf{KL}(\mu_0|\mathcal{F}_0\|\pi^\star) = O(d)$ (see Appendix B). Under this setting, choosing $\gamma = O(\frac{(Kd)^\eta}{T^{1-\eta}})$ ensures that $\mathbb{E}[\mathsf{KSD}^2_{\pi^\star}(\bar{\mu}\|\pi^\star)] = O(\frac{d^{1-\eta}}{(KT)^\eta})$ where $\eta = \frac{\alpha}{2(1+\alpha)}$. Thus, for $\alpha = 2$, (i.e, sub-Gaussian $\pi^\star$), $\mathsf{KSD}^2 = O(\frac{d^{2/3}}{(KT)^{1/3}})$. For $\alpha = 1$ (i.e, sub-Exponential $\pi^\star$), the rate (in squared KSD) becomes $O(\frac{d^{3/4}}{(KT)^{1/4}})$. To the best of our knowledge, our convergence guarantee for sub-exponential $\pi^\star$ is the first of its kind.

**Comparison with Prior Works** Salim et al. [42] analyzes population-limit SVGD for subgaussian $\pi^\star$, obtaining $\mathsf{KSD}^2 = O(d^{3/2}/T)$ rate. We note that population-limit SVGD (which requires infinite particles) is not implementable whereas VP-SVGD is an implementable algorithm whose outputs are conditionally i.i.d samples from a distribution with guaranteed convergence to $\pi^\star$.

## 6.2   GB-SVGD

We now use VP-SVGD as the basis to analyze GB-SVGD. Assume $n > KT$. Then, with probability at least $1 - K^2T^2/n$ (for with-replacement sampling) and $1$ (for without-replacement sampling), the random batches $\mathcal{K}_t$ in GB-SVGD (Algorithm 2) are disjoint and contain distinct elements. When conditioned on this event $\mathcal{E}$, we note that the $n - KT$ particles that were not included in any random batch $\mathcal{K}_t$ evolve exactly like the $n$ real particles of VP-SVGD. With this insight, we show that, conditioned on $\mathcal{E}$, the outputs of VP-SVGD and GB-SVGD can be coupled such that the first $n - KT$ particles of both the algorithms are exactly equal. This allows us to derive the following squared KSD bound between the empirical measures of the outputs of VP-SVGD and GB-SVGD. The proof of this result is presented in Appendix D.

**Theorem 2** (**KSD Bounds for GB-SVGD**). *Let $n > KT$ and let $\mathbf{Y} = (\mathbf{y}^{(0)}, \ldots, \mathbf{y}^{(n-1)})$ and $\bar{\mathbf{Y}} = (\bar{\mathbf{y}}^{(0)}, \ldots, \bar{\mathbf{y}}^{(n-1)})$ denote the outputs of VP-SVGD and GB-SVGD respectively. Moreover, let $\hat{\mu}^{(n)} = \frac{1}{n}\sum_{i=0}^{n-1}\delta_{\mathbf{y}^{(i)}}$ and $\hat{\nu}^{(n)} = \frac{1}{n}\sum_{i=0}^{n-1}\delta_{\bar{\mathbf{y}}^{(i)}}$ denote their respective empirical measures. Under the assumptions and parameter settings of Theorem 1, there exists a coupling of $\mathbf{Y}$ and $\bar{\mathbf{Y}}$ such that the following holds:*

$$\mathbb{E}[\mathsf{KSD}^2_{\pi^\star}(\hat{\nu}^{(n)}\|\hat{\mu}^{(n)})] \leq \begin{cases} \frac{2K^2T^2\xi^2}{n^2} & \textit{(without replacement sampling)} \\ \frac{2K^2T^2\xi^2}{n^2}\left(1 - \frac{K^2T^2}{n}\right) + \frac{2K^2T^2\xi^2}{n} & \textit{(with replacement sampling)} \end{cases} \tag{6}$$

## 6.3   Convergence of the Empirical Measure to the Target

As a corollary of Theorem 1 and Theorem 2, we show that the empirical measure of the output of VP-SVGD and GB-SVGD rapidly converges to $\pi^\star$ in KSD. We refer to Appendix E for the full statement and proof.

**Corollary 1** (**VP-SVGD and GB-SVGD: Fast Finite Particle Rates**). *Let the assumptions and parameter settings of Theorem 1 be satisfied. Let $\hat{\mu}^{(n)}$ be the empirical measure of the $n$ particles output by VP-SVGD, run with run with $KT = d^{\frac{\alpha}{2+\alpha}}$, $R = \sqrt{d/L}$ and appropriately chosen $\gamma$. Then:*

$$\mathbb{E}[\mathsf{KSD}^2_{\pi^\star}(\hat{\mu}^{(n)}\|\pi^\star)] \leq O(\frac{d^{\frac{2}{2+\alpha}}}{n^{\frac{\alpha}{2+\alpha}}} + \frac{d^{2/\alpha}}{n})$$

*Similarly, let $\hat{\nu}^{(n)}$ be the empirical measure of the output of GB-SVGD without replacement, run with $KT = \sqrt{n}$, $R = \sqrt{d/L}$ and appropriately chosen $\gamma$. Then, the following holds:*

$$\mathbb{E}[\mathsf{KSD}_{\pi^\star}^2(\hat{\nu}^{(n)}||\pi^\star)] \leq O(\frac{d^{2/\alpha}}{n} + \frac{d^{\frac{1}{1+\alpha}}}{n^{\frac{1+2\alpha}{2(1+\alpha)}}} + \frac{d^{\frac{2+\alpha}{2(1+\alpha)}}}{n^{\frac{\alpha}{4(1+\alpha)}}})$$

**Comparison to Prior Work** For subgaussian $\pi^\star$ (i.e. $\alpha = 2$), VP-SVGD has a finite-particle rate of $\mathbb{E}[\mathsf{KSD}_{\pi^\star}(\hat{\mu}^{(n)}||\pi^\star)] = O((d/n)^{1/4} + (d/n)^{1/2})$ while that of GB-SVGD is $\mathbb{E}[\mathsf{KSD}_{\pi^\star}(\hat{\nu}^{(n)}||\pi^\star)] = O(d^{1/3}/n^{1/12} + (d/n)^{1/2})$. Both these rates are a *double exponential improvement* over the $\tilde{O}(\frac{\mathsf{poly}(d)}{\sqrt{\log\log n^{\Theta(1/d)}}})$ KSD rate obtained by Shi and Mackey [43] for SVGD with subgaussian $\pi^\star$.

For subexponential $\pi^\star$ (i.e. $\alpha = 1$) the KSD rate of VP-SVGD is $O(\frac{d^{1/3}}{n^{1/6}} + \frac{d}{n^{1/2}})$ while that of GB-SVGD is $O(\frac{d^{3/8}}{n^{1/16}} + \frac{d}{n^{1/2}})$. To our knowledge, both these results are the first of their kind.

**Oracle Complexity** As illustrated in Section E.3, for subgaussian $\pi^\star$, the oracle complexity of VP-SVGD to achieve $\epsilon$-convergence in KSD is $O(d^4/\epsilon^{12})$ and that of GB-SVGD is $O(d^6/\epsilon^{18})$. To our knowledge, these results are the *first known oracle complexities for this problem with polynomial dimension dependence*, and significantly improve upon the $O(\frac{\mathsf{poly}(d)}{\epsilon^2}e^{\Theta(de^{\mathsf{poly}(d)/\epsilon^2})})$ oracle complexity of SVGD as implied by Shi and Mackey [43]. For subexponential $\pi^\star$, the oracle complexity of VP-SVGD is $O(d^6/\epsilon^{16})$ and that of GB-SVGD is $O(d^9/\epsilon^{24})$.

## 7 Proof Sketch

We now present a sketch of our analysis. As shown in Section 4, the particles $(\mathbf{x}_t^{(l)})_{l \geq Kt}$ are i.i.d conditioned on the filtration $\mathcal{F}_t$, and the random measure $\mu_t|\mathcal{F}_t$ is the law of $(\mathbf{x}_t^{(Kt)})$ conditioned on $\mathbf{x}_0^{(0)}, \ldots, \mathbf{x}_0^{(Kt-1)}$. Moreover, from equation (5), we know that $\mu_t|\mathcal{F}_t$ is a stochastic approximation of population limit SVGD dynamics, i.e., $\mu_{t+1}|\mathcal{F}_{t+1} = (I - \gamma g_t)_{\#}\mu_t|\mathcal{F}_t$. Lemma 1 (similar to Salim et al. [42, Proposition 3.1] and Korba et al. [26, Proposition 5]) shows that under appropriate conditions, the KL divergence between $\mu_t|\mathcal{F}_t$ and $\pi^\star$ satisfies a (stochastic) descent lemma . Hence $\mu_t|\mathcal{F}_t$ admits a density and $\mathsf{KL}(\mu_t|\mathcal{F}_t||\pi^\star)$ is almost surely finite.

**Lemma 1** (Descent Lemma for $\mu_t|\mathcal{F}_t$)**.** *Let Assumptions 1, 3 and 5 be satisfied and let $\beta > 1$ be an arbitrary constant. On the event $\gamma\|g_t\|_{\mathcal{H}} \leq \frac{\beta-1}{\beta B}$, the following holds almost surely*

$$\mathsf{KL}(\mu_{t+1}|\mathcal{F}_{t+1}||\pi^\star) \leq \mathsf{KL}(\mu_t|\mathcal{F}_t||\pi^\star) - \gamma\langle h_{\mu_t|\mathcal{F}_t}, g_t\rangle_{\mathcal{H}} + \frac{\gamma^2(\beta^2+L)B}{2}\|g_t\|_{\mathcal{H}}^2$$

Lemma 1 is analogous to the noisy descent lemma which is used in the analysis of SGD for smooth functions. Notice that $\mathbb{E}[g_t|\mathcal{F}_t] = h_{\mu_t|\mathcal{F}_t}$ (when interpreted as a Gelfand-Pettis integral [46], as discussed in Appendix B and Appendix C) and hence in expectation, the KL divergence decreases in time. In order to apply Lemma 1, we establish an almost-sure bound on $\|g_t\|_{\mathcal{H}}$ below.

**Lemma 2.** *Let Assumptions 1, 2, 3, 4 and 5 hold. For $\gamma \leq 1/2A_1L$, the following holds almost surely,*

$$\|g_t\|_{\mathcal{H}} \leq \xi = \zeta_0 + \zeta_1(\gamma T)^{1/\alpha} + \zeta_2(\gamma^2 T)^{1/\alpha} + \zeta_3 R^{2/\alpha}$$

*where $\zeta_0, \zeta_1, \zeta_2$ and $\zeta_3$ which depend polynomially on $A_1, A_2, A_3, B, d_1, d_2, L$ for any fixed $\alpha$.*

Let $K = 1$ for clarity. To prove Lemma 2, we first note via smoothness of $F(\cdot)$ and Assumption 3 that $\|g_t\|_{\mathcal{H}} \leq C_0\|\mathbf{x}_t^{(t)}\| + C_1$, and then bound $\|\mathbf{x}_t^{(t)}\|$. Now, $g_s(\mathbf{x}) = k(\mathbf{x}, \mathbf{x}_s^{(s)})\nabla F(\mathbf{x}_s^{(s)}) - \nabla_2 k(\mathbf{x}, \mathbf{x}_s^{(s)})$. When $\|\mathbf{x}_s^{(s)} - \mathbf{x}\|$ is large, $\|g_s(\mathbf{x})\|$ is small due to decay assumptions on the kernel (Assumption 4) implying that the particle does not move much. When $\mathbf{x}_s^{(s)} \approx \mathbf{x}$, we have $g_s(\mathbf{x}) \approx k(\mathbf{x}, \mathbf{x}_s^{(s)})\nabla F(\mathbf{x}) - \nabla_2 k(\mathbf{x}, \mathbf{x}_s^{(s)})$ and $k(\mathbf{x}, \mathbf{x}_s^{(s)}) \geq 0$. This is approximately a gradient descent update on $F(\cdot)$ along with a bounded term $\nabla_2 k(\mathbf{x}, \mathbf{x}_s^{(s)})$. Thus, the value of $F(\mathbf{x}_t^{(l)})$ cannot grow too large after $T$ iterations. By Assumption 2, $F(\mathbf{x}_t^{(l)})$ being small implies that $\|\mathbf{x}_t^{(l)}\|$ is small.

Equipped with Lemma 2, we set the step-size $\gamma$ to ensure that the descent lemma (Lemma 1) always holds. The remainder of the proof involves unrolling through Lemma 1 by taking iterated expectations on both sides. To this end we control $\langle h_{\mu_t|\mathcal{F}_t}, g_t\rangle_{\mathcal{H}}$ and $\|g_t\|_{\mathcal{H}}^2$ in expectation, in Lemma 3.

**Lemma 3.** *Let Assumptions 1,2,3,4,5 hold and $\xi$ be as defined in Lemma 2. Then, for $\gamma \leq 1/2A_1L$,*

$$\mathbb{E}\left[\langle h_{\mu_t|\mathcal{F}_t}, g_t\rangle_{\mathcal{H}} |\mathcal{F}_t\right] = \|h_{\mu_t|\mathcal{F}_t}\|_{\mathcal{H}}^2 \quad and \quad \mathbb{E}[\|g_t\|_{\mathcal{H}}^2] \leq \xi^2/K + \|h_{\mu_t|\mathcal{F}_t}\|_{\mathcal{H}}^2$$

## 8 Experiments

We compare the performance of GB-SVGD and SVGD. We take $n = 100$ and use the Laplace kernel with $h = 1$ for both. We pick the stepsize $\gamma$ by a grid search for each algorithm. Additional details are presented in Appendix G. We observe that SVGD takes fewer iterations to converge, but the compute time for GB-SVGD is lower. This is similar to the typical behavior of stochastic optimization algorithms like SGD.

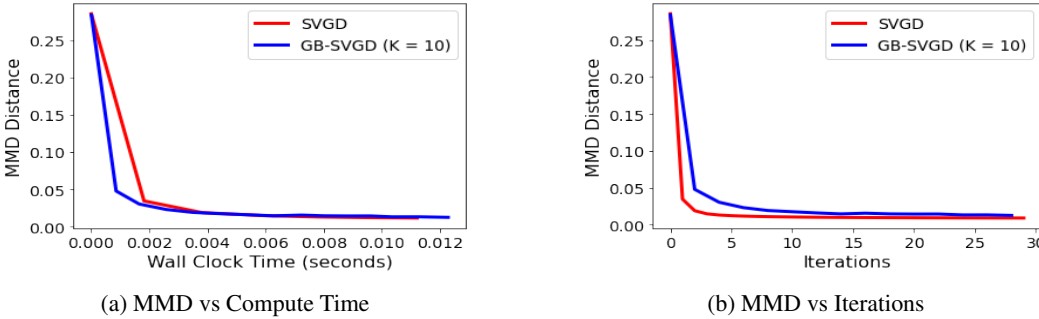

(a) MMD vs Compute Time          (b) MMD vs Iterations

Figure 1: Gaussian Experiment Comparing SVGD and GB-SVGD averaged over 10 experiments.

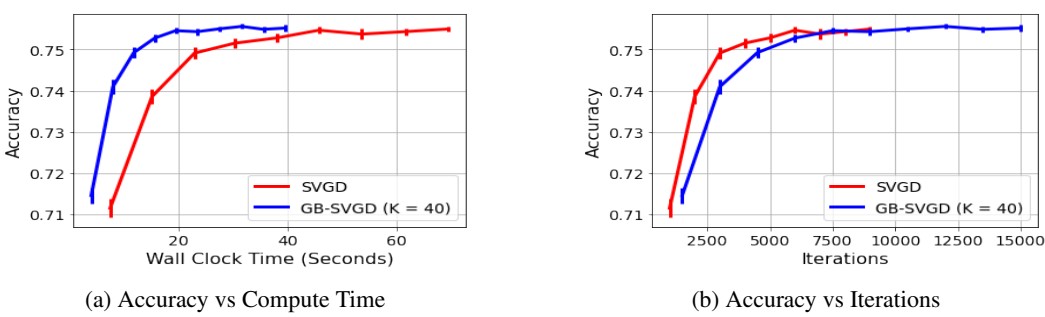

(a) Accuracy vs Compute Time          (b) Accuracy vs Iterations

Figure 2: Covertype Experiment, averaged over 50 runs. The error bars represent 95% CI.

**Sampling from Isotropic Gaussian (Figure 1):** As a sanity check, we set $\pi^\star = \mathcal{N}(0, \mathbf{I})$ with $d = 5$. We pick $K = 10$ for GB-SVGD. The metric of convergence is MMD with respect to the empirical measure of 1000 i.i.d. sampled Gaussians.

**Bayesian Logistic Regression (Figure 2)** We consider the Covertype dataset which contains $\sim 580,000$ data points with $d = 54$. We consider the same priors suggested in Gershman et al. [16] and implemented in Liu and Wang [31]. We take $K = 40$ for GB-SVGD. For both VP-SVGD and GB-SVGD, we use AdaGrad with momentum to set the step-sizes as per Liu and Wang [31]

## 9 Conclusion

We develop two computationally efficient variants of SVGD with provably fast convergence guarantees in the finite-particle regime, and present a wide range of improvements over prior work. A promising avenue of future work could be to establish convergence guarantees for SVGD with general non-logconcave targets, as was considered in recent works on LMC and SGLD [2, 12]. Other important avenues include establishing minimax lower bounds for SVGD and related particle-based variational inference algorithms. Beyond this, we also conjecture that the rates of GB-SVGD can be improved even in the regime $n \ll KT$. However, we believe this requires new analytic tools.

## Acknowledgements

We thank Jiaxin Shi, Lester Mackey and the anonymous reviewers for their helpful feedback. We are particularly grateful to Lester Mackey for providing insightful pointers on the properties of Kernel Stein Discrepancy, which greatly helped us in removing the curse of dimensionality from our finite-particle convergence guarantees.

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
