# Contents

# A    Additional Notation and Organization

We use $\Gamma$ to denote the Gamma function $\Gamma(x) = \int_0^\infty t^{x-1} e^{-t} dt$, and recall that for any $n \in \mathbb{N}$, $\Gamma(n) = (n-1)!$. For any Lebesgue measurable $A \subseteq \mathbb{R}^d$, we use $\mathrm{vol}(A)$ to denote it's Lebesgue Measure and $\mathrm{Uniform}(A)$ to denote the uniform distribution supported on $A$. We use $\mathcal{B}(R)$ to denote the ball of radius $R$ centered at the origin, and recall that $\mathrm{vol}(\mathcal{B}(R)) = \frac{\pi^{d/2}}{\Gamma(d/2+1)} R^d$. For ease of exposition, we assume $d \geq 2$. We further assume $\pi^\star(\mathbf{x}) = e^{-F(\mathbf{x})}$. We note that this can be easily ensured by absorbing the normalizing constant into $F(0)$, and does not affect the dynamics of SVGD, VP-SVGD or GB-SVGD (since they only use the gradient information of $F$). We highlight that both these assumptions are made purely for the sake of clarity and are very easily removable with negligible changes to our analysis.

In Appendix B, we discuss the technical lemmas used in our analysis, and present a short exposition to the Gelfand-Pettis integral in Appendix B.1, which we use to analyze VP-SVGD. We analyze VP-SVGD in Appendix C and GB-SVGD in Appendix D. Convergence guarantees for the empirical measure of VP-SVGD and GB-SVGD are presented in Appendix E. We give a brief review of the related work in Section F. Additional details regarding our experimental setup are stated in G.

# B    Preliminaries

The following lemma shows that setting the initial distribution $\mu_0 = \mathrm{Uniform}(\mathcal{B}(R))$ with $R = \sqrt{d/L}$ suffices to ensure $\mathrm{KL}\left(\mu_0 || \pi^\star\right) = O(d)$. The proof of this result is similar to that of Vempala and Wibisono [47, Lemma 1] with the Gaussian initialization replaced by $\mathrm{Uniform}(\mathcal{B}(R))$ initialization.

**Lemma 4** (**KL Upper Bound for Uniform Initialization**). *Let Assumption 1 be satisfied and let* $\mu_0 = \mathrm{Uniform}(\mathcal{B}(R))$ *with* $R = \sqrt{d/L}$. *Then, the following holds:*

$$\mathrm{KL}\left(\mu_0 || \pi^\star\right) \leq \frac{d}{2} \log(L/2\pi) + d + F(0) + 1/2 \leq O(d)$$

*Proof.* For any $\mathbf{x} \in \mathbb{R}^d$, the following holds by Assumption 1

$$
\begin{aligned}
F(\mathbf{x}) &\leq F(0) + \langle \nabla F(0), \mathbf{x} \rangle + \tfrac{L}{2} \|\mathbf{x}\|^2 \\
&\leq F(0) + \sqrt{L} \|\mathbf{x}\| + \tfrac{L}{2} \|\mathbf{x}\|^2 \\
&\leq F(0) + 1/2 + L \|\mathbf{x}\|^2
\end{aligned}
$$

where the second inequality uses $\|\nabla F(0)\| \leq \sqrt{L}$ and the Cauchy Schwarz inequality, and the last inequality uses the identity $ab \leq a^2 + b^2/4$. It follows that,

$$\mathbb{E}_{\mathbf{x} \sim \mu_0}[F(\mathbf{x})] \leq F(0) + 1/2 + LR^2$$

By a slight abuse of notation, let $\mu_0$ denote the density of $\mathrm{Uniform}(\mathcal{B}(R))$. Clearly. $\mu_0(\mathbf{x}) = \frac{1}{\mathrm{vol}(\mathcal{B}(R))} \mathbb{I}_{\mathbf{x} \in \mathcal{B}(R)}$. It follows that,

$$\int_{\mathbb{R}^d} \mu_0(\mathbf{x}) \ln(\mu_0(\mathbf{x})) d\mathbf{x} = \int_{\mathcal{B}(R)} \tfrac{1}{\mathrm{vol}(\mathcal{B}(R))} \log(1/\mathrm{vol}(\mathcal{B}(R))) d\mathbf{x} = -\log(\mathrm{vol}(\mathcal{B}(R)))$$

Now, $\mathrm{vol}(\mathcal{B}(R)) = \frac{\pi^{d/2}}{\Gamma(\frac{d}{2}+1)} R^d$. Furthermore, by Stirling's Approximation, $(x/e)^{x-1} \leq \Gamma(x) \leq (x/2)^{x-1}$. Hence,

$$\frac{d}{2} \log\left(\frac{2\pi R^2}{d/2+1}\right) \leq \log(\mathrm{vol}(\mathcal{B}(R))) \leq \frac{d}{2} \log\left(\frac{e\pi R^2}{d/2+1}\right)$$

Without loss of generality, assume $\pi^\star(\mathbf{x}) = e^{-F(\mathbf{x})}$ (this can be easily ensured by appropriately adjusting $F(0)$ upto constant factors). It follows that,

$$
\begin{aligned}
\mathrm{KL}\left(\mu_0 || \pi^\star\right) = \int_{\mathbb{R}^d} \mu_0(\mathbf{x}) \log(\tfrac{\mu_0(\mathbf{x})}{\pi^\star(\mathbf{x})}) d\mathbf{x} &= \int_{\mathbb{R}^d} \mu_0(\mathbf{x}) \ln(\mu_0(\mathbf{x})) d\mathbf{x} + \mathbb{E}_{\mathbf{x} \sim \mu_0}[F(\mathbf{x})] \\
&\leq -\log(\mathrm{vol}(\mathcal{B}(R))) + F(0) + 1/2 + LR^2 \\
&\leq \frac{d}{2} \log\left(\frac{d/2+1}{2\pi R^2}\right) + F(0) + 1/2 + LR^2
\end{aligned}
$$

Setting $R = \sqrt{d/L}$, we conclude that,

$$\mathsf{KL}\left(\mu_0\|\pi^\star\right) \le \frac{d}{2}\log(L/2\pi) + d + F(0) + 1/2 \le O(d)$$

$\square$

We now show that the growth condition on $F$, i.e. Assumption 2 is more general than specific concentration assumptions on $\pi^\star$ (e.g. subgaussianity, subexponentiality etc.). To this end, we define the notion of $\alpha$-tail decay as follows:

**Definition 2** ($\alpha$-Tail Decay). *A probability distribution $\nu$ on $\mathbb{R}^d$ is said to satisfy $\alpha$-tail decay for some $\alpha > 0$ if there exists some $C > 0$ such that $\mathbb{E}_{\mathbf{x}\sim\nu}\left[\exp\left(\|\frac{\mathbf{x}}{C}\|^\alpha\right)\right] < \infty$*

The $\alpha$-tail decay condition essentially implies that the tails of $\pi^\star$ decay as $\propto e^{-\|\mathbf{x}\|^\alpha}$. In particular, Vershynin [48, Proposition 2.5.2 and Proposition 2.7.1] shows that $\pi^\star$ satisfying the tail decay condition with $\alpha = 2$ is equivalent to $\pi^\star$ being subgaussian, whereas tail deay with $\alpha = 1$ is equivalent to $\pi^\star$ being subexponential.

In the following lemma, we establish that, under smoothness of $F$, the $\alpha$-tail decay condition is equivalent to the growth condition on $F$ with the same exponent $\alpha$. Consequently, Assumption 2 is much weaker than the standard isoperimetric and information transport assumptions generally used in the literature.

**Lemma 5** (Growth Condition and Tail Decay). *Let Assumption 2 be satisfied for some $\alpha > 0$. Then, $\pi^\star$ satisfies the $\alpha$-tail decay condition. Conversely, let Assumption 1 be satisfied and suppose $\pi^\star$ satisfies the $\alpha$-tail decay condition. Then, $F$ satisfies Assumption 2 with the same exponent $\alpha$.*

*Proof.* **Growth Condition Implies Tail Decay** Since Assumption 2 is satisfied, $F(\mathbf{x}) \ge d_1\|\mathbf{x}\|^\alpha - d_2$ for some $d_1, d_2, \alpha > 0$. Let $C = (2/d_1)^{1/\alpha}$. It follows that,

$$\begin{aligned}
\mathbb{E}_{\mathbf{x}\sim\pi^\star}\left[e^{\|\mathbf{x}/C\|^\alpha}\right] &= \int_{\mathbb{R}^d} e^{\frac{d_1}{2}\|\mathbf{x}\|^\alpha} \pi^\star(\mathbf{x})\mathsf{d}\mathbf{x} \\
&\le \int_{\mathbb{R}^d} e^{\frac{d_1}{2}\|\mathbf{x}\|^\alpha - d_1\|\mathbf{x}\|^\alpha + d_2}\mathsf{d}\mathbf{x} \\
&= e^{d_2}\int_{\mathbb{R}^d} e^{-\frac{d_1}{2}\|\mathbf{x}\|^\alpha}\mathsf{d}\mathbf{x} < \infty
\end{aligned}$$

From Definition 2, we conclude that $\pi^\star$ satisfies $\alpha$-tail decay.

**Smoothness and Tail Decay Imply the Growth Condition** Since $F$ is smooth, it suffices to consider $\alpha \in (0, 2]$. By Assumption 1, the following inequalities hold,

$$F(\mathbf{y}) - F(\mathbf{x}) \le \|\nabla F(\mathbf{x})\|\|\mathbf{y} - \mathbf{x}\| + \frac{L}{2}\|\mathbf{y} - \mathbf{x}\|^2 \le (L\|\mathbf{x}\| + \sqrt{L})\|\mathbf{y} - \mathbf{x}\| + \frac{L}{2}\|\mathbf{y} - \mathbf{x}\|^2 \quad (7)$$

Ww now prove this result by contradiction. Since $\pi^\star$ satisfies $\alpha$-tail decay, there exists a constant $C > 0$ such that $\mathbb{E}_{\mathbf{x}\sim\pi^\star}[e^{\|\mathbf{x}/C\|^\alpha}] < \infty$. Now, suppose $F$ does not satisfy the growth condition with exponent $\alpha$, i.e., *assume there does not exist* any $d_1, d_2 > 0$ such that $F(\mathbf{x}) \ge d_1\|\mathbf{x}\|^\alpha - d_2 \ \forall\ \mathbf{x} \in \mathbb{R}^d$. This implies that, $\liminf_{\|\mathbf{x}\|\to\infty} \frac{F(\mathbf{x})}{\|\mathbf{x}\|^\alpha} = 0$. Thus, without loss of generality, we can assume there exists a diverging sequence $a_n \in \mathbb{R}$ and a diverging sequence $\mathbf{x}_n \in \mathbb{R}^d$ that satisfy the following for every $n \in \mathbb{N}$:

$$\frac{F(\mathbf{x}_n)}{\|\mathbf{x}_n\|^\alpha} \le \frac{1}{a_n}, \quad \|\mathbf{x}_n\| \ge 2n, \quad \|\mathbf{x}_{n+1} - \mathbf{x}_n\| \ge 1 \quad (8)$$

where, without loss of generality, we assume $a_n, \|\mathbf{x}_n\| > 0$. Now, let $r_n = \frac{1}{\|\mathbf{x}_n\|^2}$ and $B_n \subseteq \mathbb{R}^d$ denote the ball of radius $r_n$ centered at $\mathbf{x}_n$. Since $r_n \le 1/4n^2$ and $\|\mathbf{x}_{n+1} - \mathbf{x}_n\| \ge 1$, $B_n$ is a family of disjoint subsets of $\mathbb{R}^d$. We shall now prove that there exists some diverging sequence $b_n \in \mathbb{R}$ such that $\frac{F(\mathbf{y})}{\|\mathbf{y}\|^\alpha} \le \frac{1}{b_n}$ for every $\mathbf{y} \in B_n$.

Consider any arbitrary $n \in \mathbb{N}$ and let $\mathbf{y} \in B_n$. Applying (7) to $\mathbf{y}$ and $\mathbf{x}_n$, we obtain,

$$
\begin{aligned}
\frac{F(\mathbf{y})}{\|\mathbf{x}_n\|^\alpha} &\leq \frac{F(\mathbf{x}_n)}{\|\mathbf{x}_n\|^\alpha} + \frac{L\|\mathbf{x}_n\|r_n}{\|\mathbf{x}_n\|^\alpha} + \frac{r_n\sqrt{L}}{\|\mathbf{x}_n\|^\alpha} + \frac{Lr_n^2}{2\|\mathbf{x}_n\|^\alpha} \\
&\leq \frac{1}{a_n} + \frac{L}{\|\mathbf{x}\|^{\alpha+1}} + \frac{\sqrt{L}}{\|\mathbf{x}_n\|^{\alpha+2}} + \frac{L}{2\|\mathbf{x}_n\|^{\alpha+4}}
\end{aligned} \tag{9}
$$

where we use (8) and $r_n = 1/\|\mathbf{x}_n\|^2$. Moreover, we note that

$$
\|\mathbf{y}\| \geq \|\mathbf{x}_n\| - \|\mathbf{y} - \mathbf{x}_n\| \geq \|\mathbf{x}_n\| - r_n = \|\mathbf{x}_n\| - \frac{1}{\|\mathbf{x}_n\|^2} \geq \frac{\|\mathbf{x}_n\|}{2} \tag{10}
$$

where we use the fact that $\|\mathbf{x}_n\| \geq 2n > 2^{1/3}$. It follows that,

$$
\begin{aligned}
\frac{F(\mathbf{y})}{\|\mathbf{y}\|^\alpha} &\leq \frac{2^\alpha F(\mathbf{y})}{\|\mathbf{x}_n\|^\alpha} \\
&\leq \frac{4}{a_n} + \frac{4L}{\|\mathbf{x}\|^{\alpha+1}} + \frac{4\sqrt{L}}{\|\mathbf{x}_n\|^{\alpha+2}} + \frac{2L}{\|\mathbf{x}_n\|^{\alpha+4}}
\end{aligned} \tag{11}
$$

where we use (9) and the fact that $\alpha \in (0, 2]$. We now define the sequence $b_n \in \mathbb{R}$ as follows:

$$
b_n = \left( \frac{4}{a_n} + \frac{4L}{\|\mathbf{x}\|^{\alpha+1}} + \frac{4\sqrt{L}}{\|\mathbf{x}_n\|^{\alpha+2}} + \frac{2L}{\|\mathbf{x}_n\|^{\alpha+4}} \right)^{-1}
$$

Since $\alpha > 0$, and $a_n, \|\mathbf{x}_n\| \to \infty$, it is clear that $b_n$ is a diverging sequence. Furthermore, from (11), we conclude that $\frac{F(\mathbf{y})}{\|\mathbf{y}\|^\alpha} \leq \frac{1}{b_n} \ \forall \ \mathbf{y} \in B_n$. Equipped with this construction, we note that

$$
\begin{aligned}
\mathbb{E}_{\mathbf{x} \sim \pi^\star} \left[ \exp\left( \frac{\|\mathbf{x}\|^\alpha}{C^\alpha} \right) \right] &= \int_{\mathbb{R}^d} \exp\left( \frac{\|\mathbf{y}\|^\alpha}{C^\alpha} \right) \exp(-F(\mathbf{y})) d\mathbf{y} \\
&\geq \sum_{n=1}^{\infty} \int_{B_n} \exp\left( \frac{\|\mathbf{y}\|^\alpha}{C^\alpha} \right) \exp(-F(\mathbf{y})) d\mathbf{y} \\
&\geq \sum_{n=1}^{\infty} \int_{B_n} \exp\left( \frac{\|\mathbf{y}\|^\alpha}{C^\alpha} - \frac{\|\mathbf{y}\|^\alpha}{b_n} \right) d\mathbf{y}
\end{aligned}
$$

where the second inequality use the fact that $B_n$ is a disjoint family of subsets of $\mathbb{R}^d$ and the third inequality uses the fact that $\frac{F(\mathbf{y})}{\|\mathbf{y}\|^\alpha} \leq \frac{1}{b_n} \ \forall \ \mathbf{y} \in B_n$. Since $b_n$ is a diverging sequence, there exists some $N_0 \in \mathbb{N}$ such that $b_n \geq 2C^\alpha \ \forall \ n \geq N_0$. It follows that,

$$
\begin{aligned}
\mathbb{E}_{\mathbf{x} \sim \pi^\star} \left[ \exp\left( \frac{\|\mathbf{x}\|^\alpha}{C^\alpha} \right) \right] &\geq \sum_{n=1}^{\infty} \int_{B_n} \exp\left( \frac{\|\mathbf{y}\|^\alpha}{C^\alpha} - \frac{\|\mathbf{y}\|^\alpha}{b_n} \right) d\mathbf{y} \\
&\geq \sum_{n=N_0}^{\infty} \int_{B_n} \exp\left( \frac{\|\mathbf{y}\|^\alpha}{2C^\alpha} \right) d\mathbf{y} \\
&= \sum_{n=N_0}^{\infty} \mathrm{vol}(B_n) \mathbb{E}_{\mathbf{y} \sim \mathsf{Uniform}(B_n)} \left[ \exp\left( \frac{\|\mathbf{y}\|^\alpha}{2C^\alpha} \right) \right]
\end{aligned}
$$

Consider the function $g : [0, \infty) \to [0, \infty)$ defined as $g(t) = e^{t^\alpha}$. We note that for $\alpha \geq 1$, $g$ is a convex function for every $t \geq 0$, and for $\alpha \in (0, 1)$, $g$ is convex for every $t \geq (1/\alpha - 1)^{1/\alpha}$. From (10), we note that $\|\mathbf{y}\| \geq \|\mathbf{x}\|/2 \geq n$ for every $\mathbf{y} \in B_n$. Hence, there exists an $N_1 \in \mathbb{N}$ such that $e^{t^\alpha}$

is a convex function for all $t \geq \|\mathbf{y}\|/2$, $\forall\, \mathbf{y} \in B_n$, $n \geq N_1$. Let $N = \max\{N_0, N_1\} + 1$. Then,

$$\mathbb{E}_{\mathbf{x}\sim\pi^\star}\left[\exp\left(\frac{\|\mathbf{x}\|^\alpha}{C^\alpha}\right)\right] \geq \sum_{n=N_0}^{\infty} \mathsf{vol}(B_n)\mathbb{E}_{\mathbf{y}\sim\mathsf{Uniform}(B_n)}\left[\exp\left(\frac{\|\mathbf{y}\|^\alpha}{2C^\alpha}\right)\right]$$

$$\geq \sum_{n=N}^{\infty} \mathsf{vol}(B_n)\exp\left(\tfrac{1}{2C^\alpha}\mathbb{E}_{\mathbf{y}\sim\mathsf{Uniform}(B_n)}[\|\mathbf{y}\|]^\alpha\right)$$

$$\geq \sum_{n=N}^{\infty} \mathsf{vol}(B_n)\exp\left(\tfrac{1}{2C^\alpha}\|\mathbb{E}_{\mathbf{y}\sim\mathsf{Uniform}(B_n)}[\mathbf{y}]\|^\alpha\right)$$

$$\geq \sum_{n=N}^{\infty} \mathsf{vol}(B_n)\exp\left(\tfrac{1}{2C^\alpha}\|\mathbf{x}_n\|^\alpha\right)$$

$$= \sum_{n=N}^{\infty} C_d(r_n)^d\exp\left(\tfrac{1}{2C^\alpha}\|\mathbf{x}_n\|^\alpha\right)$$

$$= \sum_{n=N}^{\infty} \frac{C_d\exp\left(\tfrac{1}{2C^\alpha}\|\mathbf{x}_n\|^\alpha\right)}{\|\mathbf{x}_n\|^{2d}}$$

where $C_d = \frac{\pi^{d/2}}{\Gamma(d/2+1)}$. Let $k$ be any positive integer such that $\alpha k \geq 2d + 1$. It follows that,

$$\frac{\exp\left(\tfrac{1}{2C^\alpha}\|\mathbf{x}_n\|^\alpha\right)}{\|\mathbf{x}_n\|^{2d}} \geq \frac{C_d}{2^k k! C^{\alpha k}}\|\mathbf{x}_n\|^{\alpha k - 2d} \geq \frac{C_d n}{2^{k-1} k! C^{\alpha k}}$$

Thus, we infer that,

$$\mathbb{E}_{\mathbf{x}\sim\pi^\star}\left[\exp\left(\frac{\|\mathbf{x}\|^\alpha}{C^\alpha}\right)\right] \geq \frac{C_d}{2^{k-1} k! C^{\alpha k}}\sum_{n=N_0}^{\infty} n = \infty$$

which is a contradiction. Thus, there exists some $d_1, d_2 > 0$ such that $F(\mathbf{x}) \geq d_1\|\mathbf{x}\|^\alpha - d_2$, i.e., $F$ satisfies the growth condition with exponent $\alpha$. $\qquad\square$

The following lemma establishes boundedness and contractivity properties of the function $h(\mathbf{x}, \mathbf{y}) = k(\mathbf{x}, \mathbf{y})\nabla F(\mathbf{y}) - \nabla_2 k(\cdot, \mathbf{y})$, that are vital for proving almost-sure bounds such as Lemma 2.

**Lemma 6** (Properties of $h$)**.** *Let Assumptions 1, 3 and 4 be satisfied. Then, the following holds,*

$$\|h(\cdot, \mathbf{y})\|_\mathcal{H} \leq BL\|\mathbf{y}\| + B\|\nabla F(0)\| + B$$

$$\|h(\mathbf{x}, \mathbf{y})\| \leq \frac{A_1 L}{2} + A_2 + k(\mathbf{x}, \mathbf{y})\|\nabla F(\mathbf{x})\|$$

$$-\langle\nabla F(\mathbf{x}), h(\mathbf{x}, \mathbf{y})\rangle \leq -\tfrac{1}{2}k(\mathbf{x}, \mathbf{y})\|\nabla F(\mathbf{x})\|^2 + L^2 A_1 + A_3$$

*Proof.* Recalling the definition of $h$ from Section 2, we observe that,

$$h(\cdot, \mathbf{y}) = k(\cdot, \mathbf{y})\nabla F(\mathbf{y}) - \nabla_2 k(\cdot, \mathbf{y})$$

Thus, by triangle inequality of $\|\cdot\|_\mathcal{H}$, Assumptions 1 and 3, we obtain

$$\|h(\cdot, \mathbf{y})\|_\mathcal{H} \leq \|\nabla F(\mathbf{y})\|\|k(\cdot, \mathbf{y})\|_{\mathcal{H}_0} + \|\nabla_2 k(\cdot, \mathbf{y})\|_\mathcal{H}$$
$$\leq BL\|\mathbf{y}\| + B\|\nabla F(0)\| + B$$

To prove the remaining inequalities, we first note that,

$$h(\mathbf{x}, \mathbf{y}) = k(\mathbf{x}, \mathbf{y})\nabla F(\mathbf{y}) - \nabla_2 k(\mathbf{x}, \mathbf{y})$$
$$= k(\mathbf{x}, \mathbf{y})\nabla F(\mathbf{x}) + k(\mathbf{x}, \mathbf{y})\left[\nabla F(\mathbf{y}) - \nabla F(\mathbf{x})\right] - \nabla_2 k(\mathbf{x}, \mathbf{y}) \qquad (12)$$

Using Assumptions 1 and 4, we note that,

$$\|h(\mathbf{x}, \mathbf{y})\| \leq k(\mathbf{x}, \mathbf{y})\|\nabla F(\mathbf{x})\| + \frac{LA_1\|\mathbf{x} - \mathbf{y}\|}{1 + \|\mathbf{x} - \mathbf{y}\|^2} + A_2$$

$$\leq \frac{A_1 L}{2} + A_2 + k(\mathbf{x}, \mathbf{y})\|\nabla F(\mathbf{x})\|$$

where the second inequality uses the fact $\frac{t}{1+t^2} \leq \sfrac{1}{2}$

To prove the last inequality, we infer the following from (12)

$$
\begin{aligned}
-\langle \nabla F(\mathbf{x}), h(\mathbf{x}, \mathbf{y}) \rangle &\leq -k(\mathbf{x}, \mathbf{y}) \|\nabla F(\mathbf{x})\|^2 + k(\mathbf{x}, \mathbf{y}) \|\nabla F(\mathbf{x}) - \nabla F(\mathbf{y})\| \|\nabla F(\mathbf{x})\| \\
&\quad + \|\nabla_2 k(\mathbf{x}, \mathbf{y})\| \|\nabla F(\mathbf{x})\| \\
&\leq -k(\mathbf{x}, \mathbf{y}) \|\nabla F(\mathbf{x})\|^2 + L\sqrt{k(\mathbf{x}, \mathbf{y})} \sqrt{\frac{A_1 \|\mathbf{x} - \mathbf{y}\|^2}{1 + \|\mathbf{x} - \mathbf{y}\|^2}} \|\nabla F(\mathbf{x})\| \\
&\quad + \sqrt{A_3 k(\mathbf{x}, \mathbf{y})} \|\nabla F(\mathbf{x})\| \\
&\leq -\frac{1}{2} k(\mathbf{x}, \mathbf{y}) \|\nabla F(\mathbf{x})\|^2 + L^2 A_1 + A_3
\end{aligned}
$$

where the second inequality uses Assumptions 1 and 4, and the last inequality uses the identity $ab \leq a^2 + \sfrac{b^2}{4}$ $\qquad\square$

To analyze the dynamics of VP-SVGD in the Wasserstein space, we use the following lemma presented in Salim et al. [42]

**Lemma 7** (Salim et al. [42], Proposition 3.1). *Let Assumptions 1 and 3 be satisfied. Consider any $\nu_0 \in \mathcal{P}_2(\mathbb{R}^d)$ with $\mathsf{KL}\,(\nu_0 \| \pi^\star) < \infty$, $f \in \mathcal{H}$ and let $\nu_1 = (I - \eta f)_\# \nu_0$ with $\eta \|f\|_{\mathcal{H}} \leq \frac{\beta - 1}{\beta B}$ for some $\beta > 1$. Then, the following holds,*

$$
\mathsf{KL}\,(\mu_1 \| \pi^\star) \leq \mathsf{KL}\,(\mu_0 \| \pi^\star) - \eta \langle h_{\mu_0}, f \rangle + \frac{\eta^2 (\beta^2 + L) B}{2} \|f\|_{\mathcal{H}}^2
$$

### B.1   Gelfand-Pettis Integrals for Reproducing Kernel Hilbert Spaces

The Gelfand-Pettis integral is a generalization of the Lebesgue integral to functions that take values in an arbitrary topological vector space. In this section, we describe the Gelfand-Pettis integral for an arbitrary Hilbert space $(V, \langle \cdot, \cdot \rangle_V)$ and refer the readers to Talagrand [46] for a more general treatment.

Let $(X, \Sigma, \lambda)$ be a measure space and $(V, \langle \cdot, \cdot \rangle_V)$ be a Hilbert Space. A function $g : X \to V$ is said to be Gelfand-Pettis integrable if there exists a vector $w_g \in V$ such that $\langle u, w_g \rangle_V = \int_X \langle u, g(x) \rangle_V \, \mathrm{d}\lambda(x) \ \forall\, u \in V$. The vector $w_g$ is called the Gelfand-Pettis integral of $g$

We now establish the following lemma for Gelfand-Pettis integrals with respect to the RKHS $\mathcal{H}$, which is a key component of our analysis of VP-SVGD.

**Lemma 8.** *Let $\mu$ be a probability measure on $\mathbb{R}^d$. Let $G : \mathbb{R}^d \times \mathbb{R}^d$ be a function such that for every $\mathbf{y} \in \mathbb{R}^d$, $G(., \mathbf{y}) \in \mathcal{H}$ with $\|G(., \mathbf{y})\|_{\mathcal{H}} \leq C$ holding $\mu$-almost surely. Let $G_\mu(\mathbf{x}) = \mathbb{E}_{\mathbf{y} \sim \mu}[G(\mathbf{x}, \mathbf{y})]$. Then, the map $\psi : \mathbb{R}^d \to \mathcal{H}$ defined as $\psi(\mathbf{y}) = G(\cdot, \mathbf{y})$ is Gelfand-Pettis integrable and $G_\mu$ is the Gelfand-Pettis integral of $\psi$ with respect to $\mu$, i.e. $G_\mu \in \mathcal{H}$ and for any $f \in \mathcal{H}$, $\mathbb{E}_{\mathbf{y} \sim \mu}[\langle f, G(., \mathbf{y}) \rangle_{\mathcal{H}}] = \langle f, G_\mu \rangle_{\mathcal{H}}$*

*Proof.* Let $\Phi : \mathcal{H} \to \mathbb{R}$ denote the map $\Phi(f) = \mathbb{E}_{\mathbf{y} \sim \mu}[\langle f, G(., \mathbf{y}) \rangle_{\mathcal{H}}] \ \forall\, f \in \mathcal{H}$. By linearity of expectations and inner products, we note that $\Phi$ is a linear functional on $\mathcal{H}$. Furthermore, since $\|G(., \mathbf{y})\|_{\mathcal{H}} \leq C$ holds $\mu$-almost surely, we note that for any $f \in \mathcal{H}$, $|\Phi(f)| \leq \mathbb{E}_{\mathbf{y} \sim \mu}[|\langle f, G(., \mathbf{y}) \rangle_{\mathcal{H}}|] \leq C \|f\|_{\mathcal{H}}$ by Jensen's inequality and Cauchy Schwarz inequality for $\mathcal{H}$. We conclude that $\Phi$ is a bounded linear functional of $\mathcal{H}$. Thus, by Reisz Representation Theorem [10], there exists $g \in \mathcal{H}$ such that for any $f \in \mathcal{H}$, the following holds

$$
\mathbb{E}_{\mathbf{y} \in \mu}[\langle f, G(., \mathbf{y}) \rangle_{\mathcal{H}}] = \langle f, g \rangle_{\mathcal{H}}
$$

Hence, we conclude that the map $\psi$ is Gelfand-Pettis integrable. We now use the reproducing property of $\mathcal{H}$ to show that $g = G_\mu$, i.e., $G_\mu$ is the Gelfand-Pettis integral of $\psi$. To this end, let $\mathbf{x} \in \mathbb{R}^d$ be arbitrary. Setting $f = k(\mathbf{x}, .)$ and using the fact that $g \in \mathcal{H}$, $G(., \mathbf{y}) \in \mathcal{H}$ for any $\mathbf{y} \in \mathbb{R}^d$,

$$
g(\mathbf{x}) = \mathbb{E}_{\mathbf{y} \in \mu}[G(\mathbf{x}, \mathbf{y})] = G_\mu(\mathbf{x})
$$

Hence, $g = G_\mu$, i.e., $\mathbb{E}_{\mathbf{y} \sim \mu}[\langle f, G(., \mathbf{y}) \rangle_{\mathcal{H}}] = \langle f, G_\mu \rangle_{\mathcal{H}}$ $\qquad\square$

# C   Analysis of VP-SVGD

In this section, we present our analysis of VP-SVGD. Throughout this section, we define the random function $g_t : \mathbb{R}^d \times \mathbb{R}^d$ as $g_t(\mathbf{x}) = \frac{1}{K} \sum_{l=0}^{K-1} h(\mathbf{x}, \mathbf{x}_t^{(Kt+l)})$ where $t \in \mathbb{N} \cup \{0\}$, $K$ is the batch-size of VP-SVGD, and $h : \mathbb{R}^d \times \mathbb{R}^d$ is as defined in Section 2, i.e., $h(\mathbf{x}, \mathbf{y}) = k(\mathbf{x}, \mathbf{y}) \nabla F(\mathbf{y}) - \nabla_2 k(\mathbf{x}, \mathbf{y})$.

After proving the key lemmas required for our analysis of VP-SVGD, we present the proof of Theorem 1 in Appendix C.4. We also present a high-probability version of Theorem 1 in Appendix C.5

## C.1   Population Level Dynamics : Proof of Lemma 1

*Proof.* We now derive the population-limit dynamics of VP-SVGD for arbitrary batch-size $K$, and subsequently prove the descent lemma (i.e. Lemma 1) for VP-SVGD. The arguments of this section are a straightforward generalization of that used in Section 4.

To this end, we recall from Section 4 that the countably infinite number of particles $\mathbf{x}_0^{(l)}$, $l \in \mathbb{N} \cup \{0\}$ are i.i.d samples from the measure $\mu_0$, which has a density w.r.t the Lebesgue measure. Thus, by the strong law of large numbers (Dudley [13, Theorem 11.4.1]), the empirical measure of $(\mathbf{x}_0^{(l)})_{l \geq 0}$ is almost surely equal to $\mu_0$. Furthermore, we recall the filtration $\mathcal{F}_t$ defined in Section 4 as $\mathcal{F}_t = \sigma(\mathbf{x}_0^{(l)} \mid l \leq Kt-1)$, $t \in \mathbb{N}$ with $\mathcal{F}_0$ being the trivial $\sigma$ algebra. We now consider the following dynamics in $\mathbb{R}^d$:

$$\mathbf{x}_{t+1}^{(s)} = \mathbf{x}_t^{(s)} - \frac{\gamma}{K} \sum_{l=0}^{K-1} h(\mathbf{x}_t^{(s)}, \mathbf{x}_t^{(tK+l)}), \quad s \in \mathbb{N} \cup \{0\} \tag{13}$$

We note that the above updates are the same as that of VP-SVGD for $s \in \{0, \ldots, KT+n-1\}$. Now, for each time-step $t$, we focus on the lower triangular evolution, i.e., the time evolution of the particles $(\mathbf{x}_t^{(l)})_{l \geq Kt}$. From (13), we infer that for any $t \in \mathbb{N}$ and $s \geq Kt$, $\mathbf{x}_t^{(s)}$ depends only on $(\mathbf{x}_0^{(l)})_{l \leq Kt-1}$ and $\mathbf{x}_0^{(s)}$. Hence, there exists a measurable function $H_t$ for every $t \in \mathbb{N}$ such that the following holds almost surely:

$$\mathbf{x}_t^{(s)} = H_t(\mathbf{x}_0^{(0)}, \ldots, \mathbf{x}_0^{(Kt-1)}, \mathbf{x}_0^{(s)}); \quad \forall \, s \geq Kt \tag{14}$$

Since $\mathbf{x}_0^{(0)}, \ldots, \mathbf{x}_0^{(Kt-1)}, \mathbf{x}_0^{(s)} \overset{i.i.d.}{\sim} \mu_0$, we conclude from (14) that $(\mathbf{x}_t^{(s)})_{s \geq Kt}$ are i.i.d when conditioned on $\mathbf{x}_0^{(0)}, \ldots, \mathbf{x}_0^{(Kt-1)}$. To this end, we define the random measure $\mu_t|\mathcal{F}_t$ as the law of $\mathbf{x}_t^{(Kt)}$ conditioned on $\mathcal{F}_t$, i.e. $\mu_t|\mathcal{F}_t$ is a probability kernel $\mu_t(\cdot; \mathbf{x}_0^{(0)}, \ldots, \mathbf{x}_0^{(Kt-1)})$ with $\mu_0|\mathcal{F}_0 := \mu_0$. By the strong law of large numbers, $\mu_t|\mathcal{F}_t$ is equal to the empirical measure of $(\mathbf{x}_t^{(l)})_{l \geq Kt}$ conditioned on $\mathcal{F}_t$. Furthermore, we infer from (13) that the particles satisfy the following:

$$\mathbf{x}_{t+1}^{(s)} = (I - \gamma g_t)(\mathbf{x}_t^{(s)}), \quad s \geq K(t+1)$$

Recall that $\mathbf{x}_{t+1}^{(s)}|\mathbf{x}_0^{(0)}, \ldots, \mathbf{x}_0^{(K(t+1)-1)} \sim \mu_{t+1}|\mathcal{F}_{t+1}$ for any $s \geq K(t+1)$. Furthermore, from Equation (14), we note that for $s \geq K(t+1)$, $\mathbf{x}_t^{(s)}$ depends only on $\mathbf{x}_0^{(0)}, \ldots, \mathbf{x}_0^{(Kt-1)}$ and $\mathbf{x}_0^{(s)}$, which implies that $\text{Law}(\mathbf{x}_t^{(s)}|\mathbf{x}_0^{(0)}, \ldots, \mathbf{x}_0^{(K(t+1)-1)}) = \text{Law}(\mathbf{x}_t^{(s)}|\mathbf{x}_0^{(0)}, \ldots, \mathbf{x}_0^{(Kt-1)}) = \mu_t|\mathcal{F}_t$. Finally, we note that $g_t$ is an $\mathcal{F}_{t+1}$-measurable random function. With these insights, we conclude that the population-level dynamics of the lower triangular evolution in $\mathcal{P}_2(\mathbb{R}^d)$ is almost surely described by the following update:

$$\mu_{t+1}|\mathcal{F}_{t+1} = (I - \gamma g_t)_{\#} \mu_t|\mathcal{F}_t \tag{15}$$

Setting $\gamma \|g_t\|_{\mathcal{H}} \leq \frac{\beta-1}{\beta B}$ for some arbitrary $\beta > 1$ and applying Lemma 7 to the population-level update (15), we conclude that the following holds almost surely:

$$\mathsf{KL}\left(\mu_{t+1}|\mathcal{F}_{t+1}\|\pi^\star\right) \leq \mathsf{KL}\left(\mu_t|\mathcal{F}_t\|\pi^\star\right) - \gamma \left\langle h_{\mu_t|\mathcal{F}_t}, g_t \right\rangle_{\mathcal{H}} + \frac{\gamma^2(\beta^2 + L)B}{2} \|g_t\|_{\mathcal{H}}^2$$

$\square$

## C.2 Iterate Bounds : Proof of Lemma 2

To establish almost sure bounds on $\|g_t\|_{\mathcal{H}}$, we prove the following result which is stronger than Lemma 2.

**Lemma 9** (Almost-Sure Iterate Bounds for VP-SVGD). *Let Assumptions 1, 2, 3, 4 and 5 be satisfied. Then, the following holds almost surely for any $s \in \mathbb{N} \cup \{0\}$ and $t \in (T+1)$ whenever $\gamma \leq 1/2A_1L$*

$$\|\mathbf{x}_t^{(s)}\| \leq \zeta_0 + \zeta_1(\gamma T)^{1/\alpha} + \zeta_2(\gamma^2 T)^{1/\alpha} + \zeta_3 R^{2/\alpha}$$

$$\|h(\cdot, \mathbf{x}_t^{(s)})\|_{\mathcal{H}} \leq \zeta_0 + \zeta_1(\gamma T)^{1/\alpha} + \zeta_2(\gamma^2 T)^{1/\alpha} + \zeta_3 R^{2/\alpha}$$

$$\|g_t\|_{\mathcal{H}} \leq \zeta_0 + \zeta_1(\gamma T)^{1/\alpha} + \zeta_2(\gamma^2 T)^{1/\alpha} + \zeta_3 R^{2/\alpha}$$

*where $\zeta_0, \ldots, \zeta_3$ are problem-dependent constants that depend polynomially on $A_1, A_2, A_3, B, d_1, d_2, L$ for any fixed $\alpha$.*

*Proof.* Let $c_t^{(s)} = \frac{1}{K} \sum_{l=0}^{K-1} k(\mathbf{x}_t^{(s)}, \mathbf{x}_t^{(Kt+l)})$. Note that by Assumption 4, $c_t^{(s)} \geq 0$ Since $\mathbf{x}_{t+1}^{(s)} = \mathbf{x}_t^{(s)} - \gamma g_t(\mathbf{x}_t^{(s)})$, it follows from the smoothness of $F$ that,

$$F(\mathbf{x}_{t+1}^{(s)}) - F(\mathbf{x}^{(s)}) \leq -\gamma \left\langle \nabla F(\mathbf{x}_t^{(s)}), g_t(\mathbf{x}_t^{(s)}) \right\rangle + \frac{\gamma^2 L}{2} \|g_t(\mathbf{x}_t^{(s)})\|^2 \tag{16}$$

By Lemma 6, we note that,

$$-\gamma \left\langle \nabla F(\mathbf{x}_t^{(s)}), g_t(\mathbf{x}_t^{(s)}) \right\rangle = -\frac{\gamma}{K} \sum_{l=0}^{K-1} \left\langle \nabla F(\mathbf{x}_t^{(s)}), h(\mathbf{x}_t^{(s)}, \mathbf{x}_t^{(tK+l)}) \right\rangle$$

$$\leq \frac{\gamma}{K} \sum_{l=0}^{L-1} \left[ -\frac{1}{2} k(\mathbf{x}_t^{(s)}, \mathbf{x}_t^{(tK+l)}) \|\nabla F(\mathbf{x}_t^{(s)})\|^2 + L^2 A_1 + A_3 \right]$$

$$\leq -\frac{\gamma c_t^{(s)}}{2} \|\nabla F(\mathbf{x}_t^{(s)})\|^2 + \gamma L^2 A_1 + \gamma A_3 \tag{17}$$

Moreover, by Jensen's Inequality and Lemma 6

$$\|g_t(\mathbf{x}_t^{(s)})\|^2 \leq \frac{1}{K} \sum_{l=0}^{K-1} \|h(\mathbf{x}_t^{(s)}, \mathbf{x}_t^{(Kt+l)})\|^2$$

$$\leq \frac{1}{K} \sum_{l=0}^{K-1} 2(A_1 L/2 + A_2)^2 + 2k(\mathbf{x}_t^{(s)}, \mathbf{x}_t^{(tK+l)})^2 \|F(\mathbf{x}_t^{(s)})\|^2$$

$$\leq \frac{1}{K} \sum_{l=0}^{K-1} 2(A_1 L/2 + A_2)^2 + 2A_1 k(\mathbf{x}_t^{(s)}, \mathbf{x}_t^{(tK+l)}) \|F(\mathbf{x}_t^{(s)})\|^2$$

$$\leq 2(A_1 L/2 + A_2)^2 + 2A_1 c_t^{(s)} \|F(\mathbf{x}_t^{(s)})\|^2 \tag{18}$$

Substituting (17) and (18) into (16), we obtain,

$$F(\mathbf{x}_{t+1}^{(s)}) - F(\mathbf{x}_t^{(s)}) \leq -\frac{\gamma c_t^{(s)}}{2} \|\nabla F(\mathbf{x}_t^{(s)})\|^2 + \gamma L^2 A_1 + \gamma A_3$$

$$+ \gamma^2 L(A_1 L/2 + A_2)^2 + \gamma^2 L A_1 c_t^{(s)} \|F(\mathbf{x}_t^{(s)})\|^2$$

$$\leq -\frac{\gamma c_t^{(s)}}{2}(1 - 2A_1 L\gamma) \|\nabla F(\mathbf{x}_t^{(s)})\|^2 + \gamma A_3 + \gamma L^2 A_1 + \gamma^2 L(A_1 L/2 + A_2)^2$$

$$\leq \gamma A_3 + \gamma L^2 A_1 + \gamma^2 L(A_1 L/2 + A_2)^2$$

where the last inequality uses the fact that $c_t^{(s)} \geq 0$ and $\gamma \leq 1/2A_1L$. Now, iterating through the above inequality, we obtain the following for any $t \in [T], s \in \mathbb{N} \cup \{0\}$

$$F(\mathbf{x}_t^{(s)}) \leq F(\mathbf{x}_0^{(s)}) + \gamma T L^2 A_1 + \gamma T A_3 + \gamma^2 T L(A_1 L/2 + A_2)^2 \tag{19}$$

Furthermore, by Assumption 1

$$F(\mathbf{x}_0^{(s)}) \le F(0) + \|\nabla F(0)\|\|\mathbf{x}_0^{(s)}\| + \frac{L}{2}\|\mathbf{x}_0^{(s)}\|^2$$

$$\le F(0) + 1/2 + L\|\mathbf{x}_0^{(s)}\|^2$$

Substituting the above inequality into (19), and using Assumption 2, we obtain the following for any $t \in [T]$, $s \in \mathbb{N} \cup \{0\}$

$$d_1\|\mathbf{x}_t^{(s)}\|^\alpha - d_2 \le F(\mathbf{x}_t^s) \le F(0) + 1/2 + L\|\mathbf{x}_0^{(s)}\|^2 + \gamma T L^2 A_1 + \gamma T A_3$$
$$+ \gamma^2 T L(A_1 L/2 + A_2)^2$$

Rearranging and applying Assumption 5, we obtain

$$\|\mathbf{x}_t^{(s)}\| \le d_1^{-1/\alpha}\left[F(0) + 1/2 + LR^2 + \gamma T L^2 A_1 + \gamma T A_3 + \gamma^2 T L(A_1 L + A_2)^2\right]^{1/\alpha}$$
$$\le \tilde{\zeta}_0 + \tilde{\zeta}_1(\gamma T)^{1/\alpha} + \tilde{\zeta}_2(\gamma^2 T)^{1/\alpha} + \tilde{\zeta}_3 R^{2/\alpha}$$

where $\tilde{\zeta}_0, \ldots, \tilde{\zeta}_3$ are constants that depend polynomially on $L, A_1, A_2, A_3, R$. We note that, since $0 < \alpha \le 2$, the above inequality also holds for $t = 0$.

Using the above inequality Lemma 6 and Assumption 1, we conclude that the following holds almost surely for any $t \in (T+1), s \in \mathbb{N} \cup \{0\}$

$$\|h(\cdot, \mathbf{x}_t^{(s)})\|_{\mathcal{H}} \le BL\|\mathbf{x}_t^{(s)}\| + B\sqrt{L} + B$$
$$\le \tilde{\eta}_0 + \tilde{\eta}_1(\gamma T)^{1/\alpha} + \tilde{\eta}_2(\gamma^2 T)^{1/\alpha} + \tilde{\eta}_3 R^{2/\alpha}$$

where $\tilde{\eta}_0, \ldots, \tilde{\eta}_3$ are constants that depend polynomially on $L, B, A_1, A_2, A_3, R$. Using the above inequality, we conclude that the following also holds for any $t \in (T+1)$.

$$\|g_t\|_{\mathcal{H}} \le \tilde{\eta}_0 + \tilde{\eta}_1(\gamma T)^{1/\alpha} + \tilde{\eta}_2(\gamma^2 T)^{1/\alpha} + \tilde{\eta}_3 R^{2/\alpha}$$

Taking $\zeta_i = \max\{\tilde{\zeta}_i, \tilde{\eta}_i\}$, the proof is complete. $\qquad\square$

## C.3 Controlling $g_t$ in Expectation : Proof of Lemma 3

*Proof.* Let $\xi = \zeta_0 + \zeta_1(\gamma T)^{1/\alpha} + \zeta_2(\gamma^2 T)^{1/\alpha} + \zeta_3 R^{2/\alpha}$ where $\zeta_0, \ldots, \zeta_3$ are as defined in Lemma 9. Recall that $g_t = \frac{1}{K}\sum_{l=0}^{K-1} h(., \mathbf{x}_t^{(Kt+l)})$. Since $\gamma \le 1/2A_1 L$, $\|h(\cdot, \mathbf{x}_t^{(Kt+l)})\|_{\mathcal{H}} \le \xi$ holds almost surely y Lemma 9.

Consider any $l \in (K)$. Conditioned on the filtration $\mathcal{F}_t$, $\text{Law}(\mathbf{x}_t^{(Kt+l)}|\mathcal{F}_t) = \mu_t|\mathcal{F}_t$. Moreover, for any $\mathbf{x} \in \mathbb{R}^d$, $\mathbb{E}_{\mathbf{x}_t^{(Kt+l)}}[h(\mathbf{x}, \mathbf{x}_t^{(Kt+l)})|\mathcal{F}_t] = h_{\mu_t|\mathcal{F}_t}(\mathbf{x})$. Thus, from Lemma 8, we conclude that $h_{\mu_t|\mathcal{F}_t}$ is the Gelfand-Pettis Integral of the map $\mathbf{x} \to h(\mathbf{x}, \mathbf{x}_t^{(Kt+l)})$ with respect to $\mu_t|\mathcal{F}_t$. Hence, the following holds

$$\mathbb{E}_{\mathbf{x}_t^{(Kt+l)}}\left[\left\langle h(\cdot, \mathbf{x}_t^{(Kt+l)}), f\right\rangle_{\mathcal{H}}\bigg|\mathcal{F}_t\right] = \left\langle h_{\mu_t|\mathcal{F}_t}, f\right\rangle_{\mathcal{H}}$$

In particular, setting $f = h_{\mu_t|\mathcal{F}_t}$ and using linearity of expectation, we conclude,

$$\mathbb{E}\left[\left\langle g_t, h_{\mu_t|\mathcal{F}_t}\right\rangle_{\mathcal{H}}|\mathcal{F}_t\right] = \frac{1}{K}\sum_{l=0}^{K-1}\mathbb{E}_{\mathbf{x}_t^{(Kt+l)}}\left[\left\langle h(\cdot, \mathbf{x}_t^{(Kt+l)}), h_{\mu_t|\mathcal{F}_t}\right\rangle_{\mathcal{H}}\bigg|\mathcal{F}_t\right]$$
$$= \|h_{\mu_t|\mathcal{F}_t}\|_{\mathcal{H}}^2$$

To control $\mathbb{E}[\|g_t\|_{\mathcal{H}}^2|\mathcal{F}_t]$, we note that,

$$\|g_t\|_{\mathcal{H}}^2 = \frac{1}{K^2}\sum_{l_1,l_2=0}^{K-1}\left\langle h(\cdot, \mathbf{x}_t^{(Kt+l_1)}), h(\cdot, \mathbf{x}_t^{(Kt+l_2)})\right\rangle_{\mathcal{H}}$$

$$= \frac{1}{K^2}\sum_{l=0}^{K-1}\|h(\cdot, \mathbf{x}_t^{(Kt+l)})\|_{\mathcal{H}}^2 + \sum_{0 \le l_1 \ne l_2 \le K-1}\left\langle h(\cdot, \mathbf{x}_t^{(Kt+l_1)}), h(\cdot, \mathbf{x}_t^{(Kt+l_2)})\right\rangle_{\mathcal{H}}$$

$$\le \frac{\xi^2}{K} + \sum_{0 \le l_1 \ne l_2 \le K-1}\left\langle h(\cdot, \mathbf{x}_t^{(Kt+l_1)}), h(\cdot, \mathbf{x}_t^{(Kt+l_2)})\right\rangle_{\mathcal{H}}$$

where the last inequality uses the fact that $\|h(\cdot, \mathbf{x}_t^{(Kt+l)})\|_{\mathcal{H}} \leq \xi$ almost surely as per Lemma 9.

To control the off-diagonal terms, let $i = Kt + l_1$ and $j = Kt + l_2$ for any arbitrary $l_1, l_2$ with $0 \leq l_1 \neq l_2 \leq K - 1$. Conditioned on $\mathcal{F}_t$, $\mathbf{x}_t^{(i)}$ and $\mathbf{x}_t^{(j)}$ are i.i.d samples from $\mu_t|\mathcal{F}_t$. Thus, by Lemma 8 and Fubini's Theorem,

$$\mathbb{E}_{\mathbf{x}_t^{(i)}, \mathbf{x}_t^{(j)}} \left[ \left\langle h(\cdot, \mathbf{x}_t^{(i)}), h(\cdot, \mathbf{x}_t^{(j)}) \right\rangle_{\mathcal{H}} | \mathcal{F}_t \right] = \mathbb{E}_{\mathbf{x}_t^{(i)}} \left[ \mathbb{E}_{\mathbf{x}_t^{(j)}} \left[ \left\langle h(\cdot, \mathbf{x}_t^{(i)}), h(\cdot, \mathbf{x}_t^{(j)}) \right\rangle_{\mathcal{H}} | \right] \right]$$

$$= \mathbb{E}_{\mathbf{x}_t^{(i)}} \left[ \left\langle h_{\mu_t|\mathcal{F}_t}, h(\cdot, \mathbf{x}_t^{(i)}) \right\rangle_{\mathcal{H}} | \mathcal{F}_t \right]$$

$$= \|h_{\mu_t|\mathcal{F}_t}\|_{\mathcal{H}}^2$$

Thus, we conclude that,

$$\mathbb{E}\left[ \|g_t\|_{\mathcal{H}}^2 | \mathcal{F}_t \right] \leq \|h_{\mu_t|\mathcal{F}_t}\|_{\mathcal{H}}^2 + \frac{\xi^2}{K}$$

$\square$

### C.4  Proof of Theorem 1

*Proof.* Let $\xi = \zeta_0 + \zeta_1 (\gamma T)^{1/\alpha} + \zeta_2 (\gamma^2 T)^{1/\alpha} + \zeta_3 R^{2/\alpha}$ where $\zeta_0, \dots, \zeta_3$ are as defined in Lemma 9. Since $\gamma \leq 1/2A_1 L$, $\|g_t\|_{\mathcal{H}} \leq \xi$ holds almost surely as per Lemma 9.

Since $\gamma \xi \leq 1/2B$, Lemma 1 ensures that the following holds almost surely

$$\mathsf{KL}\left(\mu_{t+1}|\mathcal{F}_{t+1}\|\pi^\star\right) \leq \mathsf{KL}\left(\mu_t|\mathcal{F}_t\|\pi^\star\right) - \gamma \left\langle h_{\mu_t|\mathcal{F}_t}, g_t \right\rangle_{\mathcal{H}} + \frac{\gamma^2(4+L)B}{2}\|g_t\|_{\mathcal{H}}^2$$

Taking conditional expectations w.r.t $\mathcal{F}_t$ on both sides and applying Lemma 3, we obtain,

$$\mathbb{E}\left[\mathsf{KL}\left(\mu_{t+1}|\mathcal{F}_{t+1}\|\pi^\star\right)|\mathcal{F}_t\right] \leq \mathsf{KL}\left(\mu_t|\mathcal{F}_t\|\pi^\star\right) - \gamma\left(1 - \frac{\gamma(4+L)B}{2}\right)\|h_{\mu_t|\mathcal{F}_t}\|_{\mathcal{H}}^2 + \frac{\gamma^2(4+L)B\xi^2}{2K}$$

$$\leq \mathsf{KL}\left(\mu_t|\mathcal{F}_t\|\pi^\star\right) - \frac{\gamma}{2}\|h_{\mu_t|\mathcal{F}_t}\|^2 + \frac{\gamma^2(4+L)B\xi^2}{2K}$$

$$= \mathsf{KL}\left(\mu_t|\mathcal{F}_t\|\pi^\star\right) - \frac{\gamma}{2}\mathsf{KSD}_{\pi^\star}(\mu_t|\mathcal{F}_t\|\pi^\star)^2 + \frac{\gamma^2(4+L)B\xi^2}{2K}$$

where the second inequality uses the fact that $\gamma \leq 1/(4+L)B$. Taking expectations on both sides and rearranging,

$$\frac{\gamma}{2}\mathbb{E}\left[\mathsf{KSD}_{\pi^\star}(\mu_t|\mathcal{F}_t\|\pi^\star)^2\right] \leq \mathbb{E}\left[\mathsf{KL}\left(\mu_t|\mathcal{F}_t\|\pi^\star\right) - \mathsf{KL}\left(\mu_{t+1}|\mathcal{F}_{t+1}\|\pi^\star\right)\right] + \frac{\gamma^2(4+L)B\xi^2}{2K}$$

Telescoping and averaging, we conclude,

$$\frac{1}{T}\sum_{t=0}^{T-1} \mathbb{E}\left[\mathsf{KSD}_{\pi^\star}(\mu_t|\mathcal{F}_t\|\pi^\star)^2\right] \leq \frac{2\mathsf{KL}\left(\mu_0|\mathcal{F}_0\|\pi^\star\right)}{\gamma T} + \frac{\gamma(4+L)B\xi^2}{K} \tag{20}$$

Now, recall from the proof of Lemma 1 in Section C.1 that for any $t \in [T]$ and $l \geq Kt$, $\mathbf{x}_t^{(l)}$ depends only on $\mathbf{x}_0^{(0)}, \dots, \mathbf{x}_0^{(Kt-1)}, \mathbf{x}_0^{(l)}$, i.e., there exists a deterministic measurable function $H_t$ such that $\mathbf{x}_t^{(l)} = H_t(\mathbf{x}_0^{(0)}, \dots, \mathbf{x}_0^{(Kt-1)}, \mathbf{x}_0^{(l)})$ holds almost surely. We note that the output $\mathbf{Y} = (\mathbf{y}^{(0)}, \dots, \mathbf{y}^{(n-1)})$ satisfies $\mathbf{y}^{(l)} = \mathbf{x}_S^{(KT+l)} \ \forall \ l \in (n)$, where $S \sim \mathsf{Uniform}((T))$ is sampled independently of everything else.

Thus, we infer that $\mathbf{y}^{(l)}$ depends only on $\mathbf{x}_0^{(0)}, \dots, \mathbf{x}_0^{(KT-1)}, S, \mathbf{x}_0^{(KT+l)}$, i.e., there exists a deterministic measurable function $G$ such that $\mathbf{y}^{(l)} = G(\mathbf{x}_0^{(0)}, \dots, \mathbf{x}_0^{(KT-1)}, S, \mathbf{x}_0^{(KT+l)})$ for every $l \in (n)$. Since $\mathbf{x}_0^{(KT)}, \dots, \mathbf{x}_0^{(KT+n-1)} \overset{i.i.d.}{\sim} \mu_0$, we infer that $\mathbf{y}^{(0)}, \dots, \mathbf{y}^{(n-1)}$ are i.i.d when conditioned on $\mathbf{x}_0^{(0)}, \dots, \mathbf{x}_0^{(KT-1)}, S$.

We now show that, when conditioned on $\mathbf{x}_0^{(0)}, \dots, \mathbf{x}_0^{(KT-1)}, S$, $\mathbf{y}^{(l)}$ is distributed as $\bar{\mu}$, where $\bar{\mu}$ is the probability kernel defined as $\bar{\mu}(\cdot; \mathbf{x}_0^{(0)} = x_0, \dots, \mathbf{x}_0^{(KT-1)} = x_{KT-1}, S = s) := \mu_s(\cdot, \mathbf{x}_0^{(0)} =$

$\mathbf{x}_0, \ldots, \mathbf{x}_0^{(Ks-1)} = \mathbf{x}_{Ks-1}$). For any arbitrary fixed $s \in (T)$, note that, under the event $S = s$, $\mathbf{y}^{(l)} = \mathbf{x}_s^{(KT+l)}$ for every $l \in (n)$. Thus, for any Borel measurable set $A \subseteq \mathbb{R}^d$, $\{\mathbf{y}^{(l)} \in A\} \cap \{S = s\} = \{\mathbf{x}_s^{(KT+l)} \in A\} \cap \{S = s\}$. For the sake of clarity, we denote the conditioning $\mathbf{x}_0^{(0)} = \mathbf{x}_0, \mathbf{x}_0^{(KT-1)} = \mathbf{x}_{KT-1}$ as $\mathcal{C}$, only in this proof. Since $S$ is independent of $\mathbf{x}_t^{(l)}$ for every $t \in (T+1), l \in (KT+n)$, we infer the following:

$$\mathbb{P}\left(\{\mathbf{y}^{(l)} \in A\}|\mathcal{C}, S = s\right) = \frac{\mathbb{P}\left(\{\mathbf{y}^{(l)} \in A\} \cap \{S = s\}|\mathcal{C}\right)}{\mathbb{P}(S = s)}$$

$$= T\mathbb{P}\left(\{\mathbf{x}_s^{(KT+l)} \in A\} \cap \{S = s\}|\mathcal{C}\right)$$

$$= T\mathbb{P}(\{S = s\})\mathbb{P}\left(\{\mathbf{x}^{(KT+l)} \in A\}|\mathcal{C}\right)$$

$$= \mathbb{P}\left(\{\mathbf{x}_s^{(KT+l)} \in A\}|\mathcal{C}\right)$$

As discussed above, $\mathbf{x}_s^{(KT+l)}$ depends only on $\mathbf{x}_0^{(0)}, \mathbf{x}_0^{(Ks-1)}, \mathbf{x}_0^{(KT+l)}$. It follows that $\mathbb{P}\left(\{\mathbf{x}_s^{(KT+l)} \in A\}|\mathcal{C}\right) = \mu_s(A; \mathbf{x}_0^{(0)} = x_0, \ldots, \mathbf{x}_0^{(Ks-1)} = x_{Ks-1})$ and,

$$\mathbb{P}\left(\{\mathbf{y}^{(l)} \in A\}|\mathcal{C}, S = s\right) = \mu_s(A; \mathbf{x}_0^{(0)} = x_0, \ldots, \mathbf{x}_0^{(Ks-1)} = x_{Ks-1})$$

$$= \bar{\mu}(A; \mathbf{x}_0^{(0)} = x_0, \ldots, \mathbf{x}_0^{(KT-1)} = x_{Kt-1}, S = s)$$

Thus, $\mathbf{y}^{(0)}, \ldots, \mathbf{y}^{(n-1)}$ are i.i.d samples from $\bar{\mu}$ when conditioned on $\mathbf{x}_0^{(0)}, \ldots \mathbf{x}_0^{(KT-1)}, S$.

We now obtain an upper bound on the expected squared KSD between $\bar{\mu}$ and $\pi^\star$. We recall from the proof of Lemma 1 in Section C.1 that, for any $t \in (T+1)$, conditioned on $\mathbf{x}_0^{(0)}, \ldots, \mathbf{x}_0^{(Kt-1)}$, $(\mathbf{x}_t^{(l)})_{l \geq t}$ are i.i.d samples from $\mu_t|\mathcal{F}_t$ where $\mu_t|\mathcal{F}_t := \mu_t(\cdot; \mathbf{x}_0^{(0)}, \mathbf{x}_0^{(Kt-1)})$. Hence, from (20), we conclude that,

$$\mathbb{E}[\mathsf{KSD}_{\pi^\star}(\bar{\mu}(\cdot; (\mathbf{x}_0^{(l)})_{l \in (KT)}, S)||\pi^\star)^2] = \frac{1}{T}\sum_{t=0}^{T-1}\mathbb{E}\left[\mathbb{E}\left[\mathsf{KSD}_{\pi^\star}(\bar{\mu}(\cdot; (\mathbf{x}_0^{(l)})_{l \in (KT)}, S = t)||\pi^\star)^2|(\mathbf{x}_0^{(l)})_{l \in (KT)}\right]\right]$$

$$= \frac{1}{T}\sum_{t=0}^{T-1}\mathbb{E}\left[\mathsf{KSD}_{\pi^\star}(\mu_t(\cdot; \mathbf{x}_0^{(0)}, \cdot, \mathbf{x}_0^{(Kt-1)}))||\pi^\star)^2\right]$$

$$= \frac{1}{T}\sum_{t=0}^{T-1}\mathbb{E}\left[\mathsf{KSD}_{\pi^\star}(\mu_t|\mathcal{F}_t||\pi^\star)^2\right]$$

$$\leq \frac{2\mathsf{KL}(\mu_0|\mathcal{F}_0||\pi^\star)}{\gamma T} + \frac{\gamma(4+L)B\xi^2}{K}$$

where we use the fact that $S \sim \mathsf{Uniform}((T))$ is sampled independent of everything else. $\square$

### C.5 VP-SVGD: High-Probability Guarantees

**Theorem 3** (**VP-SVGD: High-Probability Rates**)**.** *Let the assumptions and parameter settings of Theorem 1 apply and let $\delta \in (0, 1)$. Then, the following holds with probability at least $1 - \delta$:*

$$\frac{1}{T}\sum_{t=0}^{T-1}\mathsf{KSD}_{\pi^\star}(\mu_t|\mathcal{F}_t||\pi^\star)^2 \leq \frac{4\mathsf{KL}(\mu_0|\mathcal{F}_0||\pi^\star)}{\gamma T} + \frac{2\gamma(4+L)B\xi^2}{K}$$

$$+ \frac{32\xi^2\log(2/\delta)}{KT} + 12\gamma(4+L)B\xi^2\sqrt{\frac{\log(2/\delta)}{T}}$$

*Let $\bar{\mu}(\cdot; \mathbf{x}_0^{(0)}, \ldots, \mathbf{x}_0^{(KT-1)}, S)$ be the probability kernel defined in the statement of Theorem 1. Then, conditioned on $\mathbf{x}_0^{(0)}, \ldots, \mathbf{x}_0^{(KT-1)}, S$, the $n$ particles output by VP-SVGD are i.i.d samples from*

$\bar{\mu}(\cdot; \mathbf{x}_0^{(0)}, \ldots, \mathbf{x}_0^{(KT-1)}, S)$. *Furthermore, with probability at least* $1 - \delta$

$$\mathbb{E}_S[\mathsf{KSD}_{\pi^\star}(\bar{\mu}(\cdot; \mathbf{x}_0^{(0)}, \ldots, \mathbf{x}_0^{(KT-1)}, S)||\pi^\star)^2] \leq \frac{4\mathsf{KL}\,(\mu_0|\mathcal{F}_0||\pi^\star)}{\gamma T} + \frac{2\gamma(4+L)B\xi^2}{K}$$

$$+ \frac{32\xi^2 \log(2/\delta)}{KT} + 12\gamma(4+L)B\xi^2\sqrt{\frac{\log(2/\delta)}{T}}$$

*where* $\mathbb{E}_S$ *denotes that the expectation is being taken only with respect to* $S \sim \mathsf{Uniform}((T))$

*Proof.* Following the same steps as Theorem 1, we note that the following holds almost surely.

$$\mathsf{KL}\,(\mu_{t+1}|\mathcal{F}_{t+1}||\pi^\star) \leq \mathsf{KL}\,(\mu_t|\mathcal{F}_t||\pi^\star) - \gamma \left\langle h_{\mu_t|\mathcal{F}_t}, g_t \right\rangle_{\mathcal{H}} + \frac{\gamma^2(4+L)B}{2}\|g_t\|_{\mathcal{H}}^2$$

$$\leq \mathsf{KL}\,(\mu_t|\mathcal{F}_t||\pi^\star) - \frac{\gamma}{2}\|h_{\mu_t|\mathcal{F}_t}\|_{\mathcal{H}}^2 + \gamma \left\langle h_{\mu_t|\mathcal{F}_t}, h_{\mu_t|\mathcal{F}_t} - g_t \right\rangle_{\mathcal{H}}$$

$$+ \frac{\gamma^2(4+L)B\xi^2}{2K} + \frac{\gamma^2(4+L)B}{2}\left[\|g_t\|_{\mathcal{H}}^2 - \|h_{\mu_t|\mathcal{F}_t}\|_{\mathcal{H}}^2 - \frac{\xi^2}{K}\right] \qquad (21)$$

where the last inequality uses the fact that $\gamma \leq 1/(4+L)B$. We now define $\Delta_t^{(l)}$, $\Delta_t$ and $r_t$ for $l \in (K)$, $t \in (T)$ as follows:

$$\Delta_t^{(l)} = \left\langle h_{\mu_t|\mathcal{F}_t}, h_{\mu_t|\mathcal{F}_t} - h(\cdot, \mathbf{x}_t^{(Kt+l)}) \right\rangle_{\mathcal{H}}$$

$$\Delta_t = \frac{1}{K}\sum_{l=0}^{K-1}\Delta_t^{(l)} = \left\langle h_{\mu_t|\mathcal{F}_t}, h_{\mu_t|\mathcal{F}_t} - g_t \right\rangle_{\mathcal{H}}$$

$$r_t = \|g_t\|_{\mathcal{H}}^2 - \|h_{\mu_t|\mathcal{F}_t}\|_{\mathcal{H}}^2 - \frac{\xi^2}{K}$$

Substituting the above into (21), we obtain the following:

$$\mathsf{KL}\,(\mu_{t+1}|\mathcal{F}_{t+1}||\pi^\star) \leq \mathsf{KL}\,(\mu_t|\mathcal{F}_t||\pi^\star) - \frac{\gamma}{2}\|h_{\mu_t|\mathcal{F}_t}\|_{\mathcal{H}}^2 + \gamma\Delta_t + \frac{\gamma^2(4+L)B\xi^2}{2K} + \frac{\gamma^2(4+L)Br_t}{2}$$

Telescoping and averaging both sides, and using $\|h_{\mu_t|\mathcal{F}_t}\|_{\mathcal{H}}^2 = \mathsf{KSD}_{\pi^\star}(\mu_t|\mathcal{F}_t||\pi^\star)^2$, we obtain the following:

$$\frac{1}{T}\sum_{t=0}^{T-1}\mathsf{KSD}_{\pi^\star}(\mu_t|\mathcal{F}_t||\pi^\star)^2 \leq \frac{4\mathsf{KL}\,(\mu_0|\mathcal{F}_0||\pi^\star)}{\gamma T} + \frac{2\gamma(4+L)B\xi^2}{K}$$

$$+ \frac{4}{T}\sum_{t=0}^{T-1}\left(\Delta_t - \frac{\|h_{\mu_t|\mathcal{F}_t}\|_{\mathcal{H}}^2}{4}\right) + \frac{2\gamma(4+L)B}{T}\sum_{t=0}^{T-1}r_t \qquad (22)$$

We note that the first two terms are the same as that of the in-expectation guarantee for VP-SVGD in Theorem 1. The third and fourth term are random quantities that vanish in expectation. The remainder of our analysis upper bounds them with high probability.

We begin by deriving a high probability upper bound for the fourth term in (22). To this end, we note that, since $\gamma \leq 1/2A_1L$, $\|h(\cdot, \mathbf{x}_t^{(Kt+l)})\|_{\mathcal{H}} \leq \xi$ for any $t \in (T)$, $l \in (K)$ as per Lemma 9. Furthermore, since $\mathbb{E}[h(\cdot, \mathbf{x}_t^{(Kt+l)}|\mathcal{F}_t] = h_{\mu_t|\mathcal{F}_t}$ (both pointwise and in the sense of the Gelfand-Pettis integral, see proof of Lemma 3 in Appendix C.3), it follows by Jensen's inequality that $\|h_{\mu_t|\mathcal{F}_t}\|_{\mathcal{H}} \leq \xi$. This further implies that $|r_t| \leq 3\xi^2$. Moreover, $r_t$ is $\mathcal{F}_{t+1}$ measurable (as $g_t$ is an $\mathcal{F}_{t+1}$ measurable random function) with $\mathbb{E}[r_t|\mathcal{F}_t] \leq 0$ (as per Lemma 3)

Thus, $S_t = \sum_{s=0}^{t-1} r_t$ is an $\mathcal{F}$-adapted supermartingale difference sequence with bounded increments. Thus, by the Hoeffding-Azuma inequality, we conclude that the following holds with probability at least $1 - \delta/2$

$$\frac{1}{T}\sum_{t=0}^{T-1} r_t \leq 6\xi^2\sqrt{\frac{\log(2/\delta)}{T}} \qquad (23)$$

We now proceed to control the third term in (22). Recall from the proof of Theorem 1 in Appendix C.4, that, for any fixed $t \in (T)$, $(\mathbf{x}_t^{(l)})_{l \in (KT)}$ are i.i.d when conditioned on $\mathcal{F}_t$. As discussed above, $\mathbb{E}[h(\cdot, \mathbf{x}_t^{(Kt+l)})] = h_{\mu_t|\mathcal{F}_t}$ in the sense of the Gelfand-Pettis integral, implying $\mathbb{E}[\Delta_t^{(l)}] = 0$. Moreover, $\|\Delta^{(l)}\|_t \le 2\xi\|h_{\mu_t|\mathcal{F}_t}\|$. Thus, when conditioned on $\mathcal{F}_t$, $\Delta_t^{(l)}$ are independent zero-mean bounded random variables. Hence, we conclude the following by Hoeffding's Lemma

$$\mathbb{E}\left[e^{\theta\Delta_t}|\mathcal{F}_t\right] \le \prod_{l=0}^{K-1} \mathbb{E}[e^{\frac{\theta\Delta_t^{(l)}}{K}}|\mathcal{F}_t] \le e^{\frac{2\theta^2\xi^2}{K}\|h_{\mu_t|\mathcal{F}_t}\|_{\mathcal{H}}^2}, \quad \forall\, \theta \in \mathbb{R} \tag{24}$$

We now define the sequence $M_t$ as follows, where $\lambda = K/8\xi^2$

$$M_t = \exp\left(\sum_{s=0}^{t-1} \lambda\Delta_s - \tfrac{\lambda}{4}\|h_{\mu_s|\mathcal{F}_s}\|_{\mathcal{H}}^2\right)$$

Since $g_t$ is $\mathcal{F}_{t+1}$ measurable, so is $\Delta_t$, which implies $M_t$ is $\mathcal{F}_{t+1}$ measurable. Furthermore,

$$\mathbb{E}[M_t|\mathcal{F}_t] = M_{t-1}e^{-\frac{\lambda}{4}\|h_{\mu_t|\mathcal{F}_t}\|_{\mathcal{H}}^2}\mathbb{E}[e^{\lambda\Delta_t}|\mathcal{F}_t]$$

$$\le M_{t-1}e^{(-\frac{\lambda}{4}+\frac{2\lambda^2\xi^2}{K})\|h_{\mu_t|\mathcal{F}_t}\|_{\mathcal{H}}^2} \le M_{t-1}$$

Thus, $M_t$ is an $\mathcal{F}$-adapted supermartingale sequence. Following the same steps, we conclude $E[M_1] \le 1$, which implies $\mathbb{E}[M_T] \le \mathbb{E}[M_1] \le 1$. Thus, from Markov's Inequality

$$\mathbb{P}\left[\sum_{t=0}^{T-1} \Delta_t - \tfrac{1}{4}\|h_{\mu_t|\mathcal{F}_t}\|_{\mathcal{H}}^2 > x\right] \le e^{-\lambda x}\mathbb{E}[M_T] \le e^{-\lambda x}$$

Hence, the following holds with probability at least $1 - \delta/2$.

$$\sum_{t=0}^{T-1} \Delta_t - \tfrac{1}{4}\|h_{\mu_t|\mathcal{F}_t}\|_{\mathcal{H}}^2 \le \frac{8\xi^2}{K}\log(2/\delta) \tag{25}$$

Substituting (24) and (25) into (20) and taking a union bound, we conclude that the following holds with probability at least $1 - \delta$:

$$\frac{1}{T}\sum_{t=0}^{T-1} \mathsf{KSD}_{\pi^\star}(\mu_t|\mathcal{F}_t\|\pi^\star)^2 \le \frac{4\mathsf{KL}\,(\mu_0|\mathcal{F}_0\|\pi^\star)}{\gamma T} + \frac{2\gamma(4+L)B\xi^2}{K}$$

$$+ \frac{32\xi^2\log(2/\delta)}{KT} + 12\gamma(4+L)B\xi^2\sqrt{\frac{\log(2/\delta)}{T}} \tag{26}$$

Recall from the proof of Theorem 1 in Appendix C.4 that the outputs $(\mathbf{y}^{(l)})_{l\in(n)}$ of VP-SVGD are i.i.d samples from the random measure $\bar{\mu}(\cdot; \mathbf{x}_0^{(0)}, \ldots, \mathbf{x}_{KT-1}^{(0)}, S)$ when conditioned on $\mathbf{x}_0^{(0)}, \ldots, \mathbf{x}_{KT-1}^{(0)}, S$. Furthermore, when conditioned on $S = t$, $\bar{\mu}(\cdot; \mathbf{x}_0^{(0)}, \ldots, \mathbf{x}_{KT-1}^{(0)}, S = t) = \mu_t|\mathcal{F}_t$. Thus, from (26), we conclude that, upon taking an expectation over $S \sim \mathsf{Uniform}((T))$ while conditioning on the virtual particles $\mathbf{x}_0^{(0)}, \ldots, \mathbf{x}_0^{(KT-1)}$, the following holds with probability at least $1 - \delta$:

$$\mathbb{E}_S[\mathsf{KSD}_{\pi^\star}(\bar{\mu}(\cdot; \mathbf{x}_0^{(0)}, \ldots, \mathbf{x}_0^{(KT-1)}, S)\|\pi^\star)^2] \le \frac{1}{T}\sum_{t=0}^{T-1} \mathsf{KSD}_{\pi^\star}(\mu_t|\mathcal{F}_t\|\pi^\star)^2$$

$$\le \frac{4\mathsf{KL}\,(\mu_0|\mathcal{F}_0\|\pi^\star)}{\gamma T} + \frac{2\gamma(4+L)B\xi^2}{K}$$

$$+ \frac{32\xi^2\log(2/\delta)}{KT} + 12\gamma(4+L)B\xi^2\sqrt{\frac{\log(2/\delta)}{T}}$$

$\square$

# D   Analysis of GB-SVGD

In this section, we present our analysis of GB-SVGD. For any $t \in (T)$, we use $\tilde{g}_t$ to denote the random function $\tilde{g}_t(\mathbf{x}) = \frac{1}{K} \sum_{r \in \mathcal{K}_t} h(\mathbf{x}, \mathbf{x}_t^{(r)})$ where $\mathcal{K}_t$ is the random batch of size $K$ sampled at time-step $t$ of GB-SVGD.

In order to prove Theorem 2, we first establish an almost-sure iterate bound for GB-SVGD which is similar to that of Lemma 9 for VP-SVGD.

**Lemma 10** (Almost-Sure Iterate Bounds). *Let Assumptions 1, 2, 3, 4 and 5 be satisfied. Then, the following holds almost surely for any $s \in \mathbb{N} \cup \{0\}$ and $t \in (T+1)$ whenever $\gamma \leq 1/2A_1L$*

$$\|\mathbf{x}_t^{(s)}\| \leq \zeta_0 + \zeta_1(\gamma T)^{1/\alpha} + \zeta_2(\gamma^2 T)^{1/\alpha} + \zeta_3 R^{2/\alpha}$$

$$\|h(\cdot, \mathbf{x}_t^{(s)})\|_{\mathcal{H}} \leq \zeta_0 + \zeta_1(\gamma T)^{1/\alpha} + \zeta_2(\gamma^2 T)^{1/\alpha} + \zeta_3 R^{2/\alpha}$$

$$\|\tilde{g}_t\|_{\mathcal{H}} \leq \zeta_0 + \zeta_1(\gamma T)^{1/\alpha} + \zeta_2(\gamma^2 T)^{1/\alpha} + \zeta_3 R^{2/\alpha}$$

*where $\zeta_0, \dots, \zeta_3$ are problem-dependent constants that depend polynomially on $A_1, A_2, A_3, B, d_1, d_2, L$ for any fixed $\alpha$.*

*Proof.* The proof of this Lemma is identical to that of Lemma 9. To this end, let $c_t^{(s)} = \frac{1}{K} \sum_{r \in \mathcal{K}_t} k(\mathbf{x}_t^{(s)}, \mathbf{x}_t^{(r)})$. Note that by Assumption 4, $c_t^{(s)} \geq 0$ Since $\mathbf{x}_{t+1}^{(s)} = \mathbf{x}_t^{(s)} - \gamma \tilde{g}_t(\mathbf{x}_t^{(s)})$, it follows from the smoothness of $F$ that,

$$F(\mathbf{x}_{t+1}^{(s)}) - F(\mathbf{x}_t^{(s)}) \leq -\gamma \left\langle \nabla F(\mathbf{x}_t^{(s)}), \tilde{g}_t(\mathbf{x}_t^{(s)}) \right\rangle + \frac{\gamma^2 L}{2} \|\tilde{g}_t(\mathbf{x}_t^{(s)})\|^2 \tag{27}$$

By Lemma 6, we note that,

$$-\gamma \left\langle \nabla F(\mathbf{x}_t^{(s)}), \tilde{g}_t(\mathbf{x}_t^{(s)}) \right\rangle = -\frac{\gamma}{K} \sum_{r \in \mathcal{K}_t} \left\langle \nabla F(\mathbf{x}_t^{(s)}), h(\mathbf{x}_t^{(s)}, \mathbf{x}_t^{(r)}) \right\rangle$$

$$\leq \frac{\gamma}{K} \sum_{r \in \mathcal{K}_t} \left[ -\tfrac{1}{2} k(\mathbf{x}_t^{(s)}, \mathbf{x}_t^{(r)}) \|\nabla F(\mathbf{x}_t^{(s)})\|^2 + L^2 A_1 + A_3 \right]$$

$$\leq -\frac{\gamma c_t^{(s)}}{2} \|\nabla F(\mathbf{x}_t^{(s)})\|^2 + \gamma L^2 A_1 + \gamma A_3 \tag{28}$$

Moreover, by Jensen's Inequality and Lemma 6

$$\|\tilde{g}_t(\mathbf{x}_t^{(s)})\|^2 \leq \frac{1}{K} \sum_{r \in \mathcal{K}_t} \|h(\mathbf{x}_t^{(s)}, \mathbf{x}_t^{(r)})\|^2$$

$$\leq \frac{1}{K} \sum_{r \in \mathcal{K}_t} 2(A_1 L/2 + A_2)^2 + 2k(\mathbf{x}_t^{(s)}, \mathbf{x}_t^{(r)})^2 \|F(\mathbf{x}_t^{(s)})\|^2$$

$$\leq \frac{1}{K} \sum_{r \in \mathcal{K}_t} 2(A_1 L/2 + A_2)^2 + 2A_1 k(\mathbf{x}_t^{(s)}, \mathbf{x}_t^{(r)}) \|F(\mathbf{x}_t^{(s)})\|^2$$

$$\leq 2(A_1 L/2 + A_2)^2 + 2A_1 c_t^{(s)} \|F(\mathbf{x}_t^{(s)})\|^2 \tag{29}$$

Substituting (28) and (29) into (27), we obtain,

$$F(\mathbf{x}_{t+1}^{(s)}) - F(\mathbf{x}_t^{(s)}) \leq -\frac{\gamma c_t^{(s)}}{2} \|\nabla F(\mathbf{x}_t^{(s)})\|^2 + \gamma L^2 A_1 + \gamma A_3$$

$$+ \gamma^2 L(A_1 L/2 + A_2)^2 + \gamma^2 L A_1 c_t^{(s)} \|F(\mathbf{x}_t^{(s)})\|^2$$

$$\leq -\frac{\gamma c_t^{(s)}}{2}(1 - 2A_1 L\gamma) \|\nabla F(\mathbf{x}_t^{(s)})\|^2 + \gamma A_3 + \gamma L^2 A_1 + \gamma^2 L(A_1 L/2 + A_2)^2$$

$$\leq \gamma A_3 + \gamma L^2 A_1 + \gamma^2 L(A_1 L/2 + A_2)^2$$

where the last inequality uses the fact that $c_t^{(s)} \geq 0$ and $\gamma \leq 1/2A_1L$. Now, iterating through the above inequality, we obtain the following for any $t \in [T], \ s \in \mathbb{N} \cup \{0\}$

$$F(\mathbf{x}_t^{(s)}) \leq F(\mathbf{x}_0^{(s)}) + \gamma T L^2 A_1 + \gamma T A_3 + \gamma^2 T L(A_1 L/2 + A_2)^2 \tag{30}$$

Furthermore, by Assumption 1

$$F(\mathbf{x}_0^{(s)}) \leq F(0) + \|\nabla F(0)\|\|\mathbf{x}_0^{(s)}\| + \frac{L}{2}\|\mathbf{x}_0^{(s)}\|^2$$

$$\leq F(0) + 1/2 + L\|\mathbf{x}_0^{(s)}\|^2$$

Substituting the above inequality into (30), and using Assumption 2, we obtain the following for any $t \in [T]$, $s \in \mathbb{N} \cup \{0\}$

$$d_1\|\mathbf{x}_t^{(s)}\|^\alpha - d_2 \leq F(\mathbf{x}_t^s) \leq F(0) + 1/2 + L\|\mathbf{x}_0^{(s)}\|^2 + \gamma T L^2 A_1 + \gamma T A_3$$
$$+ \gamma^2 T L (A_1 L/2 + A_2)^2$$

Rearranging and applying Assumption 5, we obtain

$$\|\mathbf{x}_t^{(s)}\| \leq d_1^{-1/\alpha} \left[ F(0) + 1/2 + LR^2 + \gamma T L^2 A_1 + \gamma T A_3 + \gamma^2 T L (A_1 L + A_2)^2 \right]^{1/\alpha}$$
$$\leq \tilde{\zeta}_0 + \tilde{\zeta}_1 (\gamma T)^{1/\alpha} + \tilde{\zeta}_2 (\gamma^2 T)^{1/\alpha} + \tilde{\zeta}_3 R^{2/\alpha}$$

where $\tilde{\zeta}_0, \ldots, \tilde{\zeta}_3$ are constants that depend polynomially on $L, A_1, A_2, A_3, R$. We note that, since $0 < \alpha \leq 2$, the above inequality also holds for $t = 0$.

Using the above inequality Lemma 6 and Assumption 1, we conclude that the following holds almost surely for any $t \in (T+1), s \in \mathbb{N} \cup \{0\}$

$$\|h(\cdot, \mathbf{x}_t^{(s)})\|_{\mathcal{H}} \leq BL\|\mathbf{x}_t^{(s)}\| + B\sqrt{L} + B$$
$$\leq \tilde{\eta}_0 + \tilde{\eta}_1 (\gamma T)^{1/\alpha} + \tilde{\eta}_2 (\gamma^2 T)^{1/\alpha} + \tilde{\eta}_3 R^{2/\alpha}$$

where $\tilde{\eta}_0, \ldots, \tilde{\eta}_3$ are constants that depend polynomially on $L, B, A_1, A_2, A_3, R$. Using the above inequality, we conclude that the following also holds for any $t \in (T+1)$.

$$\|\tilde{g}_t\|_{\mathcal{H}} \leq \tilde{\eta}_0 + \tilde{\eta}_1 (\gamma T)^{1/\alpha} + \tilde{\eta}_2 (\gamma^2 T)^{1/\alpha} + \tilde{\eta}_3 R^{2/\alpha}$$

Taking $\zeta_i = \max\{\tilde{\zeta}_i, \tilde{\eta}_i\}$, the proof is complete. $\qquad\square$

### D.1 Proof of Theorem 2

*Proof.* Let $\xi = \zeta_0 + \zeta_1 (\gamma T)^{1/\alpha} + \zeta_2 (\gamma^2 T)^{1/\alpha} + \zeta_3 R^{2/\alpha}$ where $\zeta_0, \ldots, \zeta_3$ are constants as described in Lemma 9 and Lemma 10. Since the assumptions and parameter settings of Theorem 1 holds, $\gamma \leq 1/2A_1 L$ and thus, by Lemma 9 and Lemma 10, the particles output by VP-SVGD and GB-SVGD are bounded as $\|\mathbf{y}^{(l)}\| \leq \xi$ and $\|\bar{\mathbf{y}}^{(l)}\| \leq \xi$.

Let $\mathbf{Y} = (\mathbf{y}^{(0)}, \ldots, \mathbf{y}^{(n-1)})$ and $\bar{\mathbf{Y}} = (\bar{\mathbf{y}}^{(0)}, \ldots, \bar{\mathbf{y}}^{(n-1)})$ denote the outputs of VP-SVGD and GB-SVGD. Let $\hat{\mu}^{(n)} = \frac{1}{n} \sum_{i=0}^{n-1} \delta_{\mathbf{y}^{(i)}}$ and $\hat{\nu}^{(n)} = \frac{1}{n} \sum_{i=0}^{n-1} \delta_{\bar{\mathbf{y}}^{(i)}}$ be their respective empirical distributions. We shall now explicitly construct a coupling between the inputs of VP-SVGD and GB-SVGD such that the first $n - KT$ particles of their respective outputs are equal. This in turn will allow us to control the expected squared KSD between $\hat{\mu}^{(n)}$ and $\hat{\nu}^{(n)}$.

To this end, let $\mathcal{E}$ denote the event that each random batch $\mathcal{K}_t$ of GB-SVGD is disjoint and contains unique elements for every $t \in (T)$. Subsequently, let $\mathcal{K}$ denote the set of all indices that were chosen to be part of some random batch $\mathcal{K}_t$. Let $\Lambda$ be a uniformly random permutation over $\{0, \ldots, n-1\}$. We note that, conditioned on $\mathcal{E}$, the distribution of the random set $\mathcal{K}$ is the same as the distribution of $\{\Lambda(0), \ldots, \Lambda(KT-1)\}$. We can couple a uniformly random permutation $\Lambda$ and $\mathcal{K}_t$ for $0 \leq t \leq T$ such that under the event $\mathcal{E}$, $\mathcal{K} = \{\Lambda(0), \ldots, \Lambda(KT-1)\}$ and $\{\Lambda(tK), \ldots, \Lambda((t+1)K-1)\}$ is the random batch $\mathcal{K}_t$. Thus, under the event $\mathcal{E}$, one can couple a uniformly random permutation $\Lambda$ and $\mathcal{K}_t$ for $t \in (T)$ such that $\mathcal{K} = \{\Lambda(0), \ldots, \Lambda(KT-1)\}$ and $\mathcal{K}_t = \{\Lambda(tK), \ldots, \Lambda((t+1)K-1)\}$

With this insight, we couple VP-SVGD and GB-SVGDas follows. We note that, the random batch $\mathcal{K}_t$ in GB-SVGD is sampled independently of the initial particles. To this end, let $\bar{\mathbf{x}}_0^{(0)}, \ldots, \bar{\mathbf{x}}_0^{(n-1)} \overset{i.i.d.}{\sim} \mu_0$, and let the random batches $\mathcal{K}_t$ and permutation $\Lambda$ be jointly distributed as described above, independently of $\bar{\mathbf{x}}_0^{(0)}, \ldots, \bar{\mathbf{x}}_0^{(n-1)}$, i.e.

$$\Lambda \sim \mathsf{Uniform}(\mathbb{S}_{(n)}), \quad \mathcal{K}_t = \{\Lambda(tK), \ldots, \Lambda((t+1)K-1)\}, \quad t \in (T)$$

We now define $\mathbf{x}_0^{(0)}, \ldots, \mathbf{x}_0^{(KT+n-1)}$ as:

$$\mathbf{x}_0^{(l)} := \begin{cases} = \bar{\mathbf{x}}_0^{(\Lambda(l))} & \text{for } 0 \le l \le n-1 \\ \sim \mu_0 \text{ independent of everything else} & \text{for } n \le l \le KT+n-1 \end{cases} \quad (31)$$

Let $\bar{\mathbf{x}}_0^{(0)}, \ldots, \bar{\mathbf{x}}_0^{(n-1)}$ and $\mathcal{K}_t$ as the initialization and random batches for GB-SVGD, and let $\mathbf{x}_0^{(0)}, \ldots, \mathbf{x}_0^{(KT+n-1)}$ be the initialization for GB-SVGD. We first show that this construction is indeed a valid coupling between VP-SVGD and GB-SVGD.

**Claim 1.** *Conditioned on $\mathcal{E}$, the inputs to VP-SVGD and GB-SVGD, as constructed above is a valid coupling, i.e., the marginal distribution of $\mathbf{x}_0^{(0)}, \ldots, \mathbf{x}_0^{(KT+n-1)}$ is equal to the distribution of initial particles in VP-SVGD, and the marginal distribution of $\bar{\mathbf{x}}_0^{(0)}, \ldots, \bar{\mathbf{x}}_0^{(n-1)}, (\mathcal{K}_t)_{t \in (T)}$ is the same as the distribution of initial particles and random batches in $\mathcal{K}_t$*

*Proof.* By construction $\bar{\mathbf{x}}_0^{(0)}, \ldots, \bar{\mathbf{x}}_0^{(n-1)} \overset{i.i.d.}{\sim} \mu_0$. Moreover, conditioned on $\mathcal{E}$, the distribution of $\mathcal{K}_t = \{\Lambda(tK), \ldots, \Lambda((t+1)K-1)\}$, has the distribution of a uniform random batch of size $K$ since $\Lambda \sim \mathsf{Uniform}(\mathbb{S}_n)$. Furthermore, since $\Lambda$ is sampled independently of $\bar{\mathbf{x}}_0^{(0)}, \ldots, \bar{\mathbf{x}}_0^{(n-1)}, \mathcal{K}_t$ is independent of $\bar{\mathbf{x}}_0^{(0)}, \ldots, \bar{\mathbf{x}}_0^{(n-1)}$ for any $t \in (T)$. Thus, the coupling constructed above has the correct marginal with respect to GB-SVGD.

To establish the same for VP-SVGD, we note that by (31), $\mathbf{x}_0^{(n)}, \ldots, \mathbf{x}_0^{(KT+n-1)} \overset{i.i.d.}{\sim} \mu_0$, *sampled independently of everything else*. Moreover, since $\bar{\mathbf{x}}_0^{(0)}, \ldots, \bar{\mathbf{x}}_0^{(n-1)} \overset{i.i.d.}{\sim} \mu_0$, we infer that $\bar{\mathbf{x}}_0^{(\Lambda(0))}, \ldots, \bar{\mathbf{x}}_0^{(\Lambda(n-1))} \overset{i.i.d.}{\sim} \mu_0$ for any arbitrary permutation $\Lambda \in \mathbb{S}_n$. From this, and (31), we conclude that $\mathbf{x}_0^{(0)}, \ldots, \mathbf{x}_0^{(KT+n-1)} \overset{i.i.d.}{\sim} \mu_0$. Hence, the coupling constructed above has the correct marginal with respect to VP-SVGD. $\square$

We now show that, under the constructed coupling, the time-evolution of the particles of VP-SVGD and GB-SVGD satisfy $\bar{\mathbf{x}}_t^{(\Lambda(l))} = \mathbf{x}_t^{(l)}$, $KT \le l \le n-1, t \in (T+1)$, when conditioned on the event $\mathcal{E}$.

**Claim 2.** *Let the inputs to VP-SVGD and GB-SVGD be coupled as per the construction above. Then, conditioned on the event $\mathcal{E}$, the particles $\mathbf{x}_t^{(s)}$ and $\bar{\mathbf{x}}_t^{(s)}$ of VP-SVGD and GB-SVGD respectively, satisfy $\bar{\mathbf{x}}_t^{(\Lambda(l))} = \mathbf{x}_t^{(l)}$ for every $KT \le l \le n-1$ and $0 \le t \le T$*

*Proof.* We prove this by an inductive argument. Clearly, the claim holds for $t = 0$ by the construction of our coupling. Assume it holds for some arbitrary $t \in (T)$. Now, writing the update equation for GB-SVGD for $KT \le l \le n-1$,

$$\bar{\mathbf{x}}_{t+1}^{(\Lambda(l))} = \bar{\mathbf{x}}_t^{(\Lambda(l))} - \frac{\gamma}{K} \sum_{r \in \mathcal{K}_t} h(\bar{\mathbf{x}}_t^{(\Lambda(l))}, \bar{\mathbf{x}}_t^{(r)})$$

$$= \bar{\mathbf{x}}_t^{(\Lambda(l))} - \frac{\gamma}{K} \sum_{l=0}^{K-1} h(\bar{\mathbf{x}}_t^{(\Lambda(l))}, \bar{\mathbf{x}}_t^{(\Lambda(Kt+l))})$$

$$= \mathbf{x}_t^{(l)} - \frac{\gamma}{K} \sum_{l=0}^{K-1} h(\mathbf{x}_t^{(l)}, \mathbf{x}_t^{(Kt+l)}) = \mathbf{x}_t^{(l+1)}$$

where the second equality uses the fact that $\mathcal{K}_t = \{\Lambda(tK), \ldots, \Lambda((t+1)K-1)\}$ when conditioned on $\mathcal{E}$ and the third equality uses the induction hypothesis $\bar{\mathbf{x}}_t^{(\Lambda(l))} = \mathbf{x}_t^{(l)}$ for $KT \le l \le n-1$. Hence, the claim is proven true by induction. $\square$

Equipped with the above coupling between the inputs of VP-SVGD and GB-SVGD, one can now couple their outputs by sampling an $S \sim \mathsf{Uniform}((n))$ and using this sampled $S$ as the random timestep chosen by both VP-SVGD (Step 6 in Algorithm 1) and GB-SVGD (Step 7 in Algorithm 2) that are run with the coupled input constructed above. It is easy to see that this results in a coupling

of the outputs $\mathbf{Y}$ and $\bar{\mathbf{Y}}$ of VP-SVGD and GB-SVGD respectively. Furthermore, by Claim 2, we note that, conditioned on the event $\mathcal{E}$, $\mathbf{y}^{(l-TK)} = \bar{\mathbf{y}}^{(\Lambda(l))}$ for every $KT \leq l \leq n-1$. We now define the permutation $\tau \in \mathbb{S}_{(n)}$ as follows:

$$
\tau(\Lambda(l)) = \begin{cases} l + n - KT & \text{for } 0 \leq l \leq KT - 1 \\ l - KT & \text{for } KT \leq l \leq n - 1 \end{cases} \tag{32}
$$

It follows that $\bar{\mathbf{y}}^{\tau(l)} = \mathbf{y}^{(l)}$ for $KT \leq l \leq n-1$. Thus, by definition of Kernel Stein Discrepancy (Definition 1), we can infer that the following holds when conditioned on the event $\mathcal{E}$

$$
\begin{aligned}
\mathbb{E}[\mathsf{KSD}_{\pi^\star}^2(\hat{\nu}^{(n)} || \hat{\mu}^{(n)}) | \mathcal{E}] &= \mathbb{E}\left[ \|h_{\hat{\nu}^{(n)}} - h_{\hat{\mu}^{(n)}}\|_{\mathcal{H}}^2 \mid \mathcal{E} \right] \\
&= \mathbb{E}\left[ \|\frac{1}{n} \sum_{l=0}^{n-1} h(\cdot, \bar{\mathbf{y}}^{(l)}) - \frac{1}{n} \sum_{l=0}^{n-1} h(\cdot, \mathbf{y}^{(l)})\|_{\mathcal{H}}^2 \mid \mathcal{E} \right] \\
&= \frac{1}{n^2} \mathbb{E}[\| \sum_{l=0}^{n-1} h(\cdot, \bar{\mathbf{y}}^{(\tau(l))}) - h(\cdot, \mathbf{y}^{(l)})\|_{\mathcal{H}}^2 \mid \mathcal{E}] \\
&= \frac{1}{n^2} \mathbb{E}[\| \sum_{l=0}^{KT-1} h(\cdot, \bar{\mathbf{y}}^{(\tau(l))}) - h(\cdot, \mathbf{y}^{(l)})\|_{\mathcal{H}}^2 \mid \mathcal{E}] \\
&\leq \frac{KT}{n^2} \sum_{l=0}^{KT-1} \mathbb{E}[\|h(\cdot, \bar{\mathbf{y}}^{(\tau(l))}) - h(\cdot, \mathbf{y}^{(l)})\|_{\mathcal{H}}^2 \mid \mathcal{E}] \\
&\leq \frac{2K^2 T^2 \xi^2}{n^2}
\end{aligned} \tag{33}
$$

where the second step uses the permutation invariance of summation, the third step uses the fact that $\bar{\mathbf{y}}^{\tau(l)} = \bar{\mathbf{y}}^{(l)}$ for $KT \leq l \leq n-1$, the fourth step uses the convexity of $\|\cdot\|_{\mathcal{H}}^2$ and the last step uses the almost-sure iterate bounds of Lemma 9 and 10

Under the event $\mathcal{E}^c$, we directly apply the almost-sure iterate bounds of Lemma 9 and 10 to obtain the following:

$$
\begin{aligned}
\mathbb{E}[\mathsf{KSD}_{\pi^\star}^2(\hat{\nu}^{(n)} || \hat{\mu}^{(n)}) | \mathcal{E}^c] &= \mathbb{E}\left[ \|h_{\hat{\mu}^{(n)}} - h_{\hat{\nu}^{(n)}}\|_{\mathcal{H}}^2 \mid \mathcal{E}^c \right] \\
&= \frac{1}{n^2} \mathbb{E}[\| \sum_{l=0}^{n-1} h(\cdot, \bar{\mathbf{y}}^{(l)}) - h(\cdot, \mathbf{y}^{(l)})\|_{\mathcal{H}}^2 \mid \mathcal{E}^c] \\
&\leq 2\xi^2
\end{aligned} \tag{34}
$$

From Equations (33) and (34), it follows that:

$$
\begin{aligned}
\mathbb{E}[\mathsf{KSD}_{\pi^\star}^2(\hat{\nu}^{(n)} || \hat{\mu}^{(n)})] &= \mathbb{E}[\mathsf{KSD}_{\pi^\star}^2(\hat{\nu}^{(n)} || \hat{\mu}^{(n)}) | \mathcal{E}] \mathbb{P}(\mathcal{E}) + \mathbb{E}[\mathsf{KSD}_{\pi^\star}^2(\hat{\nu}^{(n)} || \hat{\mu}^{(n)}) | \mathcal{E}^c] \mathbb{P}(\mathcal{E}^c) \\
&\leq \frac{2K^2 T^2 \xi^2}{n^2} \mathbb{P}(\mathcal{E}) + 2\xi^2 \mathbb{P}(\mathcal{E}^c)
\end{aligned}
$$

Recall that $P(\mathcal{E}) = 1$ under sampling without replacement and $P(\mathcal{E}) = 1 - \frac{K^2 T^2}{n}$ under sampling with replacement. Thus, we conclude that the following holds under the constructed coupling of $\mathbf{Y}$ and $\bar{\mathbf{Y}}$

$$
\mathbb{E}[\mathsf{KSD}_{\pi^\star}^2(\hat{\nu}^{(n)} || \hat{\mu}^{(n)})] \leq \begin{cases} \frac{2K^2 T^2 \xi^2}{n^2} & \text{(without replacement sampling)} \\ \frac{2K^2 T^2 \xi^2}{n^2}\left(1 - \frac{K^2 T^2}{n}\right) + \frac{2K^2 T^2 \xi^2}{n} & \text{(with replacement sampling)} \end{cases}
$$

$\square$

# E   Finite-Particle Convergence Guarantees for VP-SVGD and GB-SVGD

In this section, we show that the empirical measure of the particles output by VP-SVGD and GB-SVGD rapidly converge to the target distribution $\pi^\star$ in KSD. To this end, we prove the finite-particle

convergence rates for VP-SVGD in Appendix E.1 and that of GB-SVGD in Appendix E.2. Finally, we compare the oracle complexity (i.e., the number of evaluations of $\nabla F$) of VP-SVGD and GB-SVGD to that of SVGD in Appendix E.3

## E.1 VP-SVGD

**Corollary 2 (VP-SVGD : Fast Finite-Particle Convergence).** *Let the assumptions and parameter settings of Theorem 1 be satisfied. Let $\hat{\mu}^{(n)}$ denote the empirical measure of the $n$ particles output by VP-SVGD.*

$$\mathbb{E}[\mathsf{KSD}^2_{\pi^\star}(\hat{\mu}^{(n)}||\pi^\star)] \leq \frac{\xi^2}{n} + \frac{2\mathsf{KL}\left(\mu_0|\mathcal{F}_0||\pi^\star\right)}{\gamma T} + \frac{\gamma B(4+L)\xi^2}{K}$$

*where $\xi$ is as defined in Theorem 1. Setting $R = \sqrt{d/L}, \gamma = O(\frac{(Kd)^\eta}{T^{1-\eta}})$ with $\eta = \frac{\alpha}{2(1+\alpha)}$ and $KT = d^{\frac{\alpha}{2+\alpha}} n^{\frac{2(1+\alpha)}{2+\alpha}}$ suffices to ensure,*

$$\mathbb{E}[\mathsf{KSD}^2_{\pi^\star}(\hat{\mu}^{(n)}||\pi^\star)] \leq O\left(\frac{d^{\frac{2}{2+\alpha}}}{n^{\frac{\alpha}{2+\alpha}}} + \frac{d^{2/\alpha}}{n}\right)$$

*Proof.* Recall from Algorithm 1 that the outputs of VP-SVGD are $\mathbf{x}_S^{(KT)}, \ldots, \mathbf{x}_S^{(KT+n-1)}$ where $S \sim$ Uniform$(\{0, \ldots, T-1\})$. Hence, their empirical measure $\hat{\mu}^{(n)}$ is given by $\hat{\mu}^{(n)} = \frac{1}{n} \sum_{l=0}^{n-1} \delta_{\mathbf{x}_S^{(KT+l)}}$. From the definition of the Kernel Stein Discrepancy (Definition 1), it follows that,

$$\mathsf{KSD}^2_{\pi^\star}(\hat{\mu}^{(n)}||\pi^\star) = \|h_{\hat{\mu}^{(n)}}\|^2_{\mathcal{H}} = \|\frac{1}{n} \sum_{l=1}^{N} h(\cdot, \mathbf{x}_S^{(KT+l)})\|^2_{\mathcal{H}} \tag{35}$$

For the sake of clarity, only in this proof, we use $\mathcal{C}$ to denote the conditioning on the virtual particles $\mathbf{x}_0^{(0)}, \ldots, \mathbf{x}_0^{(KT-1)}$. Now, consider any arbitrary $t \in \{0, \ldots, T-1\}$. Taking conditional expectations on both sides of Equation (35) by conditioning on $\mathcal{C}$ and the event $\{S = t\}$, we obtain the following:

$$\mathbb{E}\left[\mathsf{KSD}^2_{\pi^\star}(\hat{\mu}^{(n)}||\pi^\star) \mid \mathcal{C}, S = t\right] = \mathbb{E}\left[\|\frac{1}{n}\sum_{l=0}^{n-1} h(\cdot, \mathbf{x}_S^{(KT+l)})\|^2_{\mathcal{H}} \mid \mathcal{C}, S = t\right]$$

$$= \mathbb{E}\left[\|\frac{1}{n}\sum_{l=0}^{n-1} h(\cdot, \mathbf{x}_t^{(KT+l)})\|^2_{\mathcal{H}} \mid (\mathbf{x}_0^{(s)})_{0 \leq s \leq KT-1}\right] \tag{36}$$

Recall from Equation (14) in Appendix C.1 that for any $l \in \{0, \ldots, n-1\}$ $\mathbf{x}_t^{(KT+l)}$ depends only on $\mathbf{x}_0^{(0)}, \ldots, \mathbf{x}_0^{(Kt-1)}$ and $\mathbf{x}_0^{(KT+l)}$. Furthermore, from Appendix C.1, we recall that the filtration $\mathcal{F}_t$ is defined as $\mathcal{F}_t = \sigma(\{\mathbf{x}_0^{(0)}, \ldots, \mathbf{x}_0^{(Kt-1)}\})$. It follows that,

$$\mathbb{E}\left[\|\frac{1}{n}\sum_{l=0}^{n-1} h(\cdot, \mathbf{x}_t^{(KT+l)})\|^2_{\mathcal{H}} \mid (\mathbf{x}_0^{(s)})_{0 \leq s \leq KT-1}\right] = \mathbb{E}\left[\|\frac{1}{n}\sum_{l=1}^{N} h(\cdot, \mathbf{x}_t^{(KT+l)})\|^2_{\mathcal{H}} \mid (\mathbf{x}_0^{(s)})_{0 \leq s \leq Kt-1}\right]$$

$$= \mathbb{E}\left[\|\frac{1}{n}\sum_{l=0}^{n-1} h(\cdot, \mathbf{x}_t^{(KT+l)})\|^2_{\mathcal{H}} \mid \mathcal{F}_t\right] \tag{37}$$

To control $\mathbb{E}\left[\|\frac{1}{n}\sum_{l=0}^{n-1} h(\cdot, \mathbf{x}_t^{(KT+l)})\|^2_{\mathcal{H}} \mid \mathcal{F}_t\right]$, we apply the arguments used in the proof of Lemma 3. To this end, note that when conditioned on the virtual particles $\mathbf{x}_0^{(0)}, \ldots, \mathbf{x}_0^{(Kt-1)}$, the particles $\mathbf{x}_t^{(KT)}, \ldots, \mathbf{x}_t^{(KT+n-1)} \overset{i.i.d.}{\sim} \mu_t|\mathcal{F}_t$. Furthermore, since $\gamma \leq 1/2A_1L$ (as per the parameter settings of Theorem 1), $\|h(\cdot, \mathbf{x}_t^{(KT+l)})\|_{\mathcal{H}} \leq \xi \ \forall \ l \in (n)$ by Lemma 6. Finally, $\mathbb{E}[h(\mathbf{x}, \mathbf{x}_t^{(KT+l)})|\mathcal{F}_t] = h_{\mu_t|\mathcal{F}_t}(\mathbf{x}) \ \forall \ l \in (n), \ \mathbf{x} \in \mathbb{R}^d$. Hence, from Lemma 8, we conclude that $h_{\mu_t|\mathcal{F}_t}$ is the Gelfand-Pettis integral of the map $\mathbf{x} \to h(\mathbf{x}, \mathbf{x}_t^{(KT+l)})$ with respect to the measure $\mu_t|\mathcal{F}_t$, i.e.,

$$\mathbb{E}[\left\langle h(\cdot, \mathbf{x}_t^{(KT+l)}), f\right\rangle_{\mathcal{H}} |\mathcal{F}_t] = \left\langle h_{\mu_t|\mathcal{F}_t}, f\right\rangle \ \forall f \in \mathcal{H} \tag{38}$$

To control $\mathbb{E}\left[\|\frac{1}{n}\sum_{l=0}^{n-1}h(\cdot,\mathbf{x}_t^{(KT+l)})\|_{\mathcal{H}}^2 \mid \mathcal{F}_t\right]$, we proceed as follows:

$$\|\frac{1}{n}\sum_{l=0}^{n-1}h(\cdot,\mathbf{x}_t^{(KT+l)})\|_{\mathcal{H}}^2 = \frac{1}{n^2}\sum_{l_1,l_2=0}^{n-1}\left\langle h(\cdot,\mathbf{x}_t^{(Kt+l_1)}), h(\cdot,\mathbf{x}_t^{(Kt+l_2)})\right\rangle_{\mathcal{H}}$$

$$= \frac{1}{n^2}\sum_{l=0}^{m-1}\|h(\cdot,\mathbf{x}_t^{(Kt+l)})\|_{\mathcal{H}}^2 + \frac{1}{n^2}\sum_{0\le l_1\neq l_2\le n-1}\left\langle h(\cdot,\mathbf{x}_t^{(Kt+l_1)}), h(\cdot,\mathbf{x}_t^{(Kt+l_2)})\right\rangle_{\mathcal{H}}$$

$$\le \frac{\xi^2}{n} + \frac{1}{n^2}\sum_{0\le l_1\neq l_2\le n-1}\left\langle h(\cdot,\mathbf{x}_t^{(Kt+l_1)}), h(\cdot,\mathbf{x}_t^{(Kt+l_2)})\right\rangle_{\mathcal{H}}$$

where the last inequality uses the fact that $\|h(\cdot,\mathbf{x}_t^{(Kt+l)})\|_{\mathcal{H}}\le\xi$ almost surely as per Lemma 9.

To control the conditional expectation of the off-diagonal terms, let $i = Kt+l_1$ and $j = Kt+l_2$ for any arbitrary $l_1, l_2$ with $0\le l_1\neq l_2\le n-1$. Conditioned on $\mathcal{F}_t$, $\mathbf{x}_t^{(i)}$ and $\mathbf{x}_t^{(j)}$ are i.i.d samples from $\mu_t|\mathcal{F}_t$. Thus, by Equation (38) and Fubini's Theorem,

$$\mathbb{E}_{\mathbf{x}_t^{(i)},\mathbf{x}_t^{(j)}}\left[\left\langle h(\cdot,\mathbf{x}_t^{(i)}), h(\cdot,\mathbf{x}_t^{(j)})\right\rangle_{\mathcal{H}}|\mathcal{F}_t\right] = \mathbb{E}_{\mathbf{x}_t^{(i)}}\left[\mathbb{E}_{\mathbf{x}_t^{(j)}}\left[\left\langle h(\cdot,\mathbf{x}_t^{(i)}), h(\cdot,\mathbf{x}_t^{(j)})\right\rangle_{\mathcal{H}}|\mathcal{F}_t\right]\right]$$

$$= \mathbb{E}_{\mathbf{x}_t^{(i)}}\left[\left\langle h_{\mu_t|\mathcal{F}_t}, h(\cdot,\mathbf{x}_t^{(i)})\right\rangle_{\mathcal{H}}|\mathcal{F}_t\right]$$

$$= \|h_{\mu_t|\mathcal{F}_t}\|_{\mathcal{H}}^2$$

It follows that,

$$\mathbb{E}\left[\|\frac{1}{n}\sum_{l=0}^{n-1}h(\cdot,\mathbf{x}_t^{(KT+l)})\|_{\mathcal{H}}^2 \mid \mathcal{F}_t\right] \le \|h_{\mu_t|\mathcal{F}_t}\|_{\mathcal{H}}^2 + \frac{\xi^2}{n}$$

Substituting the above into equation 36 and equation 37, we obtain the following:

$$\mathbb{E}\left[\mathsf{KSD}_{\pi^\star}^2(\hat{\mu}^{(n)}\|\pi^\star)\mid\mathcal{C}, S=t\right] \le \frac{\xi^2}{n} + \|h_{\mu_t|\mathcal{F}_t}\|_{\mathcal{H}}^2 = \frac{\xi^2}{n} + \mathsf{KSD}_{\pi^\star}^2(\mu_t|\mathcal{F}_t\|\pi^*)$$

where the second step applies Definition 1. Finally, taking expectations with respect to $\mathcal{C}$ and $S\sim\mathsf{Uniform}(\{0,\dots,T-1\})$ on both sides of the above inequality, we get:

$$\mathbb{E}\left[\mathsf{KSD}_{\pi^\star}^2(\hat{\mu}^{(n)}\|\pi^\star)\right] \le \frac{\xi^2}{n} + \frac{1}{T}\sum_{t=0}^{T-1}\mathbb{E}[\mathsf{KSD}_{\pi^\star}^2(\mu_t|\mathcal{F}_t\|\pi^\star)]$$

Substituting the bound from Theorem 1 into the above inequality, we conclude that:

$$\mathbb{E}[\mathsf{KSD}_{\pi^\star}^2(\hat{\mu}^{(n)}\|\pi^\star)] \le \frac{\xi^2}{n} + \frac{2\mathsf{KL}\left(\mu_0|\mathcal{F}_0\|\pi^\star\right)}{\gamma T} + \frac{\gamma B(4+L)\xi^2}{K}$$

We note that for $\gamma = O(\frac{(Kd)^\eta}{T^{1-\eta}})$ and $R = \sqrt{d/L}$, $\mathsf{KL}\left(\mu_0|\mathcal{F}_0\|\pi^\star\right) = O(d)$ by Lemma 4 and

$$\xi^2 \le 4\zeta_0 + 4\zeta_1(\gamma T)^{2/\alpha} + 4\zeta_2(\gamma^2 T)^{2/\alpha} + 4\zeta_3 R^{4/\alpha} \le O\left((KdT)^{\frac{1}{1+\alpha}} + d^{2/\alpha}\right)$$

Furthermore,

$$\frac{2\mathsf{KL}\left(\mu_0|\mathcal{F}_0\|\pi^\star\right)}{\gamma T} + \frac{\gamma B(4+L)\xi^2}{K} \le O\left(\frac{d}{\gamma T} + \frac{\gamma B(4+L)\xi^2}{2K}\right) \le O\left(\frac{d^{1-\eta}}{(KT)^\eta}\right)$$

$$\le O\left(\frac{d^{\frac{2+\alpha}{2(1+\alpha)}}}{(KT)^{\frac{\alpha}{2(1+\alpha)}}}\right)$$

It follows that,

$$\mathbb{E}[\mathsf{KSD}_{\pi^\star}^2(\hat{\mu}^{(n)}\|\pi^\star)] \le O\left(\frac{d^{2/\alpha}}{n} + \frac{(KTd)^{\frac{1}{1+\alpha}}}{n} + \frac{d^{\frac{2+\alpha}{2(1+\alpha)}}}{(KT)^{\frac{\alpha}{2(1+\alpha)}}}\right)$$

$KT = d^{\frac{\alpha}{2+\alpha}} n^{\frac{2(1+\alpha)}{2+\alpha}}$, we conclude:

$$\mathbb{E}[\mathsf{KSD}^2_{\pi^\star}(\hat{\mu}^{(n)}||\pi^\star)] \leq O\left(\frac{d^{\frac{2}{2+\alpha}}}{n^{\frac{\alpha}{2+\alpha}}} + \frac{d^{2/\alpha}}{n}\right)$$

$\square$

## E.2  GB-SVGD

**Corollary 3** (**GB-SVGD : Fast Finite-Particle Convergence**). *Let the assumptions and parameter settings of Theorem 1 be satisfied. Let $\hat{\nu}^{(n)}$ denote the empirical measure of the $n$ particles output by GB-SVGD. Then, under without-replacement sampling of the minibatches, the following holds:*

$$\mathbb{E}[\mathsf{KSD}^2_{\pi^\star}(\hat{\nu}^{(n)}||\pi^\star)] \leq \frac{4K^2T^2\xi^2}{n^2} + \frac{2\xi^2}{n} + \frac{4\mathsf{KL}\left(\mu_0|\mathcal{F}_0||\pi^\star\right)}{\gamma T} + \frac{2\gamma B(4+L)\xi^2}{K}$$

*and the following holds under with-replacement sampling of the minibatches*

$$\mathbb{E}[\mathsf{KSD}^2_{\pi^\star}(\hat{\nu}^{(n)}||\pi^\star)] \leq \frac{4K^2T^2\xi^2}{n^2}(1 - \frac{K^2T^2}{n}) + \frac{4K^2T^2\xi^2}{n} + \frac{2\xi^2}{n} + \frac{4\mathsf{KL}\left(\mu_0|\mathcal{F}_0||\pi^\star\right)}{\gamma T} + \frac{2\gamma B(4+L)\xi^2}{K}$$

*where $\xi$ is as defined in Theorem 1. In particular, for GB-SVGD under without-replacement sampling of the minibatches, setting $R = \sqrt{d/L}, \gamma = O(\frac{(Kd)^\eta}{T^{1-\eta}})$ with $\eta = \frac{\alpha}{2(1+\alpha)}$ and $KT = \sqrt{n}$ suffices to ensure the following*

$$\mathbb{E}[\mathsf{KSD}^2_{\pi^\star}(\hat{\nu}^{(n)}||\pi^\star)] \leq O\left(\frac{d^{2/\alpha}}{n} + \frac{d^{\frac{1}{1+\alpha}}}{n^{\frac{1+2\alpha}{2(1+\alpha)}}} + \frac{d^{\frac{2+\alpha}{2(1+\alpha)}}}{n^{\frac{\alpha}{4(1+\alpha)}}}\right)$$

*Proof.* Let $\bar{\mathbf{Y}} = (\bar{\mathbf{y}}^{(0)}, \ldots, \bar{\mathbf{y}}^{(n-1)})$ denote the $n$ particles output by GB-SVGD and let $\hat{\nu}^{(n)} = \frac{1}{n}\sum_{l=0}^{n-1}\delta_{\bar{\mathbf{y}}^{(l)}}$ denote their empirical measure. Let $\mathcal{E}$ denote the event that each random batch $\mathcal{K}_t$ of GB-SVGD is disjoint and contains unique elements for every $t \in (T)$. Moreover, let $\mathbf{Y} = (\mathbf{y}^{(0)}, \ldots, \mathbf{y}^{(n-1)})$ denote the $n$ particles output by VP-SVGD, run with the parameter settings stated above, and coupled with $\bar{\mathbf{Y}}$ as per the coupling constructed in the proof of Theorem 2 in Appendix D.1. Let $\hat{\mu}^{(n)} = \frac{1}{n}\sum_{l=0}^{n-1}\delta_{\mathbf{y}^{(l)}}$ denote their empirical measure. By definition of Kernel Stein Discrepancy (Definition 1) and the convexity $\|\cdot\|^2_{\mathcal{H}}$, it follows that:

$$\begin{aligned}\mathbb{E}[\mathsf{KSD}_{\pi^\star}(\hat{\nu}^{(n)}||\pi^\star)] &= \mathbb{E}[\|h_{\hat{\nu}^{(n)}}\|^2_{\mathcal{H}}] \\ &= \mathbb{E}[\|h_{\hat{\nu}^{(n)}} - h_{\hat{\mu}^{(n)}} + h_{\hat{\mu}^{(n)}}\|^2_{\mathcal{H}}] \\ &\leq 2\mathbb{E}[\|h_{\hat{\nu}^{(n)}} - h_{\hat{\mu}^{(n)}}\|^2_{\mathcal{H}}] + 2\mathbb{E}[\|h_{\hat{\mu}^{(n)}}\|^2_{\mathcal{H}}] \\ &= 2\mathbb{E}[\mathsf{KSD}^2_{\pi^\star}(\hat{\nu}^{(n)}||\hat{\mu}^{(n)})] + 2\mathbb{E}[\mathsf{KSD}^2_{\pi^\star}(\hat{\mu}^{(n)}||\pi^\star)]\end{aligned}$$

Substituting the bounds of Theorem 2 and Corollary 2 into the above inequality, we conclude the following:

$$\mathbb{E}[\mathsf{KSD}^2_{\pi^\star}(\hat{\nu}^{(n)}||\pi^\star)] \leq \frac{4K^2T^2\xi^2}{n^2}\mathbb{P}(\mathcal{E}) + 4\xi^2\mathbb{P}(\mathcal{E}^c) + \frac{2\xi^2}{n} + \frac{4\mathsf{KL}\left(\mu_0|\mathcal{F}_0||\pi^\star\right)}{\gamma T} + \frac{2\gamma B(4+L)\xi^2}{K}$$

We recall that, $\mathbb{P}(\mathcal{E}) = 1$ under without-replacement sampling of the random batches $\mathcal{K}_t$ and $\mathbb{P}(\mathcal{E}) = 1 - K^2T^2/n$ under with-replacement sampling. Thus, under without-replacement sampling, the following holds:

$$\mathbb{E}[\mathsf{KSD}^2_{\pi^\star}(\hat{\nu}^{(n)}||\pi^\star)] \leq \frac{4K^2T^2\xi^2}{n^2} + \frac{2\xi^2}{n} + \frac{4\mathsf{KL}\left(\mu_0|\mathcal{F}_0||\pi^\star\right)}{\gamma T} + \frac{2\gamma B(4+L)\xi^2}{K}$$

Moreover, the following holds under with-replacement sampling

$$\mathbb{E}[\mathsf{KSD}^2_{\pi^\star}(\hat{\nu}^{(n)}||\pi^\star)] \leq \frac{4K^2T^2\xi^2}{n^2}\left(1 - \frac{K^2T^2}{n}\right) + \frac{4K^2T^2\xi^2}{n} + \frac{2\xi^2}{n} + \frac{4\mathsf{KL}\left(\mu_0|\mathcal{F}_0||\pi^\star\right)}{\gamma T} + \frac{2\gamma B(4+L)\xi^2}{K}$$

Now, let us consider GB-SVGD without replacement with $R = \sqrt{d/L}$, $\gamma = O(\frac{(Kd)^\eta}{T^{1-\eta}})$ and $KT = n^{1/2}$ It follows that $\mathsf{KL}\,(\mu_0|\mathcal{F}_0||\pi^\star) = O(d)$ by Lemma 4 and

$$\xi^2 \leq 4\zeta_0 + 4\zeta_1(\gamma T)^{2/\alpha} + 4\zeta_2(\gamma^2 T)^{2/\alpha} + 4\zeta_3 R^{4/\alpha}$$

$$\leq O\left((KdT)^{\frac{1}{1+\alpha}} + d^{2/\alpha}\right)$$

$$\leq O\left(d^{2/\alpha} + d^{\frac{1}{1+\alpha}} n^{\frac{1}{2(1+\alpha)}}\right)$$

Furthermore,

$$\frac{4\mathsf{KL}\,(\mu_0|\mathcal{F}_0||\pi^\star)}{\gamma T} + \frac{2\gamma B(4+L)\xi^2}{K} \leq O\left(\frac{d}{\gamma T} + \frac{\gamma B(4+L)\xi^2}{2K}\right) \leq O\left(\frac{d^{1-\eta}}{(KT)^\eta}\right)$$

$$\leq O\left(\frac{d^{\frac{2+\alpha}{2(1+\alpha)}}}{n^{\frac{\alpha}{4(1+\alpha)}}}\right)$$

Hence, we conclude that,

$$\mathbb{E}[\mathsf{KSD}^2_{\pi^\star}(\hat\nu^{(n)}||\pi^\star)] \leq \frac{4K^2T^2\xi^2}{n^2} + \frac{2\xi^2}{n} + \frac{4\mathsf{KL}\,(\mu_0|\mathcal{F}_0||\pi^\star)}{\gamma T} + \frac{2\gamma B(4+L)\xi^2}{K}$$

$$\leq \frac{6\xi^2}{n} + \frac{4\mathsf{KL}\,(\mu_0|\mathcal{F}_0||\pi^\star)}{\gamma T} + \frac{2\gamma B(4+L)\xi^2}{K}$$

$$\leq O\left(\frac{d^{2/\alpha}}{n} + \frac{d^{\frac{1}{1+\alpha}}}{n^{\frac{1+2\alpha}{2(1+\alpha)}}} + \frac{d^{\frac{2+\alpha}{2(1+\alpha)}}}{n^{\frac{\alpha}{4(1+\alpha)}}}\right)$$

$\square$

### E.3 Oracle Complexity of SVGD, VP-SVGD and GB-SVGD

We now compare the gradient oracle complexity, (i.e., the number of evaluations of $\nabla F$) of VP-SVGD (as implied by Corollary 2) and GB-SVGD (as implied by Corollary 3) with that of SVGD as implied by the state-of-the-art finite particle guarantee of Shi and Mackey [43].

#### E.3.1 SVGD

From Equation (1), We note that $T$ steps of SVGD run with $n$ particles requires $n^2T$ evaluations of $\nabla F$.

**Subgaussian $\pi^\star$** For subgaussian $\pi^\star$, the finite-particle convergence rate obtained by Shi and Mackey [43] is $\mathsf{KSD}_{\pi^\star}(\hat\mu^{(n)}_{\mathsf{SVGD}}||\pi^\star) = \tilde{O}(\frac{\mathsf{poly}(d)}{\sqrt{\log\log n^{\Theta(1/d)}}})$, where $\hat\mu^{(n)}_{\mathsf{SVGD}}$ denotes the empirical measure of the $n$ particles output by SVGD. By carefully following the analysis of Shi and Mackey [43], we infer that, to achieve $\mathsf{KSD}_{\pi^\star}(\hat\mu^{(n)}_{\mathsf{SVGD}}||\pi^\star) \leq \epsilon$, SVGD requires $T = \tilde{O}(\frac{\mathsf{poly}(d)}{\epsilon^2})$ and $n = \tilde{O}(\exp(\Theta(de^{\frac{\mathsf{poly}(d)}{\epsilon^2}})))$. Thus the oracle complexity of SVGD (as implied by Shi and Mackey [43]) for achieving $\mathsf{KSD}_{\pi^\star}(\hat\mu^{(n)}_{\mathsf{SVGD}}||\pi^\star)$ is $\tilde{O}(\frac{\mathsf{poly}(d)}{\epsilon^2} \cdot \exp(\Theta(de^{\frac{\mathsf{poly}(d)}{\epsilon^2}})))$

#### E.3.2 VP-SVGD

From Algorithm 1, we note that $T$ steps of VP-SVGD run with $n$ particles and a batch-size of $K$ requires $K^2T^2 + KTn$ evaluations of $\nabla F$.

**Subgaussian $\pi^\star$** For subgaussian $\pi^\star$, Corollary 2 implies a finite-particle convergence rate of $\mathbb{E}[\mathsf{KSD}^2_{\pi^\star}(\hat\mu^{(n)}||\pi^\star)] = O(\frac{d^{1/2}}{n^{1/2}} + \frac{d}{n})$ (where $\hat\mu^{(n)}$ denotes the empirical measure of the $n$ particles output by VP-SVGD) assuming $KT = d^{1/2}n^{3/2}$. Hence, to achieve $\mathbb{E}[\mathsf{KSD}_{\pi^\star}(\hat\mu^{(n)}||\pi^\star)] \leq \epsilon$, VP-SVGD requires $n = O(\frac{d}{\epsilon^4})$ and $KT = d^{1/2}n^{3/2} = \frac{d^2}{\epsilon^6}$. The resulting oracle complexity for achieving

$\mathbb{E}[\mathsf{KSD}_{\pi^\star}(\hat{\mu}^{(n)}||\pi^\star)] \leq \epsilon$ is $O(\frac{d^4}{\epsilon^{12}})$. Compared to the oracle complexity of SVGD obtained above, this is a *double exponential improvement in both $d$ and $1/\epsilon$*. Notably, the obtained oracle complexity guarantee *completely eliminates the curse of dimensionality*.

**Subexponential $\pi^\star$**  For subexponential $\pi^\star$, Corollary 2 implies a finite-particle convergence rate of $\mathbb{E}[\mathsf{KSD}_{\pi^\star}^2(\hat{\mu}^{(n)}||\pi^\star)] = O(\frac{d^{2/3}}{n^{1/3}} + \frac{d^2}{n})$ (where $\hat{\mu}^{(n)}$ denotes the empirical measure of the $n$ particles output by VP-SVGD) assuming $KT = d^{1/3}n^{4/3}$. Hence, to achieve $\mathbb{E}[\mathsf{KSD}_{\pi^\star}(\hat{\mu}^{(n)}||\pi^\star)] \leq \epsilon$, VP-SVGD requires $n = O(\frac{d^2}{\epsilon^6})$ and $KT = d^{1/3}n^{4/3} = \frac{d^3}{\epsilon^8}$. The resulting oracle complexity for achieving $\mathbb{E}[\mathsf{KSD}_{\pi^\star}(\hat{\mu}^{(n)}||\pi^\star)] \leq \epsilon$ is $O(\frac{d^6}{\epsilon^{16}})$.

### E.3.3  GB-SVGD

From Algorithm 2, we note that $T$ steps of GB-SVGD run with $n$ particles and a batch-size of $K$ requires $KTn$ evaluations of $\nabla F$.

**Subgaussian $\pi^\star$**  For subgaussian $\pi^\star$, Corollary 3 implies a finite-particle convergence rate of $\mathbb{E}[\mathsf{KSD}_{\pi^\star}^2(\hat{\nu}^{(n)}||\pi^\star)] = O(\frac{d^{2/3}}{n^{1/6}} + \frac{d}{n})$ (where $\hat{\nu}^{(n)}$ denotes the empirical measure of the $n$ particles output by GB-SVGD) assuming $KT = n^{1/2}$. Hence, to achieve $\mathbb{E}[\mathsf{KSD}_{\pi^\star}(\hat{\nu}^{(n)}||\pi^\star)] \leq \epsilon$, GB-SVGD requires $n = \frac{d^4}{\epsilon^{12}}$ and $KT = \sqrt{n} = \frac{d^2}{\epsilon^6}$. Under this setting, the oracle complexity of GB-SVGD as implied by Corollary 3 is $O(\frac{d^6}{\epsilon^{18}})$. Compared to the oracle complexity of SVGD obtained above, this is *a double exponential improvement in both $d$ and $1/\epsilon$*. Notably, the obtained oracle complexity guarantee *completely eliminates the curse of dimensionality*

**Subexponential $\pi^\star$**  For subexponential $\pi^\star$, Corollary 3 implies a finite-particle convergence rate of $\mathbb{E}[\mathsf{KSD}_{\pi^\star}^2(\hat{\nu}^{(n)}||\pi^\star)] = O(\frac{d^{3/4}}{n^{1/8}} + \frac{d^2}{n})$ (where $\hat{\nu}^{(n)}$ denotes the empirical measure of the $n$ particles output by GB-SVGD) assuming $KT = n^{1/2}$. Hence, to achieve $\mathbb{E}[\mathsf{KSD}_{\pi^\star}(\hat{\nu}^{(n)}||\pi^\star)] \leq \epsilon$, GB-SVGD requires $n = \frac{d^6}{\epsilon^{16}}$ and $KT = \sqrt{n} = \frac{d^3}{\epsilon^8}$. Under this setting, the oracle complexity of GB-SVGD as implied by Corollary 3 is $O(\frac{d^9}{\epsilon^{24}})$.

## F  Literature Review

Initial works on the analysis of SVGD such as Liu [30], Lu et al. [35], Duncan et al. [14], Chewi et al. [7], Nüsken and Renger [38] consider the continuous-time population limit, i.e., the limit of infinite particles and vanishing step-sizes. In this regime, Liu [30], Lu et al. [35], Nüsken and Renger [38] show that the behavior of SVGD is characterized by a Partial Differential Equation (PDE), and established asymptotic convergence of this PDE to the target distribution. The work of Duncan et al. [14] proposes the Stein Logarithmic Sobolev Inequality which ensures exponential convergence of this PDE to the target distribution. However, characterizing the conditions under which this inequality holds is an open problem. The work of Chewi et al. [7] show that the PDE governing SVGD in the continuous-time population limit can be interpreted as an approximate Wasserstein gradient flow of the Chi-squared divergence. To this end, Chewi et al. [7] shows that the (exact) Wasserstein gradient flow of the Chi-squared divergence exhibits exponential convergence to the target distribution when $\pi^\star$ satisfies a Poincare Inequality. To the best of our knowledge, the first discrete-time non-asymptotic convergence result for population-limit SVGD was established in Korba et al. [26], where the authors interpreted population-limit SVGD as projected Wasserstein gradient descent. Their result relied on the assumption that the Kernel Stein Discrepancy to the target is uniformly bounded along the trajectory of SVGD, a condition which is hard to verify apriori. This result was significantly improved in Salim et al. [42], which established convergence of population-limit SVGD assuming the potential $F$ is smooth the target $\pi^\star \propto e^{-F}$ satisfies Talagrand's inequality $\mathsf{T}_1$, an assumption which is equivalent to subgaussianity of $\pi^\star$. This result was extended in Sun et al. [45] to accommodate for potentials $F$ that satisfy a more general smoothness condition.

In comparison to prior works on population-limit SVGD, the literature on finite-particle SVGD is relatively sparse. The works of Liu [30] and Gorham et al. [18] establish that the dynamics of finite-particle SVGD asymptotically converge to that of population-limit SVGD in bounded Lipschitz

distance and Wasserstein-1 distance respectively, as the number of particles approaches infinity. Under the stringent condition of bounded $F$ (which is violated in various scenarios, e.g. log-strongly concave $\pi^\star$), Korba et al. [26] derived a non-asymptotic bound between the expected Wasserstein-2 distance between finite-particle SVGD and population-limit SVGD. The work of Liu et al. [33] considers the special case of Gaussian initialization and bilinear kernels, and provides a finite-particle convergence guarantee for the first and second moments whenever the target density is also a Gaussian. To the best of our knowledge, Shi and Mackey [43] is the only prior work that explicitly establishes a complete non-asymptotic convergence guarantee of finite-particle SVGD for subgaussian targets, and shows that the empirical measure of SVGD run with $n$ particles converges to the target density in KSD at a rate of $O\left(\sqrt{\frac{\mathrm{poly}(d)}{\log\log n^{\Theta(1/d)}}}\right)$.

# G  Additional Experimental Details

As discussed in Section 8, we benchmark SVGD and GB-SVGD on the task of Bayesian Logistic Regression using the Covertype dataset, obtained from the UCI Machine Learning Repository, whch contains around $580,000$ datapoints with dimension $d = 54$. We use $n = 100$ particles for both SVGD and GB-SVGD, and use batch-size $K = 40$ for GB-SVGD. For both SVGD and GB-SVGD, we use AdaGrad with momentum to set step-sizes as per the procedure used by Liu and Wang [31]. Our experiments were performed using Python 3 on a 2.20 GHz Intel Xeon CPU with 13 GB of memory.