# OpenReview forum: "Provably Fast Finite Particle Variants of SVGD via Virtual Particle Stochastic Approximation"
_NeurIPS.cc/2023/Conference — NeurIPS 2023 spotlight_

### Official Review · Reviewer_ArA7 · 2023-07-04

**Soundness:** 2 fair
**Presentation:** 2 fair
**Contribution:** 3 good
**Rating:** 5
**Confidence:** 3

**Summary:**

The authors propose a implementable stochastic approximation algorithm based on Stein Variational Gradient Descent, which the idea is based on a stochastic approximation at the probability measure level.  The efficiency and convergence analysis of the proposed algorithm, i.e. VP-SVGD and GB-SVGD,  are presented. A better convergence rate for the proposed algorithm is proved given a weaker assumption for the potential function

**Strengths:**

The authors propose a new algorithm, a stochastic approximation at the probability measure level according to the authors, to study the SVGD. The focus is to improve the convergence rate and make the algorithm more efficient. The idea of the proposed algorithm in (3) seems to be new. The intuition and design of the algorithm is interesting, and it seems to work as shown in the numerical experiments.

**Weaknesses:**

The reviewer feels that the intuition and the theoretical results have some gaps. To be more precise, some of the arguments are only explained at the intuition level without rigorous proof. It might be possible the reviewer missed the key point of the proof, if this is the case, please correct me.

**Questions:**

1. What is exactly the mild assumption for potential function F, which is weaker then Poincare and Talagrand inequality? How significantly different for this condition, e.g. regarding the proof ?

2. Section 3 is not well presented. The push-forward map $(\cdot)_{\sharp}$ is not defined, but only mentioned, which is not clear. But this is a key step in designing the algorithm.

3. Section 4, the notation for filtration is not good, $\mathcal F_t$ should be measurable up to time $t$, but here it is measurable up to time $t-1$, which results the notation $\mu_t|\mathcal F_t$ confusing.

4. Up this this point,  updates in (2) is not rigorously derived, but only intuitively mentioned. Then the algorithm (3) is based on (2), which is not clear to me why (3) is a stochastic approximation? It seems to be deterministic given the previous time step particle values. The same questions for $\mu_t|\mathcal F_t$, the conditional measure based on updated in (3) does not seem to be random. If there is a better way to present this idea, please clarify.

5. The random function $g_t(\mathbf X)$ is not clear, where does the randomness come?

6. the mini batch size comes in Algorithm 1. which does not show before in section 4. There is a gap between the Algorithm 1 & 2  and (2)(3)(4).

7. What is exactly the role of countably infinite many particles, as in the algorithm only $KT+n-1$ are used? I seems that this assumption is important, if this can be explained more clear, it should help improving the presentation.

**Limitations:**

see above

---

> ### Author Rebuttal · Authors · 2023-08-09
>
> $\newcommand{\vxl}[1]{\vx^{(l)}\_{#1}} \newcommand{\s}[1]{\\{0, \dots, #1\\}} \newcommand{\muFt}{\mu\_t | \cF\_t} \newcommand{\xtt}{\vx^{(t)}\_t} \newcommand{\XS}[1]{\vx^{(0)}\_0, \dots, \vx^{(#1)}\_0}$Thanks for your helpful feedback! The reviewer mentions that some of our proofs are not rigorous/complete. We believe all our results are proved fully rigorously, with many proofs shown in the Appendix (as is common for NeurIPS papers) due to page limits. We are happy to discuss any specific proofs which the reviewer believes are not rigorous. We hope our response below addresses your questions
> ## Assumption on $F$
> Assumption 2 imposes a mild growth condition on $F$. As stated in Sec. 5 (Lines 217-223) & proved in Appendix B (Lines 526-573) , Assmp. 2 is implied by Talagrand's inequality $T\_1$ (used by prior works on SVGD [1,2]), as well as Poincare inequality (considered a mild condition in sampling [3]). We use Assmp. 2 to bound $\\|g_t\\|\_{\mathcal{H}}$ (Lemma 2) which is essential for applying the descent lemma for VP-SVGD (Lemma 1)
> ## Defn. of Pushforward
> It is defined in Sec. 2, Lines 126-127
> ## Batchsize in VP-SVGD
> For ease of exposition, we motivate VP-SVGD in Sec. 4 using batchsize $K=1$ (see Line 178). Our analysis of VP-SVGD in Appendix C presents a complete proof for any $K\geq1$ (Appendix C.1)
> ## Derivation of Eqn. 2
> Eqn. 2 refers to the dynamics of ordinary SVGD in the mean-field (i.e. infinite particle) limit, which has been well established in several prior works [1,4,5]. Due to page limits & the fact that mean-field SVGD is the key object of study in many prior works, we provide a brief exposition of Eqn. 2 in Sec. 3, where we also refer the readers to prior works [1,5] for a more detailed treatment
> ## Notation for Filtration
> We use the comp. sci. convention of starting indices from zero. Thus, the particles are indexed as $\vxl{t};l,t\geq0$. Thus, for $K=1$ we denote the filtration $\cF_t$ as $\cF_t = \sigma(\\{\vxl{0}| l \in \s{t-1} \\})$.
> ## $g_t$ is a Random Function
> From Sec. 4, Line 190, we infer that $g_t(\vx) = h(\vx, \xtt)$ is a random vector $\forall \vx \in \mathbb{R}^d$ since $\xtt$ is a random vector. Thus, $g_t$ is a random function due to the randomness induced by $\xtt$
> ## $\muFt$ is a Random Measure
> For $K=1$, $\muFt$ is the conditional distribution of the random vector $\xtt$ when conditioned on the virtual particles $\XS{t-1}$, i.e., $\muFt = Law(\xtt | \XS{t-1})$ (Sec. 4 Lines 186-189). Since $\muFt$ is a conditional distribution, it can be expressed as a probability kernel $\muFt = \mu\_t(.;\XS{t-1})$ (see [6, Sec. 4.1.3]). Since $\XS{t-1}$ are random vectors, $\mu_t | \cF_t := \mu_t(\cdot ; \mathbf{x}^{(t)}_0, \dots, \mathbf{x}^{(t-1)}_0)$ is a random probability distribution, where the randomness comes from $\XS{t-1}$
> ## VP-SVGD as a Stochastic Approx. to Mean-Field SVGD
> VP-SVGD is an unbiased stochastic approximation to mean-field SVGD **in the space of probability measures**. This differs significantly from the path-space stochastic approximations usually considered in the literature (see Sec. 1.2 Lines 99-109). We briefly describe this as follows (see Section 4, Appendix C & our common response for details). From (3), we note that $\vxl{t+1} = (I - \gamma g_t)(\vxl{t}) \ \forall l\geq t+1$ & $g_t(\vx) = h(\vx, \xtt)$ is an $\cF\_{t+1}$ measurable random function (as $\xtt$ is $\cF\_{t+1}$ measurable). Now, $\forall t \geq0,l\geq t;\vxl{t}$ depends only on $\XS{t-1}, \vxl{0}$ and $Law(\xtt |\XS{t-1}) = \muFt$. Thus, $\forall l\geq t+1,Law(\vxl{t}|\XS{t})=Law(\vxl{t}|\XS{t-1})=\muFt$ and $Law(\vxl{t+1}|\XS{t})=\mu\_{t+1} | \cF\_{t+1}$. Hence, The dynamics of VP-SVGD in the space of measures is $\mu\_{t+1} | \cF_{t+1} = (I - \gamma g_t)\_{\\#} \muFt$. Finally, note that $\bE[g_t(\vx)|\cF_t] = h_{\muFt}(\vx)$ which is the kernelized Wasserstein gradient of $\KL{.}{\pi^*}$. Comparing with the mean-field SVGD update $\mub\_{t+1} = (I - \gamma h_{\mub\_t})\_{\\#} \mub_t$, we conclude that VP-SVGD is indeed an unbiased stochastic approximation to mean-field SVGD in the space of measures
> ## Countably Infinite Particles
> Our analysis of VP-SVGD considers the dynamics of countably infinite particles, whose initial positions $(\vxl{0})\_{l\geq0}$ are drawn i.i.d. from an initial density $\mu_0$. **This is done only for the purposes of our analysis** (as Algo. 1 needs only $KT+n$ particles to converge) **i.e., the particles $(\vxl{0})_{l\geq n+KT}$ are fictitious and are not used anywhere except for our proofs**. The advantages of this approach is described in Section 1.2 Lines 110-120, which we summarize below:
>
> As discussed in Sec. 3, mean-field SVGD is analyzed by tracking the KL divergence, i.e., $\KL{\mub_t}{\pi^*}$. This approach doesn't apply to finite particle SVGD since the empirical measure $\hmu_t$ of $n$ particles does not admit a density w.r.t the Lebesgue measure, i.e., $\KL{\hmu_t}{\pi^*}=\infty$. To overcome this, we note that the empirical measure of countably infinitely many i.i.d random vectors is almost surely equal to it's population measure by Strong Law of Large Numbers [6]. Thus, in each timestep $t$, we consider countably infinitely many conditionally i.i.d. samples from $\muFt$. By choosing $\mu_0$ such that $\KL{\mu_0}{\pi^*}<\infty$, we ensure $\muFt$ admits a density at each step using Lemmas 1 & 2, which allows us to track $\KL{\muFt}{\pi^*}$. Our algorithm design ensures that VP-SVGD needs only $KT+n$ particles to output $n$ conditionally i.i.d samples from $\muFt$. *We contrast this to mean-field SVGD which actually requires infinite particles for exact implementation, and thus, is not practically realizable*.
>
> 1. Shi et.al.'22, Finite-Particle Rate for SVGD
> 2. Salim et. al., Convergence Theory for SVGD Under Talagrand's Inequality T1
> 3. Chewi et.al.'21 LMC from Poincare to LSI
> 4. Liu'17, SVGD as Gradient Flow
> 5. Korba et.al.'20,  Non-Asymptotic Analysis of SVGD
> 5. Durrett'05, Probability Theory and Examples

---

> > ### Comment · Reviewer_ArA7 · 2023-08-13
> >
> > Thanks for your response. I have upgraded my rate for this paper.

---

> > > ### Author Response · Authors · 2023-08-17
> > > **Thank You!**
> > >
> > > We thank the reviewer for their prompt and helpful response.

---

### Official Review · Reviewer_UYmX · 2023-07-04

**Soundness:** 4 excellent
**Presentation:** 2 fair
**Contribution:** 3 good
**Rating:** 5
**Confidence:** 4

**Summary:**

This study analyzed the finite particle behavior of Stein Variational Gradient Descent (SVGD) in an asymptotic manner. For that purpose, the authors designed two computationally efficient variants of SVGD, namely VP-SVGD and GB-SVGD, and prove their convergence rates. In those methods, the authors introduced a new concept of "virtual particles" and develop a new stochastic approximation of the population-limited SVGD dynamics in the probability measure space. The proposed algorithm shows a significant improvement of the convergence rates under  looser conditions than the usual SVGD.

**Strengths:**

- The convergence rates in this paper are significantly better than existing methods.

- The idea of virtual particles introduced for the analysis is very novel and very interesting in itself.

- Numerical experiments, although simple,  suggest that the proposed method is indeed an improvement over existing methods.

**Weaknesses:**

- Notation in Sec 4 is very confusing, especially about $l, s, t$

-  In the explanation of Sec 4, only the case of K=1 is clearly stated, and I had to spend a lot of time thinking about what happens when K>2. I think an additional explanation of K > 2 is needed in the Appendix.

- The algorithm of GB-SVGD is difficult to understand. In particular, the third line mentions "l", but "l" does not appear at all in the fourth line. This may be because the "l" in the third line may be a mistake for "s".

- I am not sure if the theoretical comparison of the convergence order with the existing analysis is justified since the VP-SVGD is no longer the original SVGD. The GB-SVGD is close to the existing SVGD, but it is also difficult to compare with the existing SVGD because of the approximation errors as shown in Theorem 2.

**Questions:**

- What is $\nabla_2$ in Line 143 ? Partial derivative with respect to the second variable?

- Is my following understanding correct? Intuitively, the motivation for introducing virtual particles is to ensure that the resulting particles are i.i.d, which is required to use Lemma 7 in the theoretical analysis.

- Algorithm 1 requires a larger number of particles in proportion to the time step T than ordinary SVGD, and therefore the number of gradient calculations increases considerably. To achive $KSD(particles,\pi^*)\leq \epsilon$, does VP-SVGD increase the order of magnitude of the number of the gradient calculation compared to standard SVGD?

- I was wondering why S is taken randomly in Algorithm 1 line 6 and Algorithm 2 line 7, etc., and I thought it would greatly reduce the practical performance. But after reading the proof, I understand that this is necessary to convert the bounds of $\frac{1}{T}$KSD^2(\mu_t|\pi^*)$ to KSD(\bar{\mu}|\pi^*}) as shown in Theorem 1. Is this understanding correct?

- In the numerical experiment, are random $S$ used as written in TAlgorithm 1 and 2? Since the objective function is convex, such random S may not have such a strong effect on the final performance. I wonder how harmful this random $S$ is when the objective distribution is non-log concave?




**Limitations:**

Theoretical limits are discussed in detail.

---

> ### Author Rebuttal · Authors · 2023-08-09
>
> $\newcommand{\e}{\epsilon} \newcommand{\gd}{\nabla} \newcommand{\KSD}[3]{KSD\_{#1}(#2||#3)}$ Thank you for your constructive feedback. We hope to address your questions in our response below:
> ## Notation in Sec. 4
> Thanks for the suggestion! This is incorporated in our updated draft where we consistently use $\vx^{(l)}\_t;l, t\geq0$ to denote the particles. As before, our indices start from $0$ (as is common in comp. sci. literature).
>
> ## $l,s$ in GB-SVGD
> Thanks for the pointer! $l$ in the third line of Algorithm 2 should be $s$. This typo is fixed in our updated draft.
>
> ## Batchsize $K>2$ in VP-SVGD
> For ease of exposition, the discussion in Sec. 4 considers $K=1$. Our analysis of VP-SVGD in Appendix C presents a complete proof for $K \geq 1$ (see Appendix C.1)
>
> ## $\gd\_2$ in Line 143
> Yes, it refers to the derivative w.r.t second variable. This is mentioned in Section 2 Lines 138-139.
>
> ## Oracle Complexity of VP-SVGD & SVGD
>  We compare the gradient oracle complexity of VP-SVGD & ordinary SVGD as implied by [1], which to our knowledge, is the best known finite-particle rate for SVGD. Note that $T$ iterations of $n$-particle SVGD needs $n^2 T$ evaluations of $\gd F$. Carefully following the analysis of [1] shows that the oracle complexity of SVGD for $KSD\leq\e$ is $\tilde{O}(\frac{poly(d)}{\e^2}\exp(\Theta(d\exp(\tfrac{poly(d)}{\e^2}))))$
>
> VP-SVGD run for $T$ iterations with batch-size $K$ & $n$ real particles requires $K^2 T^2 + KTn$ evaluations of $\gd F$. From Corollary 1, we find that the oracle complexity of VP-SVGD for $KSD \leq \epsilon$ is $\tilde{O}(\exp(\Theta(d \log(\tfrac{d^2}{\e}))))$. This is a **double exponential improvement** over SVGD. Note that GB-SVGD (which needs $KTn$ evaluations of $\nabla F$) also enjoys this improved gradient complexity. Thus, we conclude that **VP-SVGD requires far fewer gradient evaluations than SVGD** due to its superior convergence rate. Our updated draft shows a complete derivation of the oracle complexity of SVGD, VP-SVGD & GB-SVGD, where we also show that **our improved analysis of the finite-particle convergence of VP-SVGD & GB-SVGD** (see common response) **leads to further improvements in oracle complexity.**
>
> ## Comparing Ordinary SVGD & VP-SVGD
> We respectfully disagree with the reviewer's claim that ordinary SVGD & VP-SVGD are not comparable. We refer the reviewer to our common response where we elucidate that **both ordinary SVGD & VP-SVGD are finite-particle approximations of the same mean-field dynamics**. Thus we firmly believe that the two algorithms can be justifiably compared.
>
> ## Motivation Behind Virtual Particles
> We note that Lemma 7 (i.e. the descent lemma of [6]) holds without any conditional independence requirement. As such, we do not use the conditionally i.i.d structure of VP-SVGD outputs to apply Lemma 7. We refer the reviewer to our common response where we explain the key ideas behind the use of virtual particles in VP-SVGD. We note that the **conditionally i.i.d structure is used for applying the empirical-to-population Wasserstein bound of Lemma 8**, which is a key step in our proof of the finite-particle rate for VP-SVGD (Corollary 2)
>
> ## Comparing ordinary SVGD and GB-SVGD
> We note that **GB-SVGD is a random batch approximation of ordinary SVGD** (see Section 1.2 Lines 87-96). Thus, we firmly believe that **they can be meaningfully compared just like SGD is compared to GD in optimization [2] & SGLD is compared to LMC in sampling [3,4]**. Moreover, Theorem 2 does not say anything about the approximation error of GB-SVGD w.r.t ordinary SVGD, but instead is a Wasserstein-1 comparison bound between the empirical measures of the outputs of GB-SVGD and VP-SVGD (used to prove Corollary 3)
>
> ## Use of Uniform Random $S$
> Yes, using $S \sim Unif(\\{0, \dots, T-1\\})$ allows us to convert the average bound $\tfrac{1}{T} \sum\_{t=0}^{T-1} \bE[\KSDs{\pi^*}{\mu\_t | \mathcal{F}\_t}{\pi^*}]$ into a bound in terms of $\bE[\KSDs{\pi^*}{\bar{\mu}}{\pi^*}]$. This ensures that the particles output by VP-SVGD are (conditionally i.i.d) samples from a distribution which is provably close to $\pi^*$ in KSD. Such average bounds are commonplace in prior works on SVGD [1,5,6], sampling in general [4,7], and optimization [2,8].
>
> We note that obtaining last iterate guarantees, even in optimization, involves sophisticated techniques & algorithmic modifications [9,10]. We believe achieving this in the context of SVGD-type sampling algorithms (using only tail decay assumptions on $\pi^*$) is an important future direction.
>
> ## $S$ in Experiments
> For all algorithms considered, we benchmark the performance of the current iterate against iterations/wall clock time. We note that this is standard practice in stochastic optimization [9,10,11]. Our updated draft states this explicitly.
>
> Regarding the performance w.r.t non-log-concave $\pi^*$: Based on our intuition from stochastic convex optimization, we do not believe that such an averaging will theoretically affect the performance since: 1. $\mu \to \KL{\mu}{\pi^*}$ & $\mu \to \KSD{\pi^*}{\mu}{\pi^*}$ are both convex functionals of probability distributions, 2. The algorithm can be seen as a projected/kernelized stochastic gradient descent of $\KL{\mu}{\pi^*}$ in the space of probability distributions. Further investigation is necessary to answer these questions precisely.
>
> 1. Shi et.al.'22, A Finite Particle Rate for SVGD
> 2. Hazan'19, Optimization for ML
> 3. Welling et.al.'11, Bayesian Learning via SGLD
> 4. Das et.al.'23, Utilising the CLT Structure in SGLD
> 5. Korba et.al.'20, A Nonasymptotic Analysis of SVGD
> 6. Salim et.al.'21, Convergence Theory for SVGD in Pop. Limit
> 7. Balasubramanian et.al.'22, Towards a Theory of Non-Logconcave Sampling
> 8. Polyak et.al.'92, Acceleration of Stochastic Approx. by Averaging
> 9. Shamir'12, Open Problem: Is Averaging Needed for Str. Cvx. SGD?
> 10. Rakhlin et.al.'12, Making GD Optimal for Str. Cvx. Stochastic Opti.
> 11. Shalev-Shwartz et.al.'11, Pegasos

---

### Official Review · Reviewer_T9Sw · 2023-07-08

**Soundness:** 4 excellent
**Presentation:** 3 good
**Contribution:** 4 excellent
**Rating:** 8
**Confidence:** 4

**Summary:**

The paper studies Stein Variational Gradient Descent (SVGD) which is a popular variational inference algorithm that simulates an interacting particle system to approximately sample from a target distribution. The authors propose two variants of SVGD, Virtual Particle SVGD (VP-SVGD) and Global Batch SVGD (GB-SVGD), that differ in their implementation details and computational efficiency. Both variants use the so-called "virtual particles", which are additional particles that evolve in time but are not part of the output, to compute information about the current population-level distribution of the real particles and enable exact implementation of the stochastic approximation using only a finite number of particles. The use of virtual particles allows for faster variants of SVGD with provably fast finite-particle convergence.

**Strengths:**

This paper introduced two variants of SVGD, VP-SVGD and GB-SVGD, that use virtual particles to enable the exact implementation of the stochastic approximation using only a finite number of particles. The authors present a non-asymptotic analysis of the convergence of VP-SVGD and GB-SVGD to the target distribution, which demonstrates a double exponential improvement over the best-known finite-particle analysis of SVGD. The paper proposes a new algorithm, VP-SVGD, that achieves provably fast finite-particle convergence and is computationally more efficient than ordinary SVGD.

The paper is well-organized and clearly written, with detailed explanations of key concepts and technical details that make it accessible to a wide range of readers. Literature review of related work on SVGD and its variants was conducted comprehensively, which helps to contextualize the proposed algorithms and their contributions.

I went through the proofs of Theorem 1 and Corollary 1 or 2 in Section E in detail, and believe they are technically sound. The KSD metric satisfies the triangle inequality where the upper bound is in two additional parts. In (34), the KSD metric satisfies a triangle inequality, and the first term (probability kernel vs population limit) is polynomial with a polynomial decaying exponent that is independent of dimension $d$ ($d^{1/3}T^{-1/6}$ when subgaussian). The second term appears dominating, where the polynomial decaying exponent in Wasserstein-1 distance is $\Theta(1/d)$, which is due to Lemma 8 by Lei [29].

**Weaknesses:**

Despite its significant contributions, this work assumes a certain level of familiarity with the concepts of variational inference and particle methods and is relatively specialized and may make it difficult for readers who are not already familiar with or not working on the topic of SVGD and its variants to follow.  Despite the technical contributions of proposed SVGD-style algorithms and their convergence properties, the authors did not provide a detailed discussion of the limitations of the proposed algorithms or potential avenues for future research, which may limit their impact and relevance in the long term it and make it less accessible to readers who are more interested in the broader implications of the work. Lastly, the work does not provide sufficient details on the proposed algorithms' empirical evaluations of real-world datasets, which limits the ability to assess their practical usefulness. These weaknesses did affect my current rating, but I reserve the possibility to raise the score if any of the above points can be resolved in the authors' rebuttal.

**Questions:**

I am not a leader in the topic of SVGD but had a chance to work on the topic at a senior educative level. Hence I recognize the importance of this topic. Some questions are as follows:

How do the VP-SVGD and GB-SVGD variants of SVGD differ from the original SVGD introduced by Liu and Wang [32]?
My understanding is that the key difference between VP-SVGD and GB-SVGD variants of SVGD is that VP-SVGD is a stochastic approximation of the exact SVGD dynamics, while GB-SVGD is a specific random-batch approximation of SVGD which enjoys computationally more efficient than ordinary SVGD. Is that correct?

Can you explain more about the concept of virtual particles and how they are used in these algorithms? I understand that virtual particles are additional particles that evolve in time but are not part of the output (i.e., real particles) which are used only to compute information about the current population-level distribution of the real particles. Are they absolutely necessary --- are there any equivalent or similar implementations that do not necessarily require the introduction of virtual particles?

In the technical analysis of Corollary 2, is there any hope to make the decaying exponent in polynomial rate dimension-free or have a weaker dependence on dimension (perhaps using an alternative metric or developing new analytic tools to yield an improved bound)?

**Limitations:**

The work is purely theoretical and admits no negative social impacts.

---

> ### Author Rebuttal · Authors · 2023-08-09
>
> $ \newcommand{\iidsim}{{\overset{\mathrm{iid}}{\sim}}} \newcommand{\cH}{\mathcal{H}} \newcommand{\hmu}{\hat{\mu}^{(n)}}$
> Thank you for your thoughtful review and suggestions, which have significantly helped us improve our work. We have updated our draft to include a detailed discussion of the limitations and broader applications of our results which could be interesting avenues of future work (some of which are discussed below). We also present a more elaborate comparison of our results to that of prior work. Since our work is primarily theoretical, we perform our experiments on a few of the simple (yet high-dimensional) benchmarks considered in prior work [2] to illustrate the efficacy of our algorithm. Empirical evaluation of our algorithms on sophisticated architectures and datasets is an important avenue of future work which we intend to pursue. We hope our response below addresses your questions.
>
> ## Dimension-Free Exponent in Corollary 2 & 3
> Thank you for the pointer! Based on your suggestion, we remove the dimension dependence in the $n$ exponent of Corollary 2 and 3 and show that for subgaussian $\pi^*$, VP-SVGD has a finite-particle KSD rate of $O(\tfrac{d^{1/4}}{n^{1/4}} + \tfrac{d^{1/2}}{n^{1/2}})$ while GB-SVGD has a finite-particle KSD rate of $O(\tfrac{d^{1/3}}{n^{1/12}} + \tfrac{d^{1/2}}{n^{1/2}})$ (the constant factors in the exponents may be further improved via careful analysis). We briefly discuss this result in our common response and present a full proof in our updated draft. A proof sketch for VP-SVGD is presented below:
>
> Recall that the outputs of VP-SVGD are $(\vx^{(l)}\_S)_{KT \leq l \leq KT+n-1}$ where $S \sim Unif(\\{0, \dots, T-1 \\})$. Let $\hmu$ be their empirical measure. By defn of KSD, $\KSDs{\pi^*}{\hmu}{\pi^*} = \\|h\_{\hmu}\\|^2\_\cH = \\| \tfrac{1}{n} \sum\_{l=0}^{n-1} h(., \vx^{(KT+l)}\_S) \\|^{2}\_{\cH}$. Let $V$ denote the conditioning on the virtual particles $(\vx^{(l)}\_0)\_{l < KT}$. Conditioning on $V$ and the event $S=t$, we get:
> $$\bE[\KSDs{\pi^*}{\hmu}{\pi^*}|S=t,V] = \bE[\\|\tfrac{1}{n} \sum\_{l=0}^{n-1} h(., \vx^{(KT+l)}\_t)\\|^{2}\_{\cH} |V] = \bE[\\|\tfrac{1}{n} \sum\_{l=0}^{n-1} h(., \vx^{(KT+l)}\_t)\\|^{2}\_{\cH} |(\vx^{(l)}\_{0})\_{l < KT}]$$
> where we use Equation 14 from Appendix C.1 to note that $\vx^{(KT+l)}_t$ depends only on $(\vx^{(s)}\_0)\_{s < Kt}$ & $\vx^{(KT+l)}\_0$. Recall that $$\bE[\\|\tfrac{1}{n} \sum\_{l=0}^{n-1} h(., \vx^{(KT+l)}\_t)\\|^{2}\_{\cH} |(\vx^{(l)}\_{0})\_{l < KT}] = \bE[\\|\tfrac{1}{n} \sum\_{l=0}^{n-1} h(., \vx^{(KT+l)}\_t)\\|^{2}\_{\cH} |\cF\_t]$$
> Note that $(\vx^{(l)}\_t)\_{KT \leq l \leq KT+n-1}$ are conditionally i.i.d samples from $\mu_t | \cF_t$. Applying the same arguments as Lemma 3, we get:
> $$\bE[\\|\tfrac{1}{n} \sum\_{l=0}^{n-1} h(., \vx^{(KT+l)}\_t)\\|^{2}\_{\cH} |\cF\_t] \leq \tfrac{\xi^2}{n} + \\| h\_{\mu_t | \cF_t} \\|^{2}\_{\cH} = \tfrac{\xi^2}{n} + \KSDs{\pi^*}{\mu_t | \cF_t}{\pi^*}$$
> Taking expectations over $S$ and the virtual particles gives us:
> $$\bE[\KSDs{\pi^*}{\hmu}{\pi^*}] \leq \tfrac{\xi^2}{n} + \tfrac{1}{T} \sum\_{t=0}^{T-1} \bE[\KSDs{\pi^*}{\mu_t | \cF_t}{\pi^*}]$$
> We conclude by substituing the bound of Theorem 1 and setting $\eta$ & $KT$ appropriately.
>
> ## VP-SVGD and Concept of Virtual Particles
> We refer to the common response for a detailed explanation on the similarities and differences between SVGD & VP-SVGD, as well as the key concepts behind the use of virtual particles.
>
> ## GB-SVGD and the Necessity of Virtual Particles
>
> Yes, GB-SVGD is a specific random batch approximation of SVGD (discussed in Section 1.2 Lines 87-96) which is computationally more efficient than ordinary SVGD (SVGD needs $n^2 T$ evaluations of $\nabla F$ while GB-SVGD needs $nKT$ evaluations, and typically $K \ll n$). We show that because of the random batches used in every timestep, GB-SVGD behaves similar to VP-SVGD (see Lines 204-207 and 260-266), and thus can also be viewed as an approximation to mean-field (infinite-particle) SVGD. This insight helps us prove a fast finite-particle rate for GB-SVGD (Corollary 3) which is double-exponentially faster than the best known finite-particle rate for SVGD [1].
>
> We highlight that **GB-SVGD does not require any virtual particles in its implementation** (see Algorithm 2 and Lines 205-207) and the comparison to VP-SVGD (Theorem 2) is only for the sake of mathematical analysis. In fact, Corollary 3 GB-SVGD needs only $n$ input particles to output $n$ particles whose empirical measure rapidly converges to $\pi^*$ in KSD.
>
> ## Potential Future Directions
> A few interesting avenues of future work include: 1. Extensions of VP-SVGD and GB-SVGD for sampling from manifolds by considering kernelized Riemannian Wasserstein gradient flows [3]; 2. Learning-rate free variants of VP-SVGD and GB-SVGD by applying techniques from parameter-free online optimization [4]. Beyond this, we believe our virtual particle technique is of independent interest beyond sampling and can be applied to obtain provably converging finite-particle implementations of more sophisticated mean-field dynamics (such as Mean-Field Langevin Dynamics [5] and Wasserstein Fisher-Rao Gradient Flows [6])
>
> 1. Shi et.al.'22, A Finite Particle Convergence Rate for SVGD
> 2. Liu & Wang'16, SVGD
> 3. Ohta'09, Gradient flows on Wasserstein spaces over compact Alexandrov spaces
> 4. Cutkosky et.al.'18, Blackbox Reductions for Parameter-Free Online Learning in Banach Spaces
> 5. Chizat et.al.'22, Trajectory Inference via Mean-Field Langevin in Path Space
> 6. Chizat et.al.'18, An interpolating distance between Optimal Transport and Fisher–Rao metrics.

---

> > ### Comment · Reviewer_T9Sw · 2023-08-17
> >
> > I thank the authors for their detailed responses (which are far beyond my expectation). I am a bit surprised that the authors were able to provide a simple proof of polynomial rates with dimension-free decaying exponents --- and I feel pleased having asked this question in retrospect. In addition, I appreciate the authors' clarification that GB-SVGD require no "virtual particles in its implementation", in stark contrast with VP-SVGD. Seeing all these new efforts, I raised my score from 6 to 8.

---

> > > ### Author Response · Authors · 2023-08-17
> > > **Thank You!**
> > >
> > > We thank the reviewer for their considerate response and for their highly constructive suggestions, which have been very helpful for improving our work.

---

### Official Review · Reviewer_9R2A · 2023-07-19

**Soundness:** 4 excellent
**Presentation:** 3 good
**Contribution:** 4 excellent
**Rating:** 8
**Confidence:** 3

**Summary:**

This paper propose two variants of stein variational gradient descent (SVD) named as Virtual Particle SVGD (VP-SVGD) and Global Batch SGVD (GB-SVGD) with provably fast finite-particle convergence rates to a target distribution. Introducing the notion of virtual particles, the authors show that VP-SVGD is an exactly implementable finite-particle stochastic approximation of population-limit SVGD dynamics in the space of probability measures. Viewing VP-SVGD as a specific random-batch SVGD algorithm, the authors argue that  the output particles from GB-SVGD after T time steps with batch-size K are as good as sampling iid from a distribution whose Kernel Stein Discrepancy (KST) to a sub-gaussian target distribution is at most $O(d^{1/3}/(KT)^{1/6})$. Furthermore, under mild growth conditions, the authors show that the empirical distribution of the n-output particles from both VP-SVGD and GB-SVGD converges to the target in KSD at a rate of $O(d^{2}/(n)^{\Theta(1/d)})$. The authors note that this is a doubly exponential improvement over the state-of-the-art finite-particle analysis of SVGD under weaker assumptions. Finally, the authors provide numerical experiments to show fast convergence of GB-SVGD which is computationally more efficient than SVGD.








**Strengths:**

1)Originality: This paper is introduces two new variants of SVGD appealing to a novel construction involving countably many virtual particles.  Although the analysis involves infinitely many particles, a finite-number of particles exactly suffice for the actual algorithm VP-SVGD. This enables the authors to claim that VP-SVGD performs a finite-particle stochastic approximation of the population (infinite-particle) dynamics of SVGD over the space of probability distributions. Finally, they show VP-SVGD and the computationally efficient GB-SVGD behave very closely.

2)Quality: I think this paper is technically quite solid. The theoretical claims are all sound and the assumptions are reasonable.

3)Clarity: This paper is well written. It is easy to follow the main ideas including the problem statement, technical challenges, relevance to past works, and the contributions.

4)Significance: It is quite significant to improve the SOTA result doubly exponentially under weaker assumptions. Besides, the virtual particle method is worthwhile studying by itself ( I am not aware of a similar method discussed somewhere else).

**Weaknesses:**

It wasn't easy for me to parse some of the theoretical results, which can be expected as this paper is theory heavy.

A table of past works comparing the results in terms of assumptions, method, and the convergence rate might be a helpful addition.

**Questions:**

I suspect there are a few typos in the text.

1)bold x with varying subscript  in line 249 vs bold x with varying superscript in the equation after line 250. also the index runs to T but I would expect T-1.

2)bold x instead of bold y in line 224

3) In line 204, we are already given that $g_s(x) = ...$ and then in line 306 we are told that $g_s(x) \approx ...$, Is this a typo ?


**Limitations:**

Limitations and future works are discussed.

---

> ### Author Rebuttal · Authors · 2023-08-08
>
> $\newcommand{\cF}{\mathcal{F}} \newcommand{\vx}{\mathbf{x}} \newcommand{\mub}{\bar{\mu}} \newcommand{\KL}[2]{KL\left(#1||#2\right)} \newcommand{\bE}{\mathbb{E}} \newcommand{\KSDs}[3]{KSD^{2}\_{#1}(#2||#3)} \newcommand{\iidsim}{{\overset{\mathrm{iid}}{\sim}}}$ Thank you for your kind and thoughtful review and your helpful suggestions which we have incorporated in our updated draft. In particular, our updated draft presents a more detailed account of related work, including a table that compares our results to that of prior work.
>
> ## Line 224
> Thank you for pointing this out. Our updated draft corrects for this typo.
>
> ## Regarding $g_s(\vx) \approx \dots$
>
> We note that $g_s(\vx) = k(\mathbf{x}, \mathbf{x}^{(s)}_s) \nabla F(\mathbf{x}^{(s)}_s) - \nabla_2 k(\mathbf{x}, \mathbf{x}^{(s)}_s)$ as per Lines 204 and 305. In Line 307, we state that, *whenever* $\mathbf{x} \approx \mathbf{x}^{(s)}_s$, $g_s(\mathbf{x}) \approx k(\mathbf{x}, \mathbf{x}^{(s)}_s) \nabla F(\mathbf{x}) - \nabla_2 k(\mathbf{x}, \mathbf{x}^{(s)}_s)$ *is satisfied*. This holds because the smoothness assumption on the potential $F$ (Assumption 1) ensures that $\nabla F(\mathbf{x}) \approx \nabla F(\mathbf{x}^{(s)}_s)$ whenever $\mathbf{x} \approx \mathbf{x}^{(s)}_s$ (note that since $\\| \nabla F(\mathbf{x}) - \nabla F(\mathbf{x}^{(s)}_s)\\| \leq L \\| \mathbf{x} - \mathbf{x}^{(s)}_s \\|$, $\\| \nabla F(\mathbf{x}) - \nabla F(\mathbf{x}^{(s)}_s)\\|$ is small whenever $\\| \mathbf{x} - \mathbf{x}^{(s)}_s \\|$ is small)
>
> ## Equation after Line 250
>
> Thank you for the pointer! For arbitrary $K \geq 1$, the indexing should indeed be $\bE[\KSDs{\pi^*}{\mub(\cdot; \vx^{(0)}\_{0}, \dots, \vx^{(KT-1)}\_{0})}{\pi^*}]$. We have corrected this in our updated draft. To the best of our understanding, the use of bold $\vx$ with varying superscript is not a typo. To observe this, we note that $\mub$ is defined as $\mub(\cdot;x\_0, \dots, x\_{KT-1}, s) := \mu\_s(\cdot;\vx^{(0)}\_{0} = x\_0, \dots, \vx^{(Ks-1)}\_{0} = x\_{Ks-1}) = Law(\vx^{(Ks)}\_s | (\vx^{(l)}\_0)\_{l < Ks} = (x\_{0}^{(l)})\_{l < Ks})$ (by convention, $\mub(\cdot;x\_0, \dots, x\_{KT-1}, s=0) := \mu\_0(\cdot)$). In particular, $\mub(\cdot; x\_0, \dots, x\_{KT-1}, s)$ gives us a distribution corresponding to any value $(x\_{0}^{(l)})\_{l < KT} \in \mathbb{R}^{d\times KT},s \in \\{ 0, \dots, T \\}$ . Thus, we interpret $\mub(\cdot ; \vx^{(0)}\_{0}, \dots, \vx^{(KT-1)}\_{0},S)$ as a random probability measure, with the randomness being induced by $\vx^{(0)}\_{0}, \dots, \vx^{(KT-1)}\_0 \ \iidsim \ \mu_0$ and $S \sim Unif(\\{0, \dots, T-1\\})$.

---

> > ### Comment · Reviewer_9R2A · 2023-08-22
> >
> > Thank you for your time and effort to address the questions and concerns of all reviewers. I am satisfied with your response and will maintain my score.

---

### Author Rebuttal · Authors · 2023-08-08

$\newcommand{\cF}{\mathcal{F}} \newcommand{\vx}{\mathbf{x}} \newcommand{\mub}{\bar{\mu}} \newcommand{\KL}[2]{KL\left(#1||#2\right)} \newcommand{\bE}{\mathbb{E}} \newcommand{\KSDs}[3]{KSD^{2}\_{#1}(#2||#3)}$
We thank the reviewers for their thoughtful comments & feedback that have significantly helped improve our work. We have updated our draft to incorporate their suggestions (e.g. adding a table of past works as suggested by Reviewer 9R2A, further discussion of limitations & future work as suggested by Reviewer T9Sw).

**Motivated by Reviewer T9Sw's suggestions, we show that for both VP-SVGD & GB-SVGD, the exponent of $n$ in the finite-particle rates (Corollary 2 & 3) can be made dimension free**. We note that the key lemmas required to prove this were already present in our work. We discuss this further below.

**If permitted by the Area Chair & the reviewers, we would be happy to share a fully anonymized version of our updated draft**

We answer some common questions raised by the reviewers below & address specific queries in the individual responses

## Comparing VP-SVGD to SVGD & Concept of Virtual Particles

The reviewers asked several insightful questions on the similarities & differences between SVGD & VP-SVGD, & on the motivation behind virtual particles. We hope to address these as follows:

We recall several prior works ([1,2,3,4]) which show that SVGD is a finite-particle approximation of the kernelized/projected Wasserstein gradient descent of the KL divergence (i.e. $\KL{.}{\pi^*}$), in the infinite dimensional space of probability measures. This approximation becomes exact in the mean-field (i.e. infinite particle) limit, such that the mean-field SVGD update is given  by  $\mub_{t+1} = (I - \gamma h_{\mub\_t})\_\\# \mub_{t+1}$ ($\mub_t$ is a probability measure & $h_{\mub_t}$ is the kernelized Wasserstein Gradient, See Sec. 2 & 3).

The mean-field update is not computationally implementable as it describes a dynamical system in an infinite dimensional space. To this end, SVGD was originally derived in [1] by first deriving the mean-field dynamics, & then deploying a tractable particle-based Monte Carlo approximation of the update ([1, Sec. 3.2], [2, Sec. 2], [3, Sec. 2.4.1]). Notably, at iteration $t$, SVGD uses $n$ particles $(\vx^{(l)}\_{t})\_{0 \leq l \leq n-1}$ to approximate $\mub_t \approx \tfrac{1}{n} \sum\_{l=0}^{n-1}  \delta_{\vx^{(l)}\_{t}}$ & **reuses the same particles** to approximate $h_{\mub\_{t}} \approx \tfrac{1}{n} \sum_{l=0}^{n-1} h(., \vx^{(l)}\_{t})$. This is a **biased discretization** of the mean-field update (as $Law(\vx^{(l)}\_{t}) \neq \mub_t$ & ${h}\_{\mub\_{t}} \neq \tfrac{1}{n} \sum_{l=0}^{n-1} \bE[h(., \vx^{(l)}\_{t})]$) due to its complex inter-particle dependencies.

VP-SVGD, like ordinary SVGD, is also a particle based approximation of the mean field SVGD updates. The key difference between the two is that **VP-SVGD is an unbiased stochastic approximation (in the space of measures) to the dynamics of mean-field SVGD**, which can be implemented exactly with only finite particles (see Section 1.2 Lines 99-109 & Section 4). This is achieved by using **$K$ virtual particles in each iteration to construct an unbiased stochastic approximation of $h\_{\mu_t | \cF_t}$.** (via the function $g_t$) & **$n$ real particles to approximate $\mu_t | \cF_t$**. From the lens of finite dimensional optimization, this is similar to how Stochastic Gradient Descent is an unbiased stochastic approximation to population-level Gradient Descent. To justify this, we recall the VP-SVGD update $\mu_{t+1}|\cF_{t+1} = (I - \gamma g_t)\_\\# \mu_t | \cF_t$. Note that, when conditioned on the virtual particles: 1. $\bE[g_t(\vx) | \cF_t] = h\_{\mu_t | \cF_t}$ (i.e. the dynamics is an unbiased stochastic approximation) 2. The $n$ real particles $(\vx^{(l)}\_t)\_{KT \leq l \leq KT+n-1}$ are i.i.d draws from $\mu\_t | \cF\_t$ (discussed in Section 4 and proved in Appendix C). To this end, **the use of virtual particles allows VP-SVGD to exactly track the time-evolution of $\mu_t | \cF_t$ using only finite particles. Additionally, it ensures that the particles $(\vx^{(l)}\_t)\_{KT \leq l \leq KT+n-1 }$ are conditionally i.i.d samples from $\mu_t | \cF_t$**. This allows us to derive fast finite-particle convergence rates.

## Dimension-Free $n$ Exponent in Finite-Particle Rates

Based on the insightful feedback of Reviewer T9Sw, we show that the exponent of $n$ in our finite-particle rates can be made dimension-free. For subgaussian $\pi^*$, we show that VP-SVGD has a finite-particle KSD rate of $O(\tfrac{d^{1/4}}{n^{1/4}} + \tfrac{d^{1/2}}{n^{1/2}})$ while GB-SVGD has a finite-particle KSD rate of $O(\tfrac{d^{1/3}}{n^{1/12}} + \tfrac{d^{1/2}}{n^{1/2}})$. We believe the constant factors in the exponents (for both algorithms) may be further improved via careful analysis. We sketch the proof of this result in our response to Reviewer T9Sw and our updated draft presents a complete proof for more general targets (including subexponential $\pi^*$ and beyond). This result uses two novel techniques: 1. A new KSD bound between the empirical measures of the outputs of VP-SVGD & GB-SVGD 2. A direct approach for bounding $\bE[\KSDs{\pi^*}{\hat{\mu}^{(n)}}{\pi^*}]$  in terms of $\bE[\KSDs{\pi^*}{\mub}{\pi^*}]$ (where $\hat{\mu}^{(n)}$ is the empirical measure of VP-SVGD outputs and $\bar{\mu}$ is the population measure) which does not invoke any empirical-to-population Wasserstein bounds and consequently avoids the $n^{-\Theta(1/d)}$ dependence of Lemma 8 (which is unimprovable in general, see [5, Sec. 3] and [6])

1. Liu and Wang'16, SVGD
2. Liu'17, SVGD as Gradient Flow
3. Salim et. al.'21,  Convergence Theory for SVGD in Pop. Limit
4. Korba et. al.'20,  Nonasymptotic Analysis for SVGD
5. Lei'18, Convergence of Empirical Measures under Wasserstein Distance
6.  Fournier. et.al.'13, Rate of Convergence in Wass. Distance of the Empirical Measure

---

### Decision · Program_Chairs · 2023-09-21

**Decision:**

Accept (spotlight)

**Comment:**

This paper make significant progress on a major open problem in the Stein variational gradient descent literature: namely, finite-particle convergence guarantees for SVGD algorithms.